# Minimizing interfacial energy losses via multifunctional cage-like diammonium molecules for efficient perovskite/silicon tandem solar cells

Xin Li[1,2], Zhiqin Ying ®[1] ✉, Linhui Liu[1], Jun Wu[1], Haofan Ma[1], Ziyu He[1], Yunyun Yu[1], Yihan Sun[1], Meili Zhang[1], Xuchao Guo[1], Yuheng Zeng ®[1], Xi Yang ®[1,3] ✉ & Jichun Ye ®[1,3] ✉

Wide bandgap (WBG) perovskites hold tremendous potential for enabling efficient perovskite/silicon tandem solar cells. However, interfacial energy losses at the perovskite/electron selective contact interface remain a substantial obstacle in approaching its theoretical efficiency limit. Herein, for the first time, a multifunctional cage-like diammonium chloride molecule, featuring Lewis acid/base groups and strong molecular polarity, is designed to reduce film defects and modulate the interfacial dipole, thereby suppressing non-radiative recombination and optimizing surface band alignment. More importantly, the unique cage-like cation can induce the formation of a phase-pure quasi-2D perovskite with spontaneous in-plane orientation and exhibits a pronounced ferroelectric effect, facilitating carrier further apart and extraction by upshifting the surface work function. Consequently, we achieve 1.68 eV perovskite solar cells with power conversion efficiencies (PCEs) of 22.6% (0.1 cm²) and 21.0% (1.21 cm²). Furthermore, two-terminal monolithic perovskite/silicon tandem solar cells based on tunnel oxide passivating contact yield an impressive PCE of 31.1% (1.0 cm²) and demonstrate a decent operational stability (ISOS-L-1, $T_{85}$ > 1020 h in ambient conditions without encapsulation). The ferroelectric interface physics opens new possibilities for efficient and stable perovskite-based tandem photovoltaics.

Monolithic two-terminal perovskite/silicon tandem solar cells (PVSK/Si TSCs) have emerged as one of the most promising candidates for achieving exceptionally efficient photovoltaic devices along with cost-effective manufacturing, and skyrocketed to a record power conversion efficiency (PCE) of 34.6%[1], exceeding the theoretical limit of any single-junction solar cell[2]. Despite this progress, there is still considerable room for improvement relative to its theoretical efficiency limit of 45.1%, especially concerning the wide bandgap (WBG) perovskite top cells[3]. In tandem applications, WBG top cells still suffer from undesirably large open-circuit voltage ($V_{OC}$) and fill factor (FF) deficits when compared to narrow-bandgap counterparts[4,5], which dominantly stems from severe charge carrier recombination and

¹Zhejiang Provincial Engineering Research Center of Energy Optoelectronic Materials and Devices, Ningbo Institute of Materials Technology and Engineering, Chinese Academy of Sciences, Ningbo, China. ²University of Chinese Academy of Sciences, Shijingshan District, Beijing, China. ³Research Center for Wide Bandgap Semiconductors and Devices, YongJiang Laboratory, Ningbo, Zhejiang, China. ✉e-mail: yingzhiqin@nimte.ac.cn; yangx@nimte.ac.cn; jichun.ye@nimte.ac.cn

energy level mismatch at the perovskite interfaces with charge transport layers, particularly accentuated at the perovskite/$C_{60}$ contact interface[6–8]. The first monolayers of $C_{60}$ can introduce deep trap states at the interface through energy-level pinning and band misalignment, exacerbating across-interface recombination losses[6,8].

To this end, tremendous efforts concerning surface post-treatment for perovskite have been dedicated, such as inserting metal fluorides or oxides at perovskite/$C_{60}$ interface to yield a contact displacement[9–11]. Despite their efficacy, charge transport across the interface is still limited due to the intrinsically insulating nature of these substances, thereby necessitating a precise control of the spacer thickness[9]. Additionally, these passivators are usually mono-functionalized, demonstrating either single chemical or field-effect passivation, which is insufficient to eliminate the $C_{60}$-induced recombination losses. This also holds for the large organic monoammonium ligands[12–15], featuring with single Lewis acid/base group and insulation. They can indeed inhibit non-radiative recombination to a certain extent, yet this may limit the ability to block minority charges without compromising the charge transport due to the unapplicable band alignment in inverted p-i-n configuration[16]. The uncontrollable out-of-plane orientation growth of the low-dimensional perovskite may also reduce the charge mobility. Driven by this, binary molecules synergistic passivation strategies for perovskite films have been extensively attempted to achieve the best balance between minimizing recombination loss and restricting contact resistance[17–21]. However, co-deposition or sequential deposition of multiple surface-modifying molecules remain challenging due to the complicatedly competing interactions with the perovskite. Therefore, it is imperatively needed to rationally design a multifunctional single molecule that can simultaneously address the above issues, while such molecules have been rarely exploited[4,22,23].

In this work, a novel multifunctional cage-like diammonium chloride molecule, 1,4-diazabicyclo[2.2.2]octane chloride (DCl), is synthesized and intercalated into the perovskite/$C_{60}$ interface to minimize energy losses. Specifically, the cage-like diammonium cation, containing both a Lewis acid ($R_3NH^+$) and a Lewis base ($R_3N$) group, is capable of functioning as an electron acceptor and an electron donor to react with different surface defects, suppressing non-radiative recombination loss. Moreover, the spontaneously polarized cage-like cation induces a p-to-n type surface energetic transformation and forms a positive interfacial dipole, further inhibiting across-interface recombination and enhancing charge transport. More importantly, the in-plane oriented, phase-pure quasi-2D perovskite induced by the cage-like diammonium cation is endowed with pronounced ferroelectricity, promoting carrier further separation and extraction by upshifting the surface work function. Based on its multifunctionality, the DCl-treated inverted 1.68 eV perovskite solar cells achieve a champion PCE of 22.6% for 0.1 $cm^2$ and 21.0% for 1.21 $cm^2$, respectively. Meanwhile, for the first time, we also incorporate this ferroelectric quasi-2D perovskite into monolithic perovskite/silicon tandem solar cells, eventually obtaining an impressive PCE of 31.1% (1.0 $cm^2$) accompanied by an excellent long-term operational stability (85.4% retention after 1020 h under ISOS-standardized maximum power point tracking).

## Results

### Construction of the DCl-mediated quasi-2D perovskite

In this work, a cage-like diammonium chloride molecule, 1,4-diazabicyclo[2.2.2]octane chloride (DCl), is synthesized (Supplementary Fig. 1) and intercalated into the perovskite/$C_{60}$ interface to reduce the surface/interface recombination and the energetic mismatch. Supplementary Fig. 2 depicts the chemical structure, electrostatic potential (ESP) and corresponding electric dipole moment of the cage-like diammonium cation. Similar to other cyclic ammonium cations[22,24], the cage-like diammonium cation contains both electron donor ($R_3N$) and electron acceptor ($R_3NH^+$) functional groups, which can passivate both negative and positive surface trap states (Supplementary Fig. 3), thus reducing nonradiative recombination. Additionally, the formation of molecular dipole (4.08 Debye) due to the spontaneous polarity of the cage-like diammonium cation can improve surface energy band alignment of perovskite by altering the surface work function[25]. Moreover, DCl itself demonstrates exceptionally structural stability and space charge transfer properties[26], thus, it can largely improve the long-term stability of underlying perovskite and device performance.

Top-view scanning electron microscopy (SEM) was first employed to characterize the alterations in film morphology before and after DCl post-treatment. As shown in Fig. 1a, a number of discrete sheet-like $PbI_2$ grains filling the grain boundaries of the 3D perovskite in the control film is observed. Interestingly, after DCl post-treatment, distinct morphological differences are presented. The perovskite surface undergoes reconstruction, forming numerous small grains uniformly distributed at the perovskite grain surfaces, accompanied by the disappearance of residual $PbI_2$. These results indicate that cage-like diammonium cation, similar to fluorinated plane-type ammonium cations, can also react with $PbI_2$ and 3D perovskite phase to generate new substances, spontaneously assembling on the top surface of the 3D perovskite[24,27]. Surface topology of the films without and with DCl post-treatment was then characterized using atomic force microscope (AFM). As demonstrated in Fig. 1b and Supplementary Fig. 4, consistent with the SEM images, numerous uniformly assembled small grains are distinctly observed on the DCl-treated perovskite surface. In addition, the DCl-treated film exhibits a slightly decreased root mean square (RMS) surface roughness than the control one (Fig. 1b and Supplementary Fig. 4), which optimizes the interfacial contact of the perovskite with electron transport layer[28].

General X-ray diffraction (XRD) measurements were subsequently carried out to reveal the crystal structure evolution after DCl post-treatment, as depicted in Supplementary Fig. 5. No new phase was formed at low angles as the concentration of DCl raises from 0.1 $mg\,mL^{-1}$ to 0.6 $mg\,mL^{-1}$, and the best crystallinity of perovskite film is achieved at 0.4 $mg\,mL^{-1}$. Grazing incidence XRD (GIXRD) measurement (0.2° grazing incidence angle) was further conducted to reveal the product of perovskite surface upon DCl treatment (Supplementary Fig. 6). Careful observation, reveals a clearly new peak at 10.2° in the DCl-treated film, indicating the formation of pure-phase quasi-2D perovskite with $n = 3$ (n is the number of semiconducting $[PbX_6]^{4-}$ octahedron slabs within the cage-like diammonium spacers) (Supplementary Fig. 7)[29]. Quasi-2D perovskite with $n \geq 3$ usually exhibits mobilities similar to 3D perovskite, thus no inhibition of carrier transport[30]. Grazing-incidence wide-angle X-ray scattering (GIWAXS) measurements further revealed the crystal orientation of this quasi-2D perovskite, as shown in Fig. 1c, d. Most strikingly, the diffraction arcs ($q_{xy} = 0.72\,Å^{-1}$) associated with the quasi-2D perovskite is detected only at the $q_{xy}$ axes and absent at the $q_z$ axes (Fig. 1e and Supplementary Fig. 8), which indicates an in-plane orientation of the quasi-2D perovskite, i.e., growing perpendicular to the bulk perovskite surface. Differing from conventional chain-like or plane-related amine cations such as n-butylammonium or phenylethylamine cations typically featuring out-of-plane orientation on 3D perovskite surface[31,32], such cage-like diammonium cation can spontaneously induce the formation of pure-phase quasi-2D perovskite with an in-plane crystal orientation, which is favorable for interfacial charge transport in inverted perovskite solar cells[30,33]. The vertically oriented growth of the DCl-mediated pure-phase quasi-2D perovskite can be the result of Cl incorporation partially substitutes I in the equatorial position of the unit cell, leading to a vertical strain in the perovskite octahedra[34,35]. It is additionally revealed that the spatial dimensionality of cations is pivotal in dictating the orientation of 2D perovskites. Comparatively, 3D caged diammonium cation is more inclined to form in-plane oriented 2D perovskite (Supplementary Figs. 9, 10).

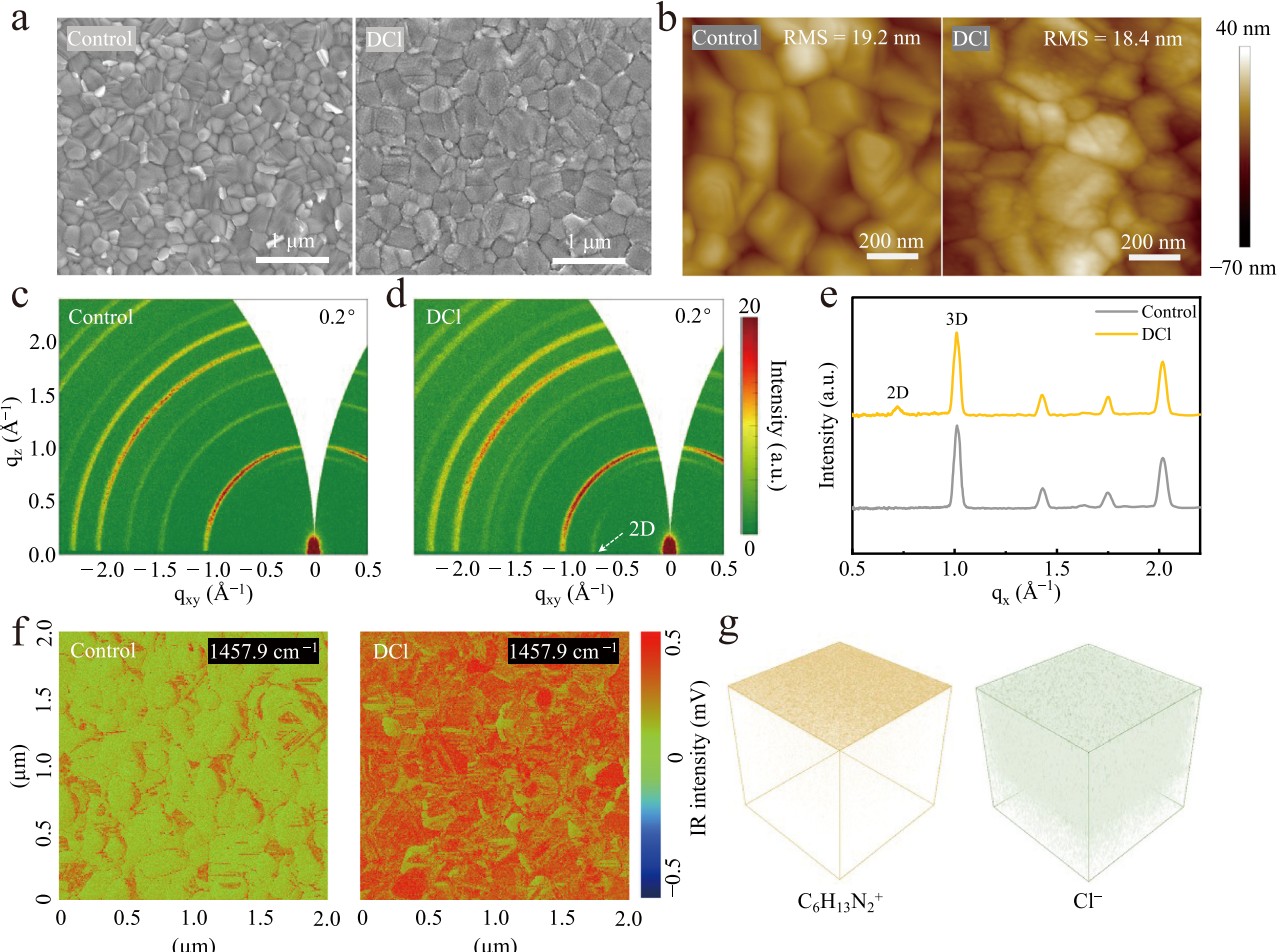

**Fig. 1 | Morphology and crystallinity of perovskite films. a** Top-view SEM images of the (left) control and (right) DCl-treated perovskite films. **b** AFM images of the (left) control and (right) DCl-treated perovskite films. **c**, **d** 2D GIWAXS patterns of the control and DCl-treated perovskite films with a grazing incidence angle of 0.2°. **e** 1D-GIWAXS profiles along the in-plane direction ($q_x$) for the control and DCl-treated perovskite films at 0.2° incidence angle. **f** AFM-IR images of the control and DCl-treated perovskite films. **g** ToF-SIMS 3D images of the cage-like diammonium cation ($C_6H_{13}N_2^+$) and chloride anion ($Cl^-$).

We subsequently employed atomic force microscopy-based infrared (AFM-IR) spectroscopy to reveal the nanoscale distribution and coverage of DCl molecules on perovskite surfaces (Supplementary Fig. 11). As shown in Fig. 1f, no distinct IR signal associated with DCl is detected on the control perovskite surface, whereas the DCl-treated film exhibits nearly full IR signal coverage across both grain surfaces and boundaries, indicating a complete coating, conducive to uniform surface passivation and interfacial dipole regulation. To further explore the diffusion properties of DCl, time-of-flight secondary-ion mass spectrometry (Tof-SIMS) was performed (Supplementary Fig. 12). The results reveal a gradient distribution of cage-like diammonium cations, with strong enrichment at the top surface that sharply decreases with probe depth, likely stemming from large steric hindrance effect and indicating an ultrathin capping layer[36]. In stark contrast, chloride anions migrate downward and ultimately distribute throughout the entire perovskite film, with primary enrichment at both interfaces. As a result, a well-defined cation/anion spatial distribution is established (Fig. 1g), achieving synergistic defect passivation through multifunctional cage-like diammonium cations coordination and Pb-Cl ionic bonding, ultimately suppressing ion migration[36].

Commonly, surface coverage of quasi-2D perovskite is proved to mitigate ion migration[14]. We thus conducted conductive atomic force microscopy (c-AFM) measurements to reveal the surface conductivity disparity after DCl post-treatment. As illustrated in Supplementary Fig. 13, the current map along with the corresponding line profile

collected from the control film clearly shows that GBs present higher conductivity as compared with GIs, indicating higher ion migration activity at the GBs, being accordance with pervious findings that GBs dominate the ion migration pathway in polycrystalline film[37,38]. In contrast, no significant conductivity differences between the GBs and GIs can be observed for the DCl perovskite surface, and the DCl film, in the entire scanning region, seems to have a lower current than the control one (Supplementary Fig. 14), strongly demonstrate the suppression of ion migration induced by DCl modification. This is further verified by temperature-dependent conductivity (Supplementary Fig. 15) and transient ionic current (Supplementary Fig. 16) measurements. The lower current flowing through the perovskite grains for the DCl sample may be attributed to the higher resistance of the low-dimensional phase[39].

## Defects passivation effects of DCl
To elucidate the interaction between perovskite and DCl, we conducted density functional theory (DFT) calculations to determine the binding energies of DCl adsorption. For simplicity, the DCl was categorized into two functional groups: $R_3N$ and $R_3NHCl$. Four potential adsorption configurations on the $PbI_2$-terminated and FAI-terminated (100) perovskite surfaces were considered, with the corresponding adsorption energies ($E_{ad}$) presented in Supplementary Fig. 17. Regarding the FAI termination (Supplementary Fig. 17a), the $E_{ad}$ values of $R_3N$/FAI and $R_3NHCl$/FAI are −0.42 and −0.50 eV, respectively, indicating comparable and relatively weak binding. On

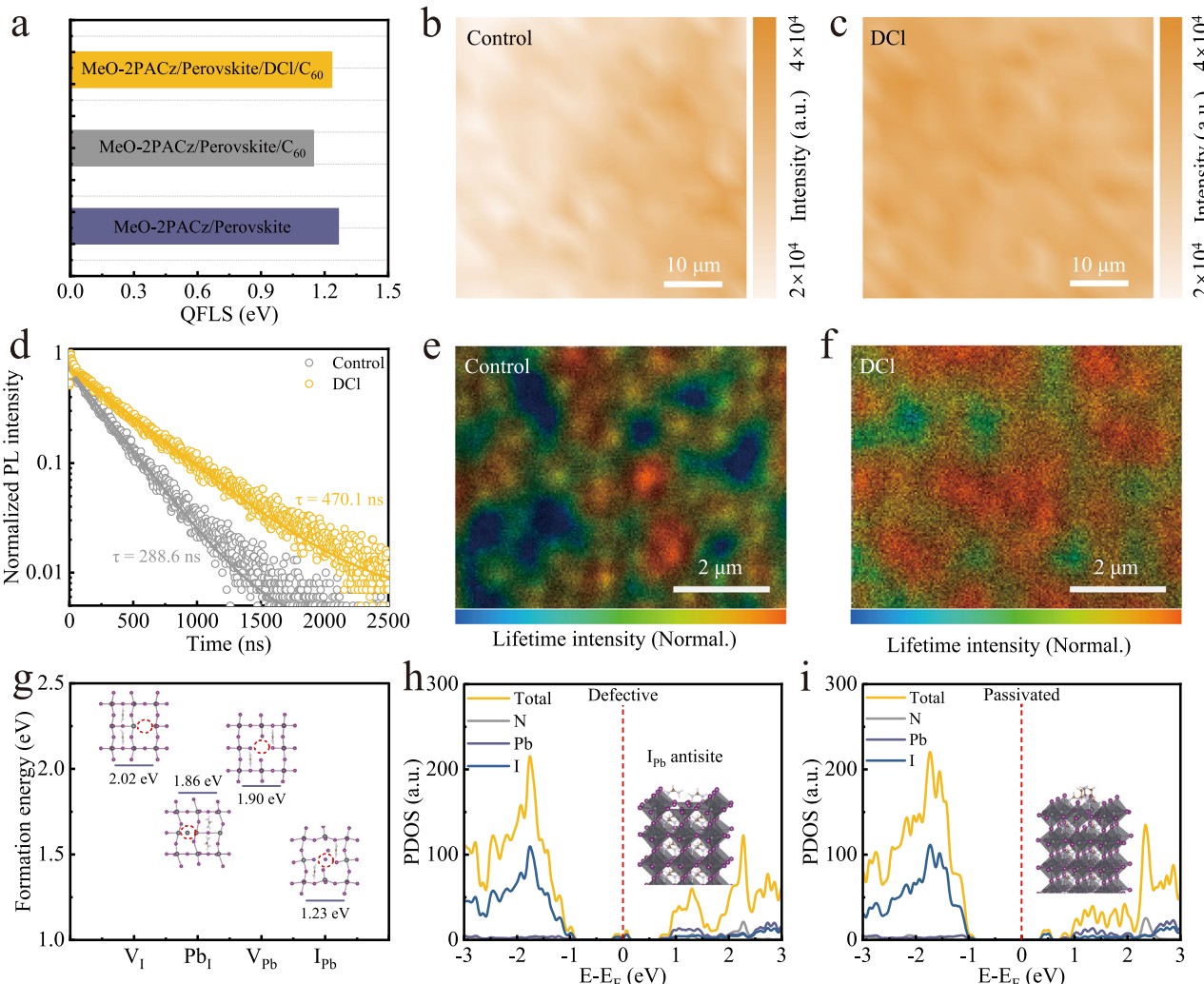

**Fig. 2 | Defect passivation of perovskite films. a** QFLS diagram for MeO-2PACz/perovskite film, MeO-2PACz/perovskite/$C_{60}$ half stack and MeO-2PACz/perovskite/DCl/$C_{60}$ half stack. PL maps of the (**b**) control and (**c**) DCl-treated perovskite films. **d** TRPL spectra of the control and DCl-treated perovskite films. Carrier lifetime images based on CLSM of the (**e**) control and (**f**) DCl-treated perovskite films. **g** Defect formation energies of various types of defects on perovskite surface (the inset represents the top view). Partial density of states (PDOS) of the (**h**) defective perovskite ($I_{Pb}$ antisite) and (**i**) passivated perovskite by DCl.

this surface, the interactions are mainly through weak hydrogen bonding (e.g., $R_3N \cdots FA^+$, $Cl \cdots FA^+$ and $R_3NH^+ \cdots I$), as supported by liquid-state $^1H$ NMR spectra (Supplementary Fig. 18a). In contrast, for the $PbI_2$ termination (Supplementary Fig. 18b), $R_3NHCl/PbI_2$ and $R_3N/PbI_2$ exhibit much stronger binding energies of −0.90 eV and −0.78 eV, respectively, indicating a clear preference for adsorption on $PbI_2$ termination and highlighting the primary interaction of DCl with $PbI_2$. Notably, in the $R_3NHCl/PbI_2$ configuration, strong Pb-Cl ionic bonding is observed, whereas the $R_3NH^+$ group exhibits only weak hydrogen bonds with I, without direct interaction with Pb. These results reveal that DCl predominantly interacts with perovskite via $R_3N$-Pb coordination and Pb-Cl ionic bonding, as further confirmed by high-resolution X-ray photoelectron spectroscopy (Supplementary Figs. 19–21 and Tables 1, 2). Furthermore, ToF-SIMS results reveal that cage-like diammonium cations accumulate at the perovskite surface, while chloride anions migrate downward (Fig. 1g). In this case, the spontaneously polarized cage-like cations directionally assemble at the perovskite/$C_{60}$ interface, forming a robust interfacial dipole layer that modulates band bending and alleviates energy level misalignment[40]. Concurrently, strong Pb-Cl ionic bonding effectively passivates bulk and interfacial defects, contributing to improved performance and stability[41].

To verify and understand the passivation effect of DCl, we first conducted photoluminescence quantum yield (PLQY) measurements, from which the internal electron-hole quasi-Fermi level splitting (QFLS) can be calculated, by spin-coating 1.68 eV perovskite $Cs_{0.05}(FA_{0.77}MA_{0.23})_{0.95}Pb(I_{0.77}Br_{0.23})_3$ films on top of indium tin oxide (ITO)/[2-(3,6-dimethoxy-9H-carbazol-9-yl) ethyl] phosphonic acid (MeO-2PACz) substrates. As shown in Fig. 2a, the perovskite deposited on the ITO/MeO-2PACz substrate exhibits an implied PLQY of 0.318%, corresponding to QFLS value of 1.265 eV. When capped with $C_{60}$, the PLQY of the MeO-2PACz/perovskite/$C_{60}$ stack is reduced from 0.318% to 0.0033%, leading to a derived QFLS of 1.147 eV, indicative of strong surface recombination[6]. In contrast, the $C_{60}$ induced PL quenching is substantially reduced for the DCl-treated MeO-2PACz/perovskite/DCl/$C_{60}$ stack. The PLQY of MeO-2PACz/perovskite/DCl/$C_{60}$ stack is increased from 0.0033% to 0.0949% (QFLS ~ 1.234 eV), indicating that the critical $C_{60}$-induced non-radiative recombination is considerably suppressed upon introducing DCl, confirming the passivation effect of DCl on perovskite.

To correlate the QFLS results with the dynamics of photo-generated charges, steady-state photoluminescence (PL) and time-resolved photoluminescence (TRPL) characterizations of MeO-2PACz/perovskite samples with and without DCl treatment were additionally

performed. The passivated perovskite film by DCl exhibits suppressed non-radiative recombination, as evidenced by the increased PL intensity (Supplementary Fig. 22). Furthermore, the spatial homogeneity of the perovskite films was intuitively evaluated by hyperspectral luminescence images (PL mapping) technique. As depicted in Fig. 2b, c, the fluorescence distribution in the DCl film is more uniform, accompanied by a higher fluorescence intensity (Supplementary Fig. 23), consistent with the PL results (Supplementary Fig. 22), which indicates a reduction in the surface trap states and the suppression of non-radiative recombination.

The enhanced PL intensity implies the prolonged charge carrier lifetimes, as confirmed by TRPL spectra in Fig. 2d, where all the PL transients are fitted with mono-exponential decay. A significant slower TRPL decay, corresponding to a longer carrier lifetime ($\tau$) of 470.1 ns, in the DCl-treated film is observed, indictive of an effective suppression of non-radiative recombination upon the introduction of DCl. Similarly, we then displayed the spatially resolved carrier lifetime images utilizing confocal laser scanning fluorescence microscopy (CLSM). As illustrated in Fig. 2e, f, the control film shows many blue regions with short carrier lifetimes, indicating a large number of unevenly distributed defects on the pristine perovskite surface[42], in agreement with the above PL mapping result. By stark contrast, almost all blue regions are converted to a yellow region with a longer carrier lifetime for the DCl-treated film, featuring a more uniform and higher carrier lifetime distribution in the entire detected region, indicative of largely reduced surface defects and suppressed non-radiative recombination. Notably, in comparison with the control film, the DCl-treated film exhibits less discrepancy of carrier lifetimes between GBs and GIs, which is benefited from the simultaneous passivation of GBs and GIs by the DCl-induced quasi-2D perovskite, as reflected by the SEM and AFM-IR images. In a word, the PL intensity and carrier lifetime images collectively verify the comprehensive defect passivation effect of DCl, homogenizing perovskite surface, which is beneficial for large-area perovskite film fabrication.

To gain deeper insights into the passivation mechanisms of DCl on the perovskite surface, we conducted DFT calculations considering four prevalent surface defects: I vacancy ($V_I$), Pb-I antisite ($Pb_I$), Pb vacancy ($V_{Pb}$), and I-Pb antisite ($I_{Pb}$) (Supplementary Fig. 24)[43]. Computational analysis reveals that deep-level defects, particularly $I_{Pb}$, exhibit lower formation energies than shallow-level defects (Fig. 2g). Consequently, both negatively charged $I_{Pb}$ and positively charged $Pb_I$ antisite defects form more readily and predominate on the surface, making them the primary non-radiative recombination centers. Upon DCl adsorption, deep-level states associated with $I_{Pb}$ defects are effectively inhibited (Fig. 2h, i), likely due to chemical interactions between negative halide traps and the $R_3NH^+$ group. A similar passivation effect is also observed on the $Pb_I$-dominated surface (Supplementary Fig. 25). Thermal admittance spectroscopy analysis further reveals the reduced deep trap states following DCl incorporation (Supplementary Fig. 26).

Moreover, since $C_{60}$ can induce the detrimental across-interface recombination[6], TRPL spectra of MeO-2PACz/perovskite/$C_{60}$ samples with and without DCl treatments was further measured and fitted with a biexponential decay with a fast and a slow component. As shown in Supplementary Fig. 27 and Table 3, the DCl sample exhibits a much longer lifetime of slow decay (∼ 43.0 ns) and shorter lifetime of fast decay (∼ 10.5 ns) in comparison with the control sample (∼ 22.9 and 10.8 ns, respectively), suggesting reduced across-interface non-radiative recombination and enhanced charge extraction at the perovskite/DCl/$C_{60}$ interface, further supported by DFT calculations. Supplementary Fig. 28 illustrates that charge accumulation within $C_{60}$ molecules is significantly repressed by DCl molecules. Additionally, the adsorption energy of $C_{60}$ on DCl-passivated (100) perovskite surfaces (−0.14 eV) is substantially lower than on bare perovskite surfaces (−0.02 eV), indicating stronger interaction,

reduced charge transfer resistance, and enhanced interfacial charge transfer[44].

## Energy band optimization and ferroelectric effect of DCl

The inhibited non-radiative recombination and enhanced charge extraction are commonly correlated with the improved surface energetics of perovskite film as well. Therefore, ultraviolet photoelectron spectroscopy (UPS) measurements were performed to seek a better understanding of the surface electronic structure for perovskite films with and without DCl treatment (Supplementary Fig. 29). The secondary electron cut-off (SECO) and the valence band maximum (VBM) spectra are depicted in Fig. 3a. The DCl treatment reduces the work function (WF) of perovskite surface from 4.03 eV to 3.82 eV and shifts the VBM from 0.77 eV below the Fermi level ($E_F$) in the control film to 1.04 eV below the $E_F$. These results show that DCl modification induces a transformation from $p$-type to $n$-type of perovskite surface energetics, which in inverted PSCs creates favorable downward band bending (i.e., forming an extra back-surface field) for enhancing electron extraction and hole blocking, thereby hindering across-interface nonradiative carrier recombination[28]. Moreover, it is also found that the ionization energy of the DCl-treated perovskite is increased by around 60 meV compared to the control film (Supplementary Fig. 30), suggesting the formation of a positive dipole on the DCl-modified perovskite surface[45]. Such positive dipole is conducive to reducing the downward offset between the lowest unoccupied molecular orbital (LUMO) of $C_{60}$ and the conduction band minimum (CBM) of perovskite, leading to a beneficial energy level alignment, thereby less voltage deficit[46].

The same shift trends of perovskite surface WFs were also verified by Kelvin Probe Force Microscopy (KPFM) measurements in the dark condition (Fig. 3b and Supplementary Fig. 31). The average contact potential difference (CPD) of DCl-treated film (−239.6 mV) is significantly raised compared to that of control film (−305.8 mV) (Supplementary Fig. 31), indicating a decrease in WF, which induces a transformation to more $n$-type perovskite surface and makes the $E_F$ located further away from the VBM, in agreement with the UPS results. The shallower $E_F$ of DCl-treated film is conducive to forming a larger gap between the splitting of electron quasi-Fermi level ($E_{fn}$) and hole quasi-Fermi level ($E_{fp}$) under light illumination, i.e., a larger QFLS, as evidenced by Fig. 2a. Moreover, DCl-functionalized perovskite displays a more uniform surface potential distribution ($\sigma = 9.0$ mV compared with 16.2 mV for the control) (Supplementary Fig. 31), which is beneficial for efficient charge transport and reducing the broadening of interfacial electronic states when in contact with $C_{60}$ to suppress nonradiative recombination[47]. Subsequent DFT calculations were conducted to explore the evolution of the perovskite surface work function (SWF). As shown in Fig. 3c, the control perovskite exhibits a high SWF of −1.95 eV. Upon adsorption of cage-like cations, the SWF dramatically deceases to −0.25 eV (Fig. 3d), corresponding to an upshift of 1.7 eV, consistent with UPS and KPFM measurements. Additionally, the introduction of cage-like cations induces a strong interfacial dipole originating from their spontaneous polarization (Fig. 3d). The charge density difference reveals that this interfacial dipole significantly drives charge transfer from the perovskite surface to the cage-like cations, thereby decreasing the SWF (Fig. 3e)[48].

Early reports have revealed that perovskite films capped with organic cations featuring large dipole moment or incorporated with fluorinated polymers with a strong permanent polarity can be potentially endowed with ferroelectricity upon ferroelectric polarization using an external electric field[24,27,49]. The presence of ferroelectricity can synergistically enhance charge extraction and collection while suppressing recombination by driving photogenerated charges further separation. Our cage-like diammonium DCl molecule possesses a spatial asymmetric structure induced by the Lewis base/acid substituents, forming an intrinsic molecular polarity, which provides a prerequisite for the spontaneous polarization of ferroelectric

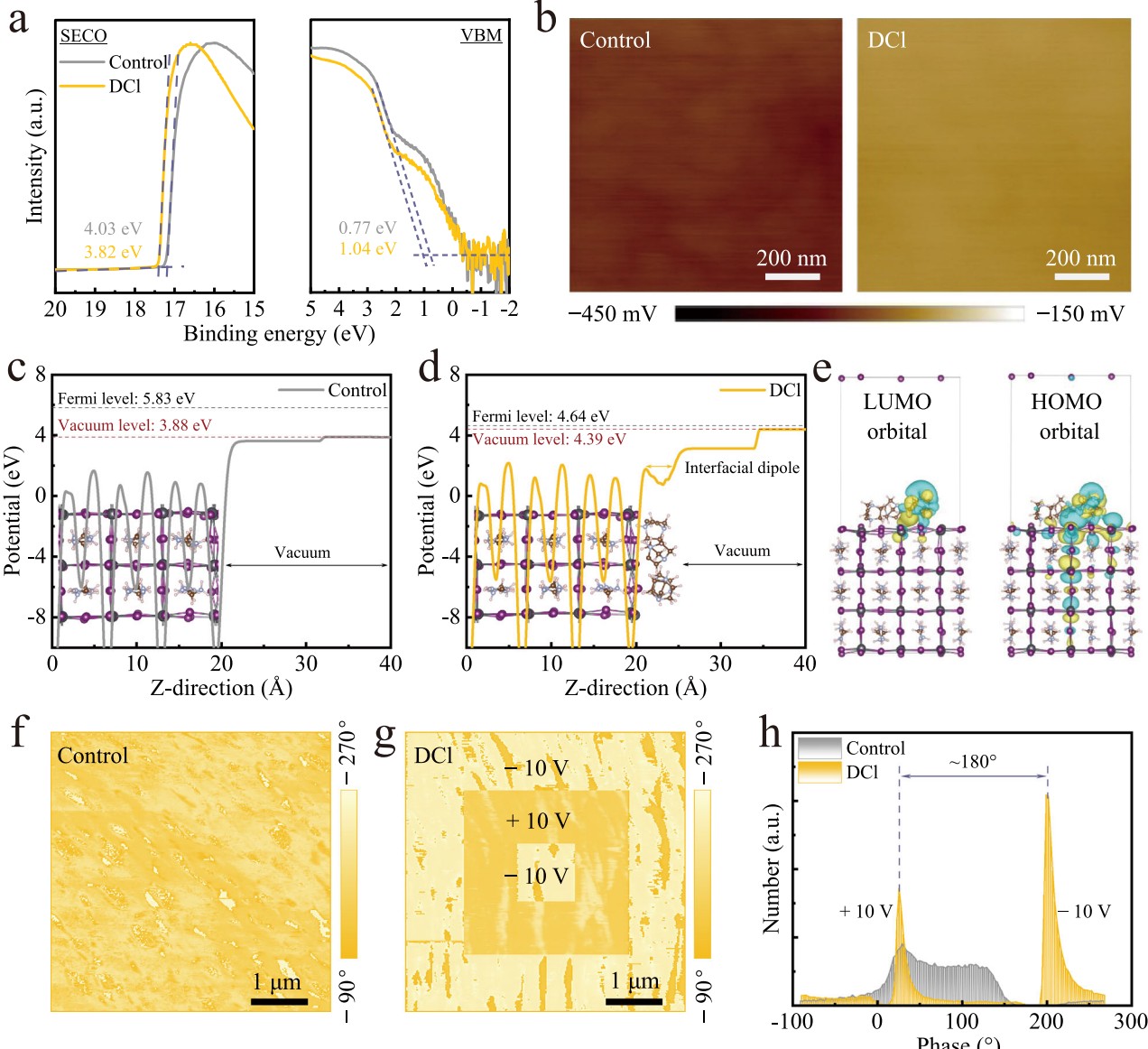

**Fig. 3 | Surface work function and ferroelectric effect of perovskite films.**
**a** Secondary electron cut-off (SECO) and valence band maximum (VBM) of the UPS spectra for the control and DCl-treated perovskite films. **b** KPFM images of the (left) control and (right) DCl-treated perovskite films. **c**, **d** DFT-calculated electrostatic potential and surface work functions of perovskites without and with the adsorption of cage-like cations. **e** Charge density difference of perovskite with cage-like cations adsorption. Yellow iso-surface signifies electron accumulation, while blue represents electron depletion. PFM written-domains images of the (**f**) control and (**g**) DCl-treated perovskite films with ±10 V bias. **h** Phase distribution histogram of the control and DCl-treated perovskite films.

materials[27]. Moreover, it has proved that organic ferroelectrics based on cage-like molecules displays the fastest polarization switching due to its unique multiaxial ferroelectric nature[50].

In this context, piezoresponse force microscopy (PFM) measurement, a typical method to enable the detection of the piezoelectric response of samples[27], was conducted to verify the ferroelectric effect of the DCl-treated surface. The phase and amplitude of PFM vibrational signal depends on the polarization direction and on the intensity of the ferroelectric domains within the material, respectively. Figure 3f, g intuitively display the PFM written-domains images of the control and DCl-treated films with externally applied polarization bias voltages of ±10 V. No phase contrast is detected for the control film when switching the polarization bias voltage, indicating the absence of ferroelectric behavior of the intrinsic perovskite film. However, a reversal of ferroelectric domain direction for the DCl film is obviously observed, exhibiting an exactly opposite phase contrast, which confirms the ferroelectric effect of

the DCl post-treated films[27]. For a more quantitative analysis, the phase signals across the entire written-domains images were additionally extracted as depicted in Fig. 3h. As expected, two peaks with a nearly 180° phase angle difference is presented for the DCl-treated film upon changing the sign of the applied external voltage and absent for the control film, which implies that the DCl-treated surface can be polarized in two completely opposite directions along with the external electric field direction. Additionally, an evidently increased amplitude contrast in the polarized DCl perovskite film is also found (Supplementary Figs. 32, 33), further confirming the strong ferroelectric response of the DCl surface[51]. Overall, these measurements verify the ferroelectric effect of the DCl-treated perovskite surface, which can be attributed to the formation of DCl-mediated quasi-2D perovskite. Notably, to our best knowledge, this is the first reported ferroelectric quasi-2D perovskite based on 3D cage-like dipole molecule (Supplementary Table 4), which opens a new avenue for photo-ferroelectric interface engineering.

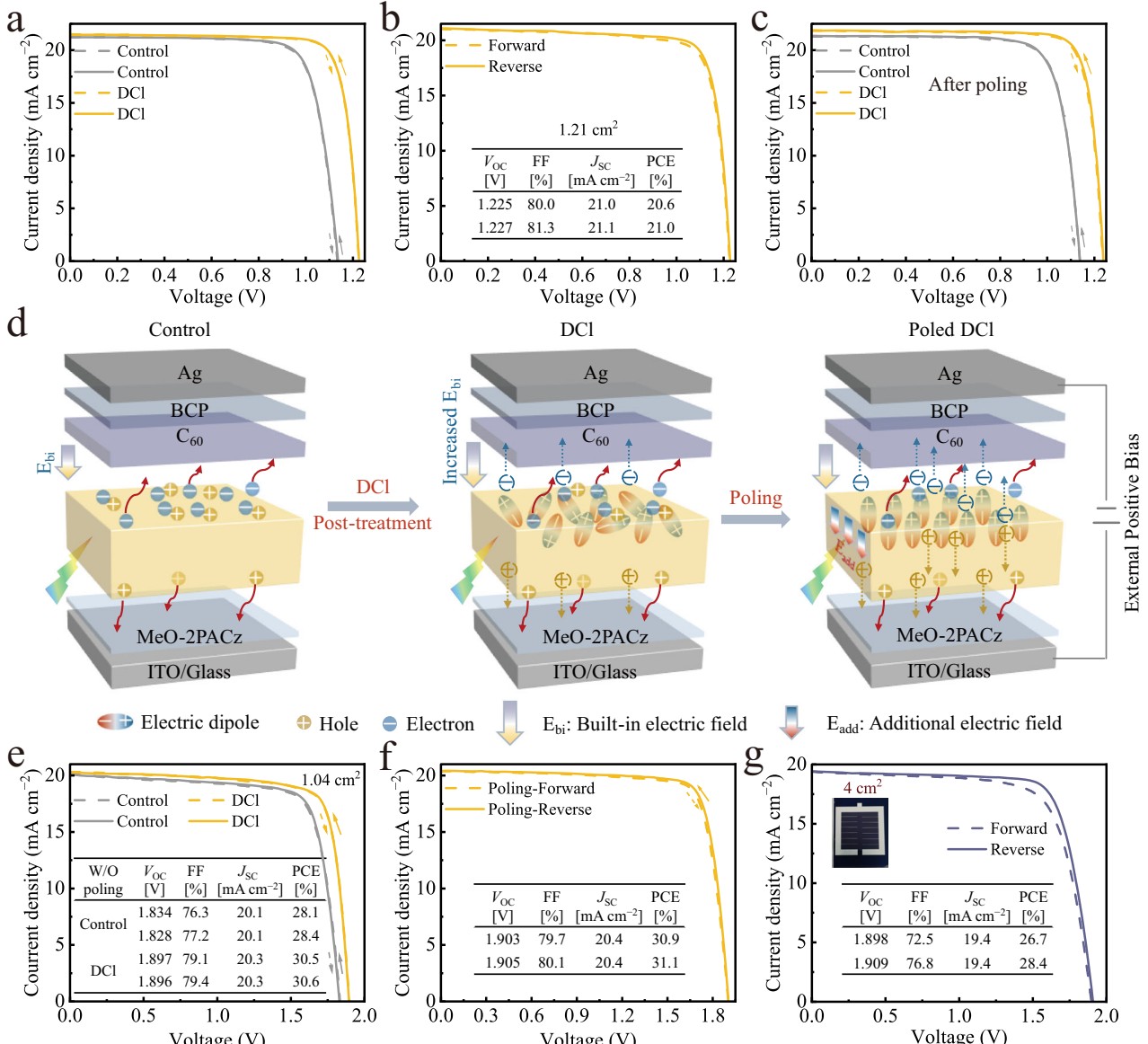

**Fig. 4 | Device performance and ferroelectric polarization mechanism. a** *J–V* curves of the champion control and DCl-treated perovskite solar cells before poling. **b** *J–V* curves of the champion DCl-treated perovskite solar cells with an aperture area of 1.21 cm². **c** *J–V* curves of the champion control and DCl-treated perovskite solar cells after 0.5 V positive poling for 5 min. **d** Schematic diagram of polarization principle induced by DCl-mediated quasi-2D perovskite. **e** *J–V* curves of the champion control and DCl-treated perovskite/silicon TSCs before poling. **f** *J–V* curves and corresponding PV parameters of the champion DCl-treated perovskite tandem after poling. **g** *J–V* curves of the large-area perovskite/silicon TSCs with an active area of 4 cm². The interfacial energy losses at the perovskite/electron selective contact interface remain an obstacle for perovskite/silicon tandem solar cells to approach its theoretical efficiency limit. Here, authors employ a cage-like diammonium chloride molecule to modulate interfacial dipole.

## Performance of perovskite single junction and tandem devices

To assess the effects of this multifunctional cage-like dipole molecule on device's PV performance, we subsequently integrated it in working perovskite solar cells (PSCs) with a configuration of Glass/ITO/MeO-2PACz/$Cs_{0.05}(FA_{0.77}MA_{0.23})_{0.95}Pb(I_{0.77}Br_{0.23})_3$/$C_{60}$/BCP/Ag (shown in Supplementary Fig. 34). The optimal DCl concentration is determined to be 0.4 mg mL$^{-1}$ (Supplementary Fig. 35 and Table 5). Figure 4a illustrates the current density versus voltage (*J–V*) curves of the champion control and DCl-treated PSCs. Obviously, the introduction of DCl drastically boosts the power conversion efficiency (PCE) of PSCs from 19.0% to 21.9%, with simultaneously improved open-circuit voltage ($V_{OC}$) to 1.226 V, fill factor (FF) to 83.2%, short-circuit current density ($J_{SC}$) to 21.5 mA cm$^{-2}$ (Supplementary Table 6) and increased steady-state PCE to 21.3% (Supplementary Fig. 36). The $J_{SC}$ values obtained from *J–V* curves are further verified by the external quantum efficiency (EQE)

spectra and match well with the integrated photocurrents (20.7 mA cm$^{-2}$ for control and 20.9 mA cm$^{-2}$ for DCl) (Supplementary Fig. 37). The slight EQE reduction between 400–480 nm in the DCl-treated device likely stems from its comparatively higher reflectance (Supplementary Fig. 38)[52]. The high reproducibility of DCl-treated PSCs is also confirmed, as shown in Supplementary Fig. 39 and Table 7. The DCl-treated PSCs present a narrower distribution of PCEs in comparison with the control PSCs, which can be attributed to comprehensively uniform passivation of DCl on the perovskite surface. Based on this, a large area device with an aperture area of 1.21 cm² reaches a PCE of 21.0% with a high $V_{OC}$ of 1.227 V, an FF of 81.3% and a $J_{SC}$ of 21.1 mA cm$^{-2}$ (Fig. 4b). Finally, DCl was successfully applied to 4PADCB-based, 1.68 eV MA-free and 1.79 eV wide-bandgap perovskites, yielding PCEs of 22.9%, 22.9% and 19.4% (Supplementary Fig. 40), respectively, corroborating its universality in various perovskite devices.

To understand the underlying mechanism for the enhanced PV performance of DCl-treated PSCs in depth, electrochemical impedance spectroscopy (EIS) were first conducted to assess charge transport and recombination dynamics. As illustrated in Supplementary Fig. 41 and Table 8, the DCl-treated device exhibits a reduced charge transport resistance ($R_{ct} = 3.5 \times 10^4$ Ω) and an increased recombination resistance ($R_{rec} = 8.6 \times 10^5$ Ω) compared to the control device, signifying ameliorated charge transport and suppressed charge recombination. Consistent results were observed in transient photocurrent (TPC) and transient photovoltage (TPV) measurements (Supplementary Fig. 42), where the DCl-treated device displays a shorter carrier transport lifetime ($\tau_{TPC} = 0.55$ μs) and a prolonged carrier recombination lifetime ($\tau_{TPV} = 4.31$ μs). Additionally, space charge limited current analysis using electron-only devices (Glass/ITO/ALD-SnO$_x$/perovskite/C$_{60}$/BCP/Ag) reveals a reduction in the trap filling limit voltage ($V_{TFL}$) from 0.19 V to 0.08 V, corresponding to a decrease in trap density ($N_{trap}$) from $2.59 \times 10^{15}$ cm$^{-3}$ to $1.09 \times 10^{15}$ cm$^{-3}$ (Supplementary Fig. 43). These results evidence that DCl treatment facilitates efficient charge transport and reduces defect density, and thus enhancing $V_{OC}$ and FF. This improvement is ascribed to the effective defect passivation and favorable interfacial dipole induced by DCl.

Since it has been proved that DCl-treated perovskite exhibits remarkable ferroelectric response, thus the effects of ferroelectric interface on device performance were additionally studied by polarizing the devices with an external poling bias (Supplementary Table 9). Figure 4c and Supplementary Table 10 display the J–V curves and corresponding PV parameters for the champion control and DCl-treated PSCs, respectively, with a positive poling bias of 0.5 V applied for 5 min under 1-sun illumination. The polarized DCl-treated device demonstrates a remarkable improvement in all PV parameters compared to the DCl-treated device without poling, obtaining a champion PCE of 22.6%, $V_{OC}$ of 1.238 V, FF of 83.6% and $J_{SC}$ of 21.8 mA cm$^{-2}$ with negligible hysteresis. In contrast, the PV parameters of the biased control device show nearly no changes with respect to the unbiased one, with $V_{OC}$ increase of only 2 mV, which is far below the $V_{OC}$ increase (12 mV) of the biased DCl device. In addition, the DCl device after applying positive poling (0.5 V for 5 min) shows holistically improved EQE spectrum as compared to that of the unpolarized device, without changing the band-gap of the perovskite absorber that typically contributes to $V_{OC}$ improvement (Supplementary Fig. 44). While the EQE spectrum of the positive poled control device exhibits nearly no variation as compared to the device without poling (Supplementary Fig. 45). This indicates that the enhancement of PV parameters for the DCl-treated device is inherently attributed to the presence of ferroelectric quasi-2D phase induced by DCl treatment[24], as schematically depicted in Fig. 4d. Before poling, the built-in electric field ($E_{bi}$) of PSCs is increased by the spontaneous polarization of the surface DCl-based ferroelectric quasi-2D perovskite (Supplementary Fig. 46), which facilitates carrier separation and extraction[51]. Nevertheless, the electric dipoles dominantly exhibit chaotic orientation, with only few electric dipoles aligning parallel to the $E_{bi}$ before applying external field polarization, resulting in weak spontaneous polarization intensity, thus incomplete charge separation. Upon polarization, the electric dipoles in the DCl-based ferroelectric quasi-2D perovskite align in contribution to the enhancement of the spontaneous polarization intensity, thereby an additional electric field ($E_{add}$) formation (Supplementary Fig. 47). This $E_{add}$ adds to the $E_{bi}$ responsible for photogenerated current, inducing more unseparated carriers to separate and transport to the hole/electron transport layers, thus improved FF and $J_{SC}$. Additionally, this $E_{add}$ enhances the electrical passivation through an additional field effect, increasing the barrier for across-interface recombination, thereby enhanced $V_{OC}$.

To delineate the ferroelectric enhancement mechanism, we performed DFT calculations on the surface work function (SWF) evolution of the perovskite (Supplementary Fig. 48). From Fig. 3c, d, the adsorption of randomly oriented cage-like cations leads to a dramatic upshift in the SWF from −1.95 eV to −0.25 eV, generating an additional dipole field that facilitates electron extraction while concurrently suppressing hole injection. Upon preliminary polarization, certain electric dipoles undergo ferroelectric switching, further lowering the SWF to −0.06 eV (Supplementary Fig. 49a). In this state, the surface electric dipoles evolve from a disordered to a directionally ordered configuration, forming a stronger interfacial dipole. When fully polarized, a uniformly oriented interfacial dipole layer forms, upshifting the SWF to 0.03 eV (Supplementary Fig. 49b), favoring more charge separation and extraction. Accordingly, these results indicate that ferroelectric-enhanced charge transport primarily originates from SWF reduction caused by the oriented align of spontaneously polarized cage-like cations under ferroelectric polarization. The polarization-induced SWF reduction establishes a reinforced interfacial dipole field that amplifies the $E_{bi}$ of the device, which promotes more effective charge carrier separation and transport to the electron transport layer, ultimately improving device performance. Light-dark cycle measurements further indicate that the ferroelectric effect offers a slight but transient boost in energy yield due to structural coupling within the 2D perovskite (Supplementary Fig. 50).

The excellent passivation and ferroelectric response of DCl was further verified in tandem solar cells (TSCs) by integrating the 1.68 eV wide-bandgap perovskite as top cell on tunnel oxide passivating contact (TOPCon) solar cell to construct the monolithic perovskite/silicon TSCs, with an aperture area of 1.0 cm$^2$. Supplementary Fig. 51 presents the schematic diagram of the tandem structure and the corresponding cross-section SEM image is given in Supplementary Fig. 52, exhibiting a conformal coating of perovskite on the silicon substrate with a sub-micron pyramid textured front-side. The best performing DCl-treated perovskite/silicon TSC displays the hysteresis-free J–V curves, with a PCE of 30.5% under forward scan ($V_{OC}$ = 1.897 V, FF = 79.1%, and $J_{SC}$ = 20.3 mA cm$^{-2}$) and a PCE of 30.6% under reverse scan ($V_{OC}$ = 1.896 V, FF = 79.4%, and $J_{SC}$ = 20.3 mA cm$^{-2}$) (Fig. 4e), which significantly outperforms the control's PCE of 28.1% under forward scan ($V_{OC}$ = 1.834 V, FF = 76.3%, and $J_{SC}$ = 20.1 mA cm$^{-2}$) and PCE of 28.4% under reverse scan ($V_{OC}$ = 1.828 V, FF = 77.2%, and $J_{SC}$ = 20.1 mA cm$^{-2}$). A steady state PCE of 30.5% is also achieved for the DCl-treated tandem cell after tracking at maximum-power-point at a fixed bias of 1.68 V for 600 s (Supplementary Fig. 53). Additionally, EQE spectra of the DCl-treated tandem cell demonstrates well-matched integrated current densities (20.28 mA cm$^{-2}$ for the perovskite top cell and 20.20 mA cm$^{-2}$ for the silicon bottom cell) to the $J_{SC}$ from the J–V scan (Supplementary Fig. 54). The best performing DCl-treated perovskite/silicon TSC was subsequently polarized for 5 min by applying an external positive bias of 1 V under 1-sun illumination. The J–V curves and corresponding PV parameters are given in Fig. 4f. A similar trend as in the DCl-based single junction devices is observed, the polarized DCl-treated tandem obtains an improved PCE of 30.9% with a $V_{OC}$ of 1.903 V, an FF of 79.7% and a $J_{SC}$ of 20.4 mA cm$^{-2}$ under forward J–V scan, and PCE of 31.1% with a $V_{OC}$ of 1.905 V, an FF of 80.1% and a $J_{SC}$ of 20.4 mA cm$^{-2}$ under reverse J–V scan, exhibiting negligible hysteresis and a certified steady-state PCE of 31.1% (Supplementary Fig. 55), which is benefited from the ferroelectric effect of DCl-mediated quasi-2D perovskite. Notably, this is the first application of ferroelectric interface physics in perovskite-based tandems, which opens a new field of possibilities for achieving efficient perovskite-based tandem photovoltaics. Prospectively, perovskites with a wider bandgap ( > 1.68 eV) can still be selected as top cells if they possess ferroelectricity, as the photocurrent mismatching induced by the perovskite can potentially be mitigated through ferroelectric polarization. Meanwhile, a greater power output will be enabled due to the more potential to generate high $V_{OC}$ from wider-bandgap perovskites. We further enlarged the tandem area to 4 cm$^2$, achieving a remarkable a PCE of 28.4% with a $V_{OC}$ of 1.909 V, an FF of 76.8% and a $J_{SC}$ of 19.4 mA cm$^{-2}$ under reverse scan (Fig. 4g). A stabilized PCE of 27.9% is also obtained (Supplementary Fig. 56). To the best

of our knowledge, this is among the highest PCEs reported for spin-coated perovskite/silicon TSCs with areas ≥ 4 cm² (Supplementary Table 11). The long-term operating stability of DCl-treated tandem cell was finally evaluated by subjecting the unencapsulated tandem devices to ambient conditions (relative humidity of 40–60% and temperature of 30 ± 5 °C) for maximum power point (MPP) tracking test under 1-sun xenon lamp illumination (AM1.5 G, 100 mW cm⁻²) without a UV filter according to the ISOS-L-1 protocol[53]. As presented in Supplementary Fig. 57, the control tandem device suffers from rapid degradation, retaining only ~43.4% of the initial PCE after 640 h of MPP tracking. In contrast, the DCl-treated tandem device maintains ~85.4% of its original PCE after 1020 h of MPP tracking, demonstrating stability comparable to the state-of-the-art perovskite/silicon TSCs reported under similar test conditions (Supplementary Table 12). The light-soaking stability was subsequently tracked under N₂ atmosphere, which shows that the DCl-treated tandem device retains 91.4% of the initial PCE after 792 h, significantly outperforming the control tandem (84.6% retention after 624 h) (Supplementary Fig. 58). The improved stability of DCl-treated tandems under light soaking is primarily attributed to the synergistic effects of DCl for comprehensively passivating charge trap states (Fig. 1g) and substantially suppressing ion migration (Supplementary Figs. 15, 16). Nevertheless, further research is urgently required for enhancing the long-term stability of perovskite/silicon TSCs, particularly in addressing intrinsic iodide oxidation and phase segregation[54].

## Discussion

In summary, we have developed a novel multifunctional cage-like diammonium chloride (DCl) molecule to modify the perovskite/C₆₀ interface. Benefiting from the Lewis acid/base groups and the dipole moment of this cage-like diammonium molecule, the surface defects are substantially suppressed and the energy level alignment is favorably improved, thereby reduced surface/interface recombination and enhanced electron selectivity. More importantly, the cage-like diammonium cation induces the formation of a ferroelectric phase-pure quasi-2D perovskite with vertical orientation, boosting carrier further apart and extraction by decreasing the surface work function. This cage-like DCl molecule ultimately leads to highly efficient perovskite solar cells (1.68 eV), achieving a maximum PCE of 22.6% for 0.1 cm² and a high PCE of 21.0% for 1.21 cm². Finally, we incorporate this DCl-mediated ferroelectric quasi-2D perovskite into monolithic perovskite/TOPCon tandem solar cells, eventually enabling a remarkable PCE of 31.1% (1.0 cm²) accompanied by an excellent long-term operational stability under ambient conditions (ISOS-L-1, $T_{85} > 1020$ h).

## Methods

### Materials

Cesium iodide (CsI, 99.999%), N, N-dimethylformamide (DMF, 99.8%), Dimethyl sulfoxide (DMSO, 99.8%), chlorobenzene (CB, 99.8%) and isopropanol (IPA, >99.5%) were ordered from Sigma-Aldrich. Methylammonium bromide (MABr), Lead(II) chloride (PbCl₂) and Formamidinium iodide (FAI) were acquired from GreatCell Solar Ltd. Indium-tin-oxide (ITO) glass substrates (7–9 Ohm per square) were purchased from Advanced Election Technology Co., Ltd. Lead(II) bromide (PbBr₂), lead(II) iodide (PbI₂)), Lead(II) Thiocyanate (Pb(SCN)₂), Potassium Iodide (KI) and [2-(3,6-dimethoxy-9H-carbazol-9-yl) ethyl] phosphonic acid (MeO-2PACz) were supplied by TCI. C₆₀ (99.5%) and bathocuproine (BCP, 99%) were procured from Lumtec Corp. Ethanol (anhydrous, ≥99.5%), 1,4-diazabicyclo[2.2.2] octane, 4,4-difluoropiperidine hydrochloride (2FPDCl) and hydrochloric acid (37%) were ordered from Aladdin. (4-(7H-dibenzo[c,g] carbazol-7yl)butyl)phosphonic acid (4PADCB) was purchased from Suzhou LiWei Tech Co., Ltd. PCl was synthesized according to our previous report[36].

### Synthesis of 1,4-diazabicyclo[2.2.2]octane chloride (DCl)

1,4-diazabicyclo[2.2.2]octane (23 mmol) was dissolved in ethanol (30 mL) and placed in an ice-water bath. 37% hydrochloric acid solution in water (23 mmol) was added dropwise and reacted for 30 min. The ethanol solvent was then rotary evaporated under reduced pressure, and the residual solid was washed with ethyl acetate for 5 times and subsequently dried for 12 h to obtain DCl powders (2.71 g, 79.3%). ¹H NMR (600 MHz, DMSO-$d_6$): δ 11.04 (s, 1H), 3.00 (s, 12H) (Supplementary Fig. 1).

### Single-junction perovskite solar cells fabrication

ITO substrates (2.5 × 2.5 cm²) were sequentially cleaned with detergent, deionized water, acetone, and anhydrous ethanol by sonication for 15 min, respectively. The substrates were then dried for 4 h at 80 °C in an oven, subjected to UV-ozone treatment for 15 min, and finally transferred into a N₂-filled glovebox before use.

In this work, self-assembled monolayer (SAM) MeO-2PACz was applied as hole selective contact. For MeO-2PACz-based devices, SAMs were dissolved in ethanol at a concentration of 0.3 mg mL⁻¹ and stirred for 30 min before use. The prepared SAM solution was then dropped onto the ITO substrate and left for 40 s before being spin-coated at 4000 rpm for 30 s, followed by annealing at 100 °C for 10 min. Afterward, the substrates were dynamically washed with ethanol (350 µl) twice using spin coating at 4000 rpm for 50 ss. For 4PADCB-based devices, 4PADCB was dissolved in ethanol (0.5 mg mL⁻¹), and the resultant solution was spin-coated at 3000 rpm for 30 s, followed by heating at 100 °C for 10 min without additional washing.

For perovskite precursor preparation, a 1.5 M perovskite precursor solution was prepared by mixing 0.075 mmol of CsI, 0.327 mmol of MABr, 0.354 mmol of PbBr₂, 1.098 mmol of FAI and 1.146 mmol of PbI₂ in 1 ml of DMF: DMSO (4: 1 v/v) mixed solvent, leading to a chemical formula of $Cs_{0.05}(FA_{0.77}MA_{0.23})_{0.95}Pb(I_{0.77}Br_{0.23})_3$, which corresponds to a 1.68 eV perovskite. For 1.6 M MA-free $Cs_{0.17}FA_{0.83}Pb(I_{0.83}Br_{0.17})_3$ perovskite precursor solution, PbCl₂ (19.5 mg), CsI (61.8 mg), PbBr₂ (131.0 mg), FAI (199.9 mg), PbI₂ (495.3 mg) were dissolved in 0.88 mL mixed solvent of DMF/DMSO (v/v, 3:1). For 1.2 M $FA_{0.8}Cs_{0.2}Pb(I_{0.6}Br_{0.4})_3$ perovskite precursor solution, 3.9 mg Pb(SCN)₂, 4.0 mg KI, 62.4 mg CsI, 264.2 mg PbBr₂,165.1 mg FAI, 221.3 mg PbI₂ were dissolved in 1 mL mixed solvent of DMF/DMSO (v/v, 4:1). Both precursor solutions were stirred for 2 h before use. For the fabrication of 1.68 eV perovskite films, the perovskite precursor was spin-coated on the SAM-coated substrates at 3500 r.p.m. for 40 s, with 200 µL of CB dripped onto the center of film at 15 s before the end of spin-coating. The substrates were immediately transferred to a hotplate and annealed at 100 °C for 20 min. For the devices with surface treatment, DCl materials were dissolved in isopropanol (IPA) at different concentrations (0.1 mg mL⁻¹, 0.2 mg mL⁻¹, 0.4 mg mL⁻¹ and 0.6 mg mL⁻¹) and dynamically spin coated onto the cooled-down perovskite films at 6000 r.p.m. for 35 s, followed by annealing at 100 °C for 10 min. For the fabrication of MA-free $Cs_{0.17}FA_{0.83}Pb(I_{0.83}Br_{0.17})_3$ perovskite films (1.68 eV), the perovskite precursor solution was spin-coated onto 4PADCB-coated substrates at 4000 rpm for 40 s. During spinning, 200 µL of CB anti-solvent was dripped onto the substrate at 10 s before the end of spin-coating, and the as-prepared films were immediately annealed at 100 °C for 20 min. For the $FA_{0.8}Cs_{0.2}Pb(I_{0.6}Br_{0.4})_3$ perovskite films (1.79 eV), the perovskite precursor solution was dripped onto 4PADCB-coated substrates and spin-coated at 4000 rpm for 40 s. 200 µL of CB was dripped after 30 s of spin-coating, followed by annealing at 100 °C for 15 min. Both film types were subsequently treated with 0.4 mg mL⁻¹ DCl via spin-coating at 6000 rpm for 35 s, followed by final annealing at 100 °C for 10 min.

Finally, 15 nm of C₆₀ (0.15 Å s⁻¹), 5 nm of BCP (1.0 Å s⁻¹) and 100 nm silver (Ag) electrode (1.0 Å s⁻¹) were thermally evaporated under high vacuum (<5 × 10⁻⁴ Pa) to obtain opaque perovskite solar cells. The device area was defined and characterized as 0.1 cm² by a metal shadow mask.

## Monolithic perovskite/silicon tandem solar cells fabrication

The bottom tunnel oxide passivating contact (TOPCon) solar cell was fabricated on $n$-type Czochralski (CZ) silicon wafers (160 μm), exhibiting a polished front-side and textured rear-side with micron-scale random pyramids. The submicron random pyramids of the front-side were obtained by immersing the wafers in the KOH solution (0.23 mol $L^{-1}$, 72 °C) for 7 min. Followed by a two-step cleaning process according to the RCA procedure: RCA1 ($NH_4/H_2O_2$) and RCA2 (HCl/$H_2O_2$). Subsequently, an ultrathin $SiO_X$ film ($\approx 1.6$ nm) followed by $n$-type/$p$-type a-Si:H ($\approx 30$ nm) was prepared on the front/rear side of the Si substrates using PECVD. Then, the samples were co-annealed at 940–1000 °C for 30 min in a furnace to crystallize the front/rear-sided a-Si:H films. The rear-sided $SiO_X$ thin layer was formed by a plasma-assisted $N_2O/H_2$ mixed gases oxidation method, while the front-sided $SiO_X$ thin layer was formed by the plasma-assisted $N_2O$ gas oxidation method. Notably, $SiH_4$, $B_2H_6$, $P_2H_6$, $H_2$, and $CH_4$ were selected as the reaction gases in the PECVD system. After the co-annealing procedure, an $AlO_X$:H capping layer with a thickness of about 15 nm was deposited by atomic layer deposition to enhance passivation quality. Afterward, the samples were annealed at 450 °C for 30 min in a nitrogen atmosphere. The surface oxide layer was subsequently removed by dipping the silicon wafers in HF solution. After that, an IZO ($\approx 20$ nm) film was sputter-deposited on the front side of the samples as a recombination layer. A forming gas annealing (FGA) at 280 °C for 20 min was implemented to recover the sputtering damage. Finally, a stack of 1 nm Cr/600 nm Ag metal electrode was thermally deposited on the rear-side through a shadow mask, resulting in a contact area of $1.1 \times 1.1$ cm², slightly larger than the active area of the completed tandem cell ($1.0 \times 1.0$ cm²). The wafers were finally cut down to $2.5 \times 2.5$ cm² substrates.

For perovskite top cells, the substrates were UV-ozone treated for 15 min. Then, the $NiO_X$ layer was RF-sputtered onto the pre-cleaned substrates at room temperature with a chamber pressure of 0.40 Pa, RF power of 90 W, argon flow rate of 20 sccm for 3 min without further annealing and UV-ozone treatment. Subsequently, the $NiO_X$ substrates were directly transferred into the $N_2$ glove box for the subsequential spin coating process. The deposition processes of hole selective layer MeO-2PACz, perovskite film, electron selective layer $C_{60}$ and hole blocking layer BCP were the same as that of the single-junction perovskite solar cells. After the deposition of BCP, 1 nm of ultrathin Ag was thermally deposited on the top of BCP to form a BCP: Ag buffer layer, after which an indium zinc oxide (IZO) transparent electrode (60 Ohm per square, $1.2 \times 1.2$ cm² defined by a shadow mask) was deposited by radio frequency (RF) magnetron sputtering at room temperature with a RF power of 80 W. For large-area perovskite/silicon TSCs with an active area of 4 cm², the perovskite precursor was deposited via spin-coating (3500 rpm, 40 s) onto MeO-2PACz-functionalized substrates. During the spinning process, 220 μL of CB was precisely dripped onto the film center 15 s prior to the spin-coating ended. The coated substrates were immediately transferred to a pre-heated hotplate for thermal annealing (100 °C, 20 min). After that, a 0.4 mg mL$^{-1}$ DCl solution (100 μL) was dynamically spin coated (6,000 rpm, 35 seconds) onto the cooled substrates, followed by annealing at 100 °C for 10 min. Subsequently, 13 nm of $C_{60}$ was thermally evaporated onto the substrates, after which 20 nm of $SnO_2$ was deposited via atomic layer deposition (MNT-PD100Oz-L3S1G3). The subsequent IZO process remained unchanged. An Ag finger with a thickness of 500 nm was thermally evaporated using a high-precision shadow mask with a deposition rate of 1.5 Å s$^{-1}$. The finger width is 50 μm (with 2.5 mm spacing in between subsequent fingers) and the busbar that surrounds the active area is 1.25 mm. To protect the samples from damaging by needle probes in subsequent device measurements, a low-temperature Ag paste was applied to a pad area (of about 5 mm²) by a handheld brush, and the samples were annealed at 85 °C for 5 min in a drying oven. Finally, a 100-nm-thick $MgF_X$ layer was full-area evaporated on

the samples by an e-beam system as an antireflection layer with a deposition rate of 1 Å s$^{-1}$. The thicknesses of the $C_{60}$, BCP, IZO and MgFx layers were calibrated by spectroscopic ellipsometry. The deposition rate and thickness of each experiment were monitored by quartz-crystal microbalance sensors. All thermal evaporation processes were performed on a thermal evaporation system (Beijing Technol Science Co., Ltd.).

## Material characterization

Liquid-state $^1$H nuclear magnetic resonance ($^1$H NMR) measurements were performed using an AVANCE NEO 600 (Bruker, Switzerland) with DMSO-d$_6$ as the locking solvent. Fourier Transform Infrared Spectroscopy (FTIR) was performed using a THEMOR-NICOLET 6700.

## Film characterization

The top-view and cross-sectional images of perovskite films and perovskite/silicon tandems were characterized using a Hitachi S-4800 FEG scanning electron microscope (SEM) at an accelerating voltage of 4.0 kV. General X-ray diffraction (XRD) and grazing incidence XRD patterns of perovskite films were measured using a D8 ADVANCE DAVINCI (Bruker, German) equipped with Cu K$\alpha_1$ irradiation of $\lambda = 1.5418$ Å. Grazing incident wide angle X-ray scattering (GIWAXS) measurements were performed with a Xeuss 3.0 UHR SAXS laboratory beamline using a Cu X-ray source (8.05 keV, 1.54 Å) and a Pilatus3R 300 K detector, with an incidence angle of 0.2°. UV–vis absorption spectra of perovskite films were obtained using a Perkin-Elmer Lambda 950 UV–vis spectrophotometer, with a measured wavelength range from 300 to 850 nm. Steady-state photoluminescence (SSPL) (Continuous mode) and corresponding PL images were obtained using a Micro-Raman spectrometer (Renishaw inVia Reflex). Time-resolved photoluminescence (TRPL) were conducted using a HORIBA FL3-111 spectrometer with 450 nm excitation. Carrier lifetime images were obtained using a confocal laser scanning fluorescence (CLSF) microscopy. X-ray photoelectron spectroscopy (XPS) and ultra-violet photoelectron spectroscopy (UPS) spectra were conducted using a multifunctional photoelectron spectrometer (Axis Ultra DLD, Kratos) under ultrahigh vacuum ($3.0 \times 10^{-8}$ Torr) with non-monochromatic He-I excitation (21.22 eV). The XPS spectra were calibrated using the binding energy of 284.8 eV for C 1s. Atomic force microscope (AFM), conductive atomic force microscopy (c-AFM) and Kelvin Probe Force Microscopy (KPFM) were conducted using a Dimension 3100 (Vecco, USA). Piezoresponse force microscopy (PFM) was measured by MFP-3D AFM (Asylum Research). For photoluminescence quantum yield (PLQY) measurements, two configurations were considered: (1) Glass/ITO/SAMs/perovskite half-stacks; (2) Glass/ITO/SAMs/perovskite/DCl/$C_{60}$ full-stack devices. Excitation for the PLQY measurements was performed with a 520 nm CW laser (Class 3b) through an optical fiber into an integrating sphere. The laser intensity was adjusted to a 1-sun equivalent intensity by illuminating $2.5 \times 2.5$ cm²-size perovskite films under short-circuit conditions and matching the current density to the $J_{SC}$ under the sun simulator (21.5 mA cm$^{-2}$ at 100 mW cm$^{-2}$).

## Device characterization

The current density versus voltage ($J$–$V$) curves, temperature-dependent conductivity and transient ionic current were measured using an Enlitech SS-F5-3A solar simulator based on the Ushio Xe short arc lamp 500 under 1-sun conditions (100 mW cm$^{-2}$, AM 1.5 G). The light source was calibrated using the National Renewable Energy Laboratories (NREL)−KG5 filtered silicon reference cell. The $J$–$V$ curves of the perovskite single-junction solar cells and perovskite/Si tandems were recorded in air. The voltage range for the forward and reverse scans of perovskite single-junction devices was maintained between $-0.1$ V → 1.25 V and 1.25 V → $-0.1$ V, respectively, and for perovskite/Si tandem solar cells, it was maintained between $-0.1$ V → 1.95 V and 1.95 V → $-0.1$ V, respectively. All $J$–$V$ measurements were conducted at

 

a scan rate of 100 mV/s with the delay time of 10 ms. The aperture areas of the small- and large-area shade masks are 0.1 and 1.0 cm$^2$, respectively, while the active area of the tandems for the large area was $1.1 \times 1.1$ cm$^2$. External quantum efficiency (EQE) measurements were conducted using a QE measurement system (QE-R, EnliTech), with light intensity at each wavelength calibrated using certified reference cells (Si and Ge reference solar cells) for the 300–1200 nm wavelength range. For perovskite/Si tandem solar cells, the perovskite top cell was measured while saturating the silicon bottom cell with continuous biased light from a white light source equipped with a long-pass (> 850 nm) filter. To maintain short-circuit conditions, a bias voltage of -0.2 V was applied during the measurement. Conversely, the silicon bottom cell was measured while saturating the perovskite top cell with continuous biased light from a white light source equipped with a low-pass (< 550 nm) filter. All EQE characterizations were performed at ambient air and without encapsulation. Mott-Schottky plots and electrical impedance spectroscopy (EIS) measurements were measured on an electrochemical workstation (CH1660E). Transient photocurrent (TPC) and transient photovoltage (TPV) measurements were conducted on a TranPVC W900 system (Oriental Spectra Technology Co., Ltd., Guangzhou) using a 517 nm pulsed laser excitation source (NanoQ series). Trap density of states (tDOS) curves were obtained using a 1240 A Impedance Analyzer.

The long-term operational stability of the monolithic perovskite/silicon tandem solar cells was assessed following ISOS-L-1 procedures. The measured devices were unencapsulated and no cooling was implemented under 1-sun constant illumination (xenon lamp without UV filter). Measurements were carried out under ambient conditions at temperatures ranging from 25 to 35 °C, with relative humidity of 40–60% (measured using a calibrated humidity tracker). Light-dark cycle testing of unencapsulated devices was conducted in N$_2$ at MPP using a commercial multichannel stability test system (Wuhan 91PVKSolar), with 4 h illumination followed by 4 h recovery in the dark.

## Calculation of the quasi-fermi level splitting (QFLS) based on PLQY

To investigate the open-circuit voltage potential for every individual stack e.g., surface treatments and depositing C$_{60}$ layer, the internal QFLS values of 1.68-eV bandgap perovskite films on glass/ITO/MeO-2PACz substrates were calculated by PLQY measurements:[5]

$$\text{QFLS} = \text{QFLS}_{\text{rad}} + K_B T \ln(\text{PLQY}) = K_B T \ln\left(\text{PLQY}\frac{J_G}{J_{0,\text{rad}}}\right) \quad (1)$$

Here, QFLS is the difference between the electron and hole quasi-Fermi levels in the perovskite layer, $K_B$ is the Boltzmann constant, and $T$ is the temperature (300 K). $J_G$ is the generated current density under illumination, in this case, approximated to the short-circuit current density $J_{SC}$ (21.5 mA cm$^{-2}$) of devices. $J_{0,\text{rad}}$ is the dark radiative recombination saturation current density. According to the detailed balance at open-circuit conditions, the $J_{0,\text{rad}}$ can be calculated by the following equations:

$$J_{0,\text{rad}} = e \int_0^\infty \text{EQE}_{\text{PV}}(E) \varnothing_{\text{BB}}(E) dE \quad (2)$$

$$\varnothing_{\text{BB}}(E) = \frac{2\pi E^2}{h^3 c^2} \frac{1}{\exp\left(\frac{E}{k_B T}\right) - 1} \quad (3)$$

External quantum efficiency (EQE) of the p-i-n PSCs and the emitted spectral photon flux are calculated when the device is in equilibrium with the black-body radiation ($T = 300$ K). $E$ is the photo energy, $h$ is the Plank constant, and $c$ is the light speed in vacuum. Based on the above equations, the $J_{0,\text{rad}}$ was calculated similarly as $4.153 \times 10^{-23}$ A m$^{-2}$ for all

systems independent of the surface treatment. Subsequently, combined with the PLQY value of the samples, the QFLS of the samples can be obtained.

## DFT calculations

**Calculations of electrostatic potential and electric dipole moment.** The geometries of 1,4-diazabicyclo[2.2.2]octane and cage-like diammonium cation were optimized within the density functional theory (DFT) framework using the M06-2X functional, coupled with the DFT-D3 dispersion correction method and the 6–31 G(d) basis set[55,56]. For the theoretical calculations of molecular dipole moments, accuracy is highly sensitive to the choice of basis set due to the strong dependence of molecular polarization properties on diffuse functions. Herein, the def2-TZVPD basis set was employed for dipole moment calculations[57]. All DFT computations were conducted with the Gaussian 16 program. Electrostatic surface potentials and extrema points were calculated using the Multiwfn program[58], while visualization of the electrostatic surface potentials was rendered using the Visual Molecular Dynamics (VMD) program[59].

**Calculations of adsorption energy.** The adsorption structures were investigated using periodic plane-wave DFT calculations performed with the Vienna Ab Initio Simulation Package (VASP). The Perdew-Burke-Ernzerhof (PBE) exchange-correlation functional was employed with a kinetic energy cutoff of 450 eV[60]. The Brillouin zone integration was sampled using a $2 \times 2 \times 1$ Monkhorst-Pack k-point mesh. During structural optimization, the convergence criterion for atomic forces was set to 0.02 eV/Å. To preserve the solid-phase characteristics, the cell parameters were kept fixed while allowing only the atomic positions to relax.

The adsorption energy ($E_{\text{ad}}$) was calculated using the following equation:

$$E_{\text{ad}} = E_{\text{adsorbate/slab}} - E_{\text{adsorbate}} - E_{\text{slab}} \quad (4)$$

where $E_{\text{adsorbate/slab}}$, $E_{\text{adsorbate}}$ and $E_{\text{slab}}$ represent the adsorbate-adsorbed kaolinite substrate, the adsorbate, and the total energies of the kaolinite substrate, respectively. The 3D iso-surfaces and 1D planar-averaged results of charge density differences (CDD) between the substrate and the adsorbate were obtained by subtracting the electron densities of the isolated substrate and adsorbate from that of the combined system. These calculations were performed using VASPKIT[61], and the charge density difference visualizations were rendered using the VESTA program[62].

## Calculations of defect formation energy, partial density of state, surface work function and charge density difference

All first-principles calculations based on DFT were performed using VASP. The electron-ion interactions were described by projector augmented-wave (PAW) pseudopotentials, and exchange-correlation effects were treated using the PBE functional within the generalized gradient approximation (GGA)[60,63,64]. The slab model was constructed from a cubic-phase primitive cell with a $2 \times 2$ surface supercell comprising eight atomic layers. The bottom four layers were fixed, while the remaining atoms were fully relaxed. A vacuum layer of at least 20 Å was introduced along the z-direction to prevent periodic interactions. Calculations employed a plane-wave energy cutoff of 500 eV and a k-point mesh of $1 \times 1 \times 1$ for slab models ($2 \times 2 \times 1$ for DOS calculations). Structural optimization was performed with a force convergence threshold of 0.05 eV/Å and an energy convergence criterion of $1 \times 10^{-5}$ eV. Van der Waals interactions were accounted for using the Grimme D3-BJ correction[65]. The defect formation energy is calculated as:

$$E_F = E_{\text{slab + defect}} - E_{\text{slab}} \pm \sum n_i \mu_i \quad (5)$$

Where $E_{slab+defect}$ is the energy of the slab with defects, $E_{slab}$ is the energy of defect-free slab, $n_i$ is the number of atoms added or removed, $\mu_i$ is the chemical potential of the corresponding element.

## Reporting summary

Further information on research design is available in the Nature Portfolio Reporting Summary linked to this article.

## Data availability

Source data are provided with this paper. All the data supporting the findings of this study are available within this article and its Supplementary Information. Any additional information can be obtained from corresponding authors upon request. Source data are provided with this paper.

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

## Acknowledgements

This work was financially supported by the National Key Research and Development Program of China (2024YFB3817304), National Natural Science Foundation of China (Grant No. 62204245, U23A200098), Baima Lake Laboratory Joint Funds of the Zhejiang Provincial Natural Science Foundation of China (Grant No. LBMHD24E020002), Key Research and Development Program of Zhejiang Province (Grant No. 2022C01215, 2024C01092), China Postdoctoral Science Foundation (Grant No. 2023M743620, 2024T170960), Key Research and Development Program of Ningbo (Grant No. 2023Z151, 2024QL037), and Zhejiang Provincial Natural Science Foundation of China (Grant No. LY24F040003).

## Author contributions

X.L., Z.Y., X.Y., and J.Y. performed conceptualization. X.L. performed data curation. X.L., Z.Y., and X.Y. performed methodology. X.L., Z.Y., X.Y., and J.Y. performed an investigation. X.L., Z.Y., L.L., J.W., H.M., Z.H., Y.Y., Y.S., M.Z., X.G., Y.Z., and X.Y. performed visualization. Z.Y., X.Y., and J.Y. acquired funding. Z.Y., X.Y., and J.Y. performed project administration. Z.Y., X.Y., and J.Y. performed supervision. X.L. and X.Y. wrote the original draft. X.L., Z.Y., X.Y., and J.Y. wrote, reviewed, and edited the final manuscript.

## Competing interests

The authors declare no competing interests.
