## [Transparent Peer Review file · Nature Communications]

Minimizing Interfacial Energy Losses via Multifunctional Cage-like Diammonium Molecules for Efficient Perovskite/Silicon Tandem Solar Cells

Corresponding Author: Professor Jichun Ye

Version 0:

Reviewer comments:

Reviewer #1

(Remarks to the Author)

In this paper, the authors developed a cage-like diammonium chloride (DACl) molecule and applied it to wide-bandgap (WBG) perovskites and tandem solar cells. However, the primary focus of the research is on WBG perovskites. Although the work demonstrates an efficiency improvement, its fundamental issue lies in how to address the phase separation problem in WBG perovskites, particularly the Br/I mixed-phase separation issue. The DACl passivation agent seems to lack practical utility, especially for surface modification. Therefore, I do not see the highlights of this work and do not recommend its publication in Nature Communications. The following issues require special attention:

1. What role does "ferroelectricity" play in the ferroelectric pure-phase quasi-2D perovskite? How does it critically contribute to the improvement of device performance? In previous studies, it has been challenging to associate efficient perovskite solar cells with ferroelectric properties.
2. Regarding the top-view SEM images of the (left) control and (right) DACl-treated perovskite films, no obvious changes can be observed from the SEM images. If the XRD results can reveal the presence of low-dimensional crystal phases, then SEM should exhibit more noticeable differences. Moreover, low-dimensional structures typically do not significantly enhance device performance.
3. In fact, the authors have merely attempted a new organic amine salt, DACl, in this work. Similar studies are too common and lack novelty. At present, an efficiency of 22% for WBG perovskite solar cells is not remarkable, even though the authors have applied it to TOPCon tandem cells.
4. In Figure 2a, what is the significance of the changes in the Pb 4f XPS spectra? These seem unrelated to the paper's core content. DACl is merely a passivating amine salt.

Reviewer #2

(Remarks to the Author)

This manuscript introduces effective passivation molecules at the interface between the perovskite and the electron transport layer (ETL), which are distinct from widely used alternatives. The molecules are ferroelectric in nature and are proposed to achieve multifunctionality, including passivation and charge transport. The IV data is properly presented, including both forward and reverse scans as well as steady-state performance. Statistical distributions of PV parameters have been shown. The tandem solar cells show improved light stability after DCI treatment, but they still fall short compared to the state-of-the-art, which demonstrates stability exceeding 1,000 hours. A more detailed analysis of the factors contributing to this discrepancy would be useful, and strategies to further enhance stability should be discussed. A longer stability measurement will be necessary to better understand the stability behavior of the as-prepared cells.

The reported efficiency of 21.9% (before poling) is relatively moderate compared to recent publications, where efficiencies for wide-bandgap perovskite solar cells have surpassed 23%: eg. *Advanced Materials*, 2024, 36(9): p. 2307701; *Nature Communications*, 2024, 15(1): p. 1-11; *Advanced Materials*, 2024: p. 2407681. The Voc was around 1.22V, considerably lower than the state-of-the-art values of >1.25V indicating a suboptimal interface that could impact both the performance and the stability. Could you elaborate on potential reasons for this lower Voc and how the interface might be further optimized?

Although performance improves to over 22% after poling, a key question remains: how much will this enhancement

contribute to energy yield over time? The ferroelectric phenomenon in the DCI-treated samples raises important concerns: Is the effect temporary or permanent? Given the light-dark cycling of solar cells, would the cells require additional biasing after being in the dark to achieve optimal ferroelectric performance again?

The manuscript claims that DCI induces a p-type to n-type surface energetic transformation. However, the fundamental mechanism for this transformation remains unclear. Please elaborate on how DCI facilitates such behaviour.

Similarly, the claim that molecular dipoles formed by the cage-like diammonium cation improve surface energy band alignment by altering the surface work function requires further explanation. Please provide detailed mechanisms for this process.

The authors state that the cage-like diammonium cation contains both electron donor (R_3N) and electron acceptor (R_3NH^+) functional groups, which passivate both negative and positive surface trap states. Please specify which types of positive and negative trap states are passivated by DCI.

The manuscript notes that the DCI-treated film exhibits higher RMS surface roughness (from 13.6 nm to 18.6 nm) due to additional island-like substances and columnar-like protrusions, which are said to optimize interfacial contact with the ETL. Considering the ETL is ultrathin (~ 10 nm), please clarify how a rougher surface facilitates improved interfacial contact.

In Figure S9, the PL spectra show significant drops at certain wavelengths for both the control and target samples. Please clarify the potential causes for these drops in the photoluminescence intensity.

In Figure S3, the XRD pattern shows a noticeable background signal. Could you please remove this background signal to improve the clarity and focus of the data?

Figure S2: Please add clear labels to indicate what each symbol represents.

Reviewer #3

(Remarks to the Author)

In this work, the authors synthesize a multifunctional cage-like diammonium chloride molecule (DCI) and incorporate it into the perovskite/C60 interface to reduce energy losses. Furthermore, the authors claimed that the DCI molecule promotes the formation of a pure-phase quasi-2D perovskite with favorable crystal orientation and ferroelectric properties, further improving charge separation and extraction. As a result, DCI-treated perovskite solar cells achieve high efficiencies (22.6% for 0.1 cm^2 and 21.0% for 1.21 cm^2), and when incorporated into perovskite/silicon tandem cells, a record PCE of 31.1% is achieved. Though the manuscript is interesting but it currently shows many scientific flaws that restrict its publication in Nature Communication.

1. As depicted in Figures 1a and 1b. It seems that DCI materials didn't coordinate well with the perovskite layer as there is no change in surface crystallinity and morphology. The post-treated material can be seen deposited on top of perovskite crystals. The increased surface roughness also confirms this. So, the authors should carefully reinvestigate this and should provide its diffusion properties.
2. In Figure 1f,g, a decreased conductivity was observed due to the increased resistance because of the low-dimensional phase. According to the line profile shown in Figure 1g, the current value of about $1.5 \mu\text{m}$ distance shows a slightly high value. However, Figure 1h has shown a similar current value at all distances. Therefore, an explanation of the line profile is necessary.
3. The DCI material didn't passivate well, so according to c-AFM data, why does the author think DCI treatment reduces ion migration? The authors should explain the reason and the mechanism.
4. There are certain surface defects for both with and without DCI-treated films, therefore, a more thorough investigation related to charge carrier dynamics should be performed especially using impedance, TPC, and SCLC techniques.
5. In Fig. 2c and Fig. S8, the authors showed the interaction of DCI with PbI_2 and FA ions, respectively. However, since perovskite is a crystal composed of various cations and anions, it should be explained the interaction between DCI to perovskite crystals. For example, it is necessary to explain what is the main interaction in the perovskite structure either Pb-Cl bonding or hydrogen bonding with FA ions.
6. The authors should provide the mechanism of passivation for untreated and DCI-treated films.
7. The authors performed PFM analysis to evaluate the ferroelectric properties, which is shown in Figure 3g. It was confirmed that the phase difference was 180 degrees due to the incorporation of DCI. However, to understand ferroelectric properties, further explanation is needed as to why there is a difference in intensity for each phase. For example, whether the intensity difference for each phase is a meaningful change or a negligible difference.
8. Figure S18, What factors are responsible for different EQE spectra, with and without DCI-treated films?
9. The surface morphology for with and without DCI-treated films is almost similar, though DCI-treated films are rougher, so why there is a steady decrease in PCE for the control device and very minor loss for the DCI-treated device? Also, the DCI-treated perovskite films didn't show any adequate passivation.
10. The authors explain that charge recombination occurs at the interface between perovskite and charge transport layers, and it is particularly emphasized between perovskite/C60. Therefore, to understand carrier dynamics at the perovskite/C60, the mechanism of charge recombination at the perovskite/C60 interface needs to be explained.

Reviewer #4

(Remarks to the Author)

This study demonstrates a multifunctional cage-like diammonium chloride molecule to reduce the surface defects and suppress non-radiative recombination. By using this molecule, the PCE of wide-bandgap perovskite solar cells reaches 22.6% and that of 2T perovskite/silicon tandem solar cells reaches 31.1%. The results are interesting but there are some issues that need to be addressed.

1. The standard deviation of the devices PCE measurements for each condition should be provided along with the number of samples measured so that the statistical reproducibility of the device can be more clearly demonstrated.
2. Instead of only the stability results under ISOS-L-1, more stability tests under other conditions, such as damp heat or light soaking, should be provided, and comparisons with results reported in other literature should also be included.
3. It would be useful to provide a control experiment in which a caged diammonium salt molecule is not used, but rather a similar molecule with only Lewis acid/base function is tested to highlight the uniqueness of the method.
4. Certain data trends (e.g., increase work-function after DCI treatment) should be cross-validated by other measurements such as density functional theory (DFT) calculations.
5. The discussion of how the ferroelectric properties of quasi-two-dimensional packet crystals can enhance carrier extraction is interesting, but a firmer theoretical foundation would be helpful. In addition, a quantitative estimate of the built-in electric field generated by the ferroelectric effect would have strengthened the argument.
6. Are there potential limitations if cage-like diammonium molecules are used in other perovskite compositions?
7. The impact of the present findings on the field of 2T perovskite/silicon tandem cells could be more clearly clarified, especially in terms of scalability and commercial viability.

Version 1:

Reviewer comments:

Reviewer #1

(Remarks to the Author)

accept as it is

Reviewer #2

(Remarks to the Author)

The revised manuscript and accompanying response letter address some of the technical questions raised and provide clarification on a few points from the original submission. However, the revision does not fundamentally enhance the quality or impact of the work. Several critical issues remain unresolved, including a lack of substantial novelty, only moderate device performance, unclear scalability, unclear relationship between the ferroelectricity and device stability, and insufficient evidence to support the claimed role of ferroelectricity in enhancing device performance, etc. These limitations prevent the manuscript from making a significant contribution to the field and, in my opinion, it does not meet the high standards required for publication in Nature Communications.

The authors state: "To the best of our knowledge, the impressive PCE of 28.4% is among the highest values for reported large-area ($\geq 4 \text{ cm}^2$) perovskite/silicon tandem solar cells based on spin-coating method... These results demonstrate that DACI is not only effective at the lab scale but also holds promise for scalable and broadly applicable device engineering." However, a 4 cm^2 device is still far too small to draw any meaningful conclusions about the scalability of the proposed passivation strategy, particularly since the molecule is applied using a spin-coating method. This approach is not compatible with industrial-scale processing, where cell areas are well above 100 cm^2 . Therefore, how this passivation method could be transferred to large-area, scalable fabrication remains an open and significant challenge.

The manuscript also lacks a clear explanation of the role of ferroelectricity in improving device stability. The authors claim enhanced stability due to ferroelectric effects, but do not present sufficient experimental or theoretical evidence to support this mechanism.

Regarding device performance, the reported efficiencies are relatively modest compared to recent state-of-the-art results. Moreover, Table R2 does not present a complete or representative overview of the literature. Several recent studies have demonstrated efficiencies exceeding 23% for wide-bandgap perovskite sub-cells with similar bandgaps. Below are just a few relevant examples (not exhaustive):

1. Li, S., et al., A Generic Strategy to Stabilize Wide Bandgap Perovskites for Efficient Tandem Solar Cells, *Advanced Materials*, 2024, 36(9), 2307701.
2. Fang, Z., et al., Surface reconstruction of wide-bandgap perovskites enables efficient perovskite/silicon tandem solar cells, *Nature Communications*, 2024, 15(1), 1–11.
3. Wang, Z., et al., Regulation of Wide Bandgap Perovskite by Rubidium Thiocyanate for Efficient Silicon/Perovskite Tandem Solar Cells, *Advanced Materials*, 2024, 2407681.

The authors suggest that the relatively low efficiency in their study may be due to suboptimal hole transport layers or additive selection. If this is the case, the claim should be substantiated by experiments using improved HTLs or optimized perovskite compositions.

It is important to note that neither the single-junction nor the tandem solar cell efficiencies reported in the manuscript have been independently certified. Certification is a critical benchmark for validating photovoltaic performance, particularly in high-impact journals.

Reviewer #3

(Remarks to the Author)

The authors have carefully addressed the reviewer's comments point by point and have made adequate revisions accordingly. I believe this version of the manuscript is suitable for publication.

Reviewer #4

(Remarks to the Author)

The authors have responded to all comments properly and added more detail data and relevant explanation. Now it is ready for publication in this journal.

Version 2:

Reviewer comments:

Reviewer #3

(Remarks to the Author)

The author addressed to the reviewer's comment point by point.

They have revised the manuscript regarding the comments and added the extra discussion with experimental data. Thus I believe this work is considerable to be published in this journal

Reviewer #4

(Remarks to the Author)

The authors have addressed the reviewers' comments as far as possible within the constraints of their current experimental conditions. The concern regarding scalability remains, but this issue is considered beyond the main scope of the present work. Regarding the certification issue, the authors have clarified the matter and submitted the device for independent third-party certification. Overall, the current content is deemed sufficient at this stage.

Reviewer #1:

In this paper, the authors developed a cage-like diammonium chloride (DACl) molecule and applied it to wide-bandgap (WBG) perovskites and tandem solar cells. However, the primary focus of the research is on WBG perovskites. Although the work demonstrates an efficiency improvement, its fundamental issue lies in how to address the phase separation problem in WBG perovskites, particularly the Br/I mixed-phase separation issue. The DACl passivation agent seems to lack practical utility, especially for surface modification. Therefore, I do not see the highlights of this work and do not recommend its publication in Nature Communications. The following issues require special attention:

Response: We thank the reviewer for highlighting this important point. Indeed, phase separation, especially Br/I demixing, remains a critical challenge for wide-bandgap (WBG) perovskites [Adv. Mater. 2025, 2418500. 10.1002/adma.202418500] and leads to poor long-term stability and substantial V_{oc} loss in perovskite/silicon TSCs. In response, researchers have developed a range of strategies including component engineering [Nat. Sustain. 2025. 10.1038/s41893-025-01529-5], additive engineering [Science 2024, 385, 533. 10.1126/science.adp1621; Nat. Commun. 2025, 16, 40. 10.1038/s41467-024-55377-6; Adv. Mater. 2025, 2416513. 10.1002/adma.202416513; Nat. Photonics 2022, 16, 588. 10.1038/s41566-022-01033-8], interface engineering [Nat Commun 2024, 15, 10554. 10.1038/s41467-024-54925-4], dimension engineering [Nat Commun 2024, 15, 7024. 10.1038/s41467-024-51345-2] to mitigate the phase separation issues, showing promising progress. In our study, although the primary focus was on improving efficiency via DCI treatment, we note that this interface treatment also contributed to enhanced phase stability, as suggested by the suppressed halide ion migration observed in the temperature-dependent conductivity and transient ionic current tests ([Figure](#)

R1).

Figure R1 (a) Temperature-dependent conductivity of control and DCI-treated perovskite films. (b) Transient ionic current (TIC) for extracting mobile ion concentration within the perovskite films of the control and DCI-treated devices.

Generally, the phase separation of perovskites under illumination is closely linked to defect-induced ion migration [Adv. Mater. 2023, 35, 2302552. 10.1002/adma.202302552]. Study has shown that the irreversible deterioration of perovskite film during light soaking primarily originates from interstitial iodide and PbI_2 defects [Science 2020, 369, 96. 10.1126/science.aba1628; Science 2021, 373, 902. 10.1126/science.abi6323; Joule 2022, 6, 816. 10.1016/j.joule.2022.03.005]. Passivating those defects by DCI can effectively mitigate the iodine generation and migration in perovskite films, and thereby suppressing phase separation and degradation of perovskites during aging [Nat. Photonics 2022, 16, 588. 10.1038/s41566-022-01033-8; Nat Commun 2024, 15, 7024. 10.1038/s41467-024-51345-2]. In our work, ToF-SIMS results demonstrate that both interfacial and bulk defects are effectively passivated (Figure R2 and R3), benefiting from the synergistic effect enabled by the coordination interaction of multifunctional cage-like diammonium cations and the formation of Pb-Cl ionic bonding. On one hand, PbI_2 defects filling the grain boundaries are eliminated, while deep level defects with low formation energy (i.e. I_{Pb} and Pb_{I} antisite defects) at the perovskite surface are calculated to be passivated by cage-like diammonium cations (Figure R4). On the other hand,

the formation of strong Pb-Cl ionic bonds throughout the perovskite film enhances the formation energy of interstitial iodide defect and effectively suppresses phase separation [Nano-Micro Lett. 2025, 17, 141. 10.1007/s40820-024-01631-x; Energy Environ. Sci. 2023, 16, 2080. 10.1039/D2EE04087E].

Although, additionally experimental and computational results reveal that DCI-induced interface engineering can contribute to enhanced phase stability. However, we agree that the issue of phase separation is not fully resolved and remains a key limitation for WBG perovskites. Further investigation into composition and additive engineering, and crystallization control strategies will be carried out in future work to address this challenge more thoroughly.

Figure R2 ToF-SIMS 2D depth profiles of DCI-treated perovskite film.

Figure R3 ToF-SIMS 3D images of (a) cage-like diammonium cations ($C_6H_{13}N_2^+$) and (b) chloride anions (Cl^-).

Figure R4 (a,b) The partial density of states (PDOSs) of (a) the defective perovskite (I_{Pb} antisite) and (b) passivated perovskite by DCI. (c, d) The partial density of states (PDOSs) of (c) the defective perovskite (Pb_i antisite) and (d) passivated perovskite by DCI.

Furthermore, we appreciate the reviewer's comment regarding the practical utility of DCI. In addition to the significant improvements in device efficiency and stability demonstrated in our main results, we have further validated the practical relevance of DCI by demonstrating its compatibility with large-area device fabrication and its applicability across perovskites with different compositions and bandgaps. Specifically, as shown in **Figure R5**, DACI treatment was successfully applied to large-area perovskite/silicon tandem devices with a 4 cm^2 active area, maintaining high performance and uniformity. We obtained an impressive efficiency of 28.4% (26.7%), with a V_{oc} of 1.909 (1.898) V, an FF of 76.8% (72.5%) and a J_{sc} of 19.4 (19.4) mA cm^{-2} under reverse (forward) scan (**Figure R5a**). Additionally, a stabilized PCE of 27.9% during 600 s with a constant bias of 1.58 V was achieved (**Figure R5b**). Notably, our large-area tandem photovoltaic devices were solution-processed

exclusively through spin-coating deposition. To the best of our knowledge, the impressive PCE of 28.4% is among the highest values for reported large-area ($\geq 4 \text{ cm}^2$) perovskite/silicon tandem solar cells based on spin-coating method (Table R1).

Figure R5 (a) J–V curves and (b) steady-state PCE with a constant bias of 1.58 V for the large-area monolithic perovskite/silicon tandem solar cells with an active area of 4 cm^2 .

Table R1 Summary of large-area monolithic perovskite/silicon tandems based on the spin-coating method reported in literatures. HJ: homojunction; SHJ: silicon heterojunction; TOPCon: tunnel oxide passivated contact.

PCE [%]	J_{sc} [mA cm^{-2}]	V_{oc} [V]	FF [%]	silicon	type	Area [cm^2]	ref.
20.5	16.1	1.68	78.0	HJ	n-i-p	4	1
21.8	16.2	1.74	78.0	HJ	n-i-p	16	2
22.6	17.5	1.72	75.0	SHJ	p-i-n	57.4	3
23.1	16.5	1.78	81.0	HJ	n-i-p	4	4
27.1	19.1	1.88	75.5	SHJ	p-i-n	3.8	5
17.3	14.1	1.78	67.0	TOPerc	n-i-p	25	6
28.3	19.1	1.84	80.2	TOPCon	p-i-n	4	7
27.1	/	/	/	SHJ	n-i-p	24	8
26.4	19.0	1.91	72.6	SHJ	p-i-n	8.9	9
28.4	19.4	1.91	76.8	TOPCon	p-i-n	4	This

Furthermore, we extended the treatment to both MA-free 1.68 eV and 1.79 eV perovskite absorbers, achieving consistent improvements in PCE (Figure R6), highlighting the bandgap universality of this passivation strategy. For 1.68 eV MA-free perovskite solar cells, the best-performing device achieves a PCE of 22.9%, with an V_{OC} of 1.25 V and an FF of 83.7% (Figure R6a). For 1.79 eV wide-bandgap perovskite solar cells, the corresponding devices yield a champion PCE of 19.4% with an V_{OC} of 1.30 V (Figure R6b). Both types of devices show negligible hysteresis. These findings demonstrate the generality of DCI in terms of various perovskite compositions and highlight the potential of DCI in all perovskite tandem solar cells. The corresponding integrated J_{SC} values of the champion perovskite solar cells based on 1.68 eV MA-free and 1.79 eV perovskites are 21.3 mA cm^{-2} and 17.5 mA cm^{-2} , respectively (Figure R7), which matches well with the J_{SC} values extracted from the J-V curves.

These results demonstrate that DACI is not only effective at the lab scale but also holds promise for scalable and broadly applicable device engineering, supporting its practical utility in future perovskite photovoltaic technologies.

Figure R6 J-V curves of perovskite solar cells fabricated with (a) 1.68 eV MA-free and (b) 1.79 eV wide-bandgap perovskites.

Figure R7 EQE curves and the corresponding integrated photocurrent densities for 1.68 eV MA-free and 1.79 eV wide-bandgap perovskite solar cells.

Our response:

The measurements of temperature-dependent conductivity and transient ionic current had been added in **Fig. S15** and **Fig. S16**, respectively.

ToF-SIMS 2D depth profiles of DCI-treated perovskite film had been added to **Fig. S12**.

ToF-SIMS 3D images of DCI-treated perovskite film had been added to **Fig. 1g**.

The partial density of states (PDOSs) of the defective perovskite (P_b antisite) and passivated perovskite by DCI had been added to **Fig. 2h** and **Fig. 2i**, respectively.

The partial density of states (PDOSs) of the defective perovskite (P_b antisite) and passivated perovskite by DCI had been added to **Fig. S25**.

J–V curves and steady-state PCE of the 4 cm² large-area monolithic perovskite/silicon tandem solar cells had been added to **Fig. 4g**.

Summary of large-area monolithic perovskite/silicon TSCs based on the spin-coating method reported in literature had been added to **Table S11**.

J–V curves and EQE spectra and the corresponding integrated J_{sc}s for 1.68 eV MA-free and 1.79 eV wide-bandgap perovskite solar cells had been added to **Fig. S40**.

The corresponding discussions of the above measurements had been highlighted in the main text.

#1-1. What role does “ferroelectricity” play in the ferroelectric pure-phase quasi-2D perovskite? How does it critically contribute to the improvement of device performance? In previous studies, it has been challenging to associate efficient perovskite solar cells with ferroelectric properties.

Our reply: We are very grateful to the reviewer’s professional comments. Ferroelectricity in pure-phase quasi-2D perovskites primarily facilitates enhanced charge transport, driven by the polarization effects of molecular ferroelectrics. As is well known, the intrinsically insulating nature of organic spacer layers leads to inferior charge transport, representing a major bottleneck for the photovoltaic performance of 2D perovskites [Chem. Rev. 2021, 121, 2230. 10.1021/acs.chemrev.0c01006]. As illustrated in Figure R8, conventional 2D perovskites consist of alternating arrangement of inorganic frameworks and non-ferroelectric organic spacer layers which typically direct crystal growth parallel to the underlying 3D perovskite structure (Figure R8b). The dramatically contrasting dielectric constant between inorganic layers and organic spacer layers results in a multiple quantum well electronic structure, thereby severely limiting out-of-plane conductivity across the inorganic slabs (Figure R8c). This restricted charge transport ultimately hinders efficient carrier extraction from the 3D perovskite to the C₆₀ layer. Therefore, developing effective strategies to enhance out-of-plane charge transport in 2D perovskites is of critical importance.

Recently, 2D molecular ferroelectrics have emerged as a highly promising approach to address this issue [Nat. Commun. 2025, 16, 835. 10.1038/s41467-025-56182-5; Nat. Commun. 2024, 15, 8753. 10.1038/s41467-024-53121-8; Nat. Commun. 2025, 16, 1042. 10.1038/s41467-025-56409-5; Small Methods 2024, 8, e2400425. 10.1002/smt.202400425]. The advantage of these materials lies in their simultaneous possession of both 2D structural features and intrinsic ferroelectricity (Figure R8d). This ferroelectricity arises from an

order-disorder transition of organic cations (Figure R8e), which generates a built-in depolarization field within the ferroelectric spacer layers. This internal field effectively promotes out-of-plane charge transport across the inorganic layers, thereby significantly improving device performance (Figure R8f). Overall, compared with the non-ferroelectric 2D perovskites, ferroelectricity has a vital influence on film optoelectronic properties and directly enhances the out-of-plane charge transport.

Figure R8 Schematic diagram of the poor out-of-plane charge transport in 2D perovskites and enhanced out-of-plane charge transport in the 2D molecular ferroelectrics [Natl. Sci. Rev. 2023, 10, nwad061. 10.1093/nsr/nwad061].

However, the critical contribution of ferroelectricity to device performance enhancement remains largely unexplored. To date, most studies on the ferroelectric effect have primarily focused on comprehensive characterizations aimed at confirming the existence of ferroelectricity and its influence on promoting charge transfer and suppressing charge recombination in perovskite solar cells. However, the fundamental mechanism by which the ferroelectric effect enhances charge carrier separation and extraction has rarely been elucidated.

In our work, we systematically elucidated, for the first time, the intrinsic correlation between the ferroelectric effect and enhanced device performance

through density functional theory (DFT) calculations. Our DFT results demonstrate that the enhancement of charge transport by the ferroelectric effect originates from a ferroelectric polarization-induced decrease in the perovskite surface work function (SWF). As illustrated in **Figure R9**, we calculated the work functions of perovskite surfaces with adsorbed cage-like cations possessing various dipole orientations. Before the adsorption of cage-like cations, the control perovskite exhibits a high SWF of -1.95 eV (**Figure R9a**). Once the cage-like cations adsorb onto the perovskite surface in a disordered orientation (**Figure R9b**), the SWF dramatically decreases to -0.25 eV. This reduction is mainly attributed to strong interfacial dipole generated by the spontaneous polarization of cage-like cations, which facilitates electron extraction while effectively blocking hole injection. This theoretical result aligns well with the surface potential changes observed in UPS and KPFM measurements. Upon preliminary polarization, certain electric dipoles undergo ferroelectric switching, leading to a further reduction of the SWF to -0.06 eV (**Figure R9c**). This enhancement facilitates more efficient charge separation and transport. Under these conditions, the surface electric dipoles tend to evolve from a disordered state to a directionally ordered configuration, thereby generating a stronger interfacial dipole. When fully polarized, a completely oriented interfacial dipole layer forms, further upshifting the surface work function (SWF) to 0.03 eV (**Figure R9d**), which is more conducive to charge separation and extraction. Overall, the underlying mechanism by which the ferroelectric effect enhances charge carrier separation and extraction is the decrease of SWF caused by the oriented dipole alignment of cage-like cations under ferroelectric polarization. The decrease of SWF is accompanied by an enhancement of the built-in field, which promotes the separation and transport of more charge carriers to the electron transport layer, thereby improving device performance.

Figure R9 DFT calculations of the surface work functions of perovskite without and with adsorption of cage-like cations. (a) Bare perovskite, (b) Cage-like cation adsorption in a disordered orientation, (c) Preliminary polarization, (d) Complete polarization.

Our response:

On page 18, line 11, “In a word, E_{bi} enhancement driven by the external polarization of the DCI-mediated ferroelectric quasi-2D perovskite is key to boosting device performance.” had been deleted.

DFT calculated surface work functions of perovskite without and with adsorption of cage-like cations and the charge density difference had been added to Fig. 3c, Fig. 3d and Fig. 3e, respectively.

DFT calculated surface work functions of perovskite during the ferroelectric polarization process had been added to Fig. S49.

On page 18, line 12, “To delineate the ferroelectric enhancement mechanism, we performed DFT calculations on the surface work function (SWF) evolution of the perovskite (Fig. S48). From Fig. 3c and d, the adsorption of randomly oriented cage-like cations leads to a dramatic upshift in the SWF from -1.95 eV to -0.25 eV, generating an additional dipole field that facilitates

electron extraction while concurrently suppressing hole injection. Upon preliminary polarization, certain electric dipoles undergo ferroelectric switching, further lowering the SWF to -0.06 eV (Fig. S49a). In this state, the surface electric dipoles evolve from a disordered to a directionally ordered configuration, forming a stronger interfacial dipole. When fully polarized, a uniformly oriented interfacial dipole layer forms, upshifting the SWF to 0.03 eV (Fig. S49b), favoring more charge separation and extraction. Accordingly, these results indicate that ferroelectric-enhanced charge transport primarily originates from SWF reduction caused by the oriented align of spontaneously polarized cage-like cations under ferroelectric polarization. The polarization-induced SWF reduction establishes a reinforced interfacial dipole field that amplifies the E_{bi} of the device, which promotes more effective charge carrier separation and transport to the electron transport layer, ultimately improving device performance.” had been added.

#1-2. Regarding the top-view SEM images of the (left) control and (right) DACI-treated perovskite films, no obvious changes can be observed from the SEM images. If the XRD results can reveal the presence of low dimensional crystal phases, then SEM should exhibit more noticeable differences. Moreover, low-dimensional structures typically do not significantly enhance device performance.

Our reply: We sincerely appreciate the reviewer’s insightful feedback and comments. We have re-investigated the surface morphology of control and DCI-treated perovskite films. As depicted in **Figure R10**, a number of discrete sheet-like PbI_2 phase filling the grain boundaries of the 3D perovskite in the control film is observed. Upon DCI modification, the perovskite surface undergoes reconstruction, forming numerous visible small grains within the perovskite grain surfaces, accompanied by the disappearance of residual PbI_2 . The above

phenomenon indicates that DCI can coordinate well with the perovskite surface and the residual PbI_2 .

Figure R10 Top-view SEM images of the (left) control and (right) DCI-treated perovskite films.

Notably, these small grains on the DCI-treated perovskite surface (Fig. 1a and b) are ascribed to the formation of a 2D phase rather than the post-treated DCI material itself, as evidenced in Figure R11. The peak position at 10.2° of the synthesized single-crystal 2D perovskite precisely matches that observed in GIXRD (Fig. S4).

Figure R11 XRD patterns of DCI powder and synthesized single-crystal 2D perovskite.

Furthermore, we summarized the relevant work on representative low-dimensional phase used in inverted perovskite solar cells in recent years (Table R2). As the reviewer pointed out, certain low-dimensional perovskites on the surface of 3D perovskite, such as pure BA_2PbI_4 , TEA_2PbI_4 , PDADI perovskite, with $n=1$ and out-of-plane orientations do not significantly improve the performance of inverted perovskite solar cells. This is primarily due to the

unfavorable energy level alignment and excessive thickness, leading to reduced transport of the free charge carriers. Alternatively, increasing the n value of 2D perovskites to ≥ 2 to form quasi-2D structures can help reduce carrier transport barriers to some extent, thus enhancing device performance [Nat. Photonics 2022, 16, 352. 10.1038/s41566-022-00985-1]. However, similar to pure 2D perovskites, when the thickness of quasi-2D structures becomes excessive, device performance can degrade significantly [Science 2022, 377, 1425. 10.1126/science.abq7652]. In addition, the thickness of quasi-2D perovskites is generally uncontrollable. Therefore, developing 2D perovskite with in-plane orientation is crucial for inverted perovskite solar cells [Nat. Energy 2024, 9, 779. 10.1038/s41560-024-01529-3]. Indeed, studies have shown that quasi-2D perovskite with in-plane orientation can significantly enhance the device performance regardless of the thickness [Energy Environ. Sci. 2022, 15, 3369. 10.1039/d2ee00759b; Energy Environ. Sci. 2025, 18, 1354. 10.1039/D4EE04136D], which holds for our designed DCI. However, in most cases, quasi-2D perovskites featuring in-plane orientation do not form spontaneously and require assistance from additional deposition processes [Energy Environ. Sci. 2022, 15, 3369. 10.1039/d2ee00759b], which undoubtedly increases the complexity of the process. Overall, for inverted perovskite solar cells, the enhancement of device performance by 2D perovskites typically requires the following conditions: high n values, in-plane orientation, phase-pure. In our work, DCI induces the formation of phase-pure quasi-2D perovskite ($n=3$) with spontaneous in-plane orientation (Fig. 1), accompanied by enhanced defect passivation (Fig. 2), optimized energy level alignment, and improved ferroelectric properties (Fig. 3), ultimately leading to significantly improved device performance and stability (Fig. 4), highlighting its uniqueness.

Table R2 Summary of surface 2D engineering for the state-of-the-art inverted

perovskite solar cells reported in literature.

Surface passivators	2D phase	Orientation	PCE (Control/Target)	Ref.
BA	BA ₂ PbI ₄	Out-of-plane	21.45/21.52	10
BLSCI	BLS ₂ MAPb ₂ I ₇	In-plane	21.6/25.8	11
PDAI ₂	-	Out-of-plane	24.1/26.1	12
CzCl-EAI	no	-	22.4/25.2	13
OAI	OA ₂ MAPb ₂ I ₇	Out-of-plane	23.9/24.8	14
3S-PEA	3F- PEA ₂ MA ₂ Pb ₃ I ₁₀	Out-of-plane	21.1/23.3	15
PEAI	PEA ₂ PbI ₄	Out-of-plane	18.6/21.1	16
PEAI	PEA ₂ ZnI ₄	Out-of-plane	19.3/22.2	17
PEAI	PEA ₂ MA ₂ Pb ₃ I ₇	Out-of-plane	16.8/19.5	18
GBAC	GBA ₂ MAPb ₂ I ₇	Out-of-plane	11.8/15.0	19
MAP	MAP ₂ PbI ₄	Out-of-plane	23.5/26.1	20
PEABr	PEA ₂ MAPb ₂ I ₇	Out-of-plane	19.2/24.7	21
OAI	OA ₂ MAPb ₂ I ₇	Out-of-plane	22.7/25.4	22
TEACl	TEA ₂ PbI ₄	Out-of-plane	17.4/19.3	23
F- PEACl/CF ₃ - PAI	A ₂ PbI ₄	Out-of-plane	17.7/19.6	24
PDADI	Mixed 2D	Out-of-plane	17.1/18.7	25
BABr	BA ₂ MAPb ₂ I ₄	In-plane	18.5/21.4	26
OLAI	OLA ₂ MAPb ₂ I ₇	Out-of-plane	22.3/24.3	27
DCI	D ₂ MA ₂ Pb ₃ I ₁₀	In-plane	19.0/22.6	This work

Our response:

The SEM images of the control and DCI-treated perovskite films had been

re-investigated in **Fig. 1a**.

On **page 4, line 19**, “On one hand, some columnar-like substances, featuring out-of-plane epitaxial growth, are discretely distributed at grain boundaries (GBs) of the 3D perovskite. On the other hand, many island-like clusters with a relatively low surface coverage are randomly distributed within the grain interiors (GIs) of the 3D perovskite.” had been revised to “**The perovskite surface undergoes reconstruction, forming numerous small grains uniformly distributed at the perovskite grain surfaces, accompanied by the disappearance of residual PbI_2** ”.

On **page 4, line 21**, “These results indicate that cage-like diammonium cation, similar to fluorinated plane-type ammonium cations, can also react with PbI_2 and 3D perovskite phase to generate discrete new substances, spontaneously assembling on the top surface of the 3D perovskite.” had been revised to “**These results indicate that cage-like diammonium cation, similar to fluorinated plane-type ammonium cations, can also react with PbI_2 and 3D perovskite phase to generate new substances, spontaneously assembling on the top surface of the 3D perovskite^{24,27}**”.

The XRD patterns of DCI powder and synthesized single-crystal 2D perovskite had been added to **Fig. S7**.

On **page 5, line 9**, “Careful observation, reveals a clearly new peak at 10.2° in the DCI-treated film, indicating the formation of pure-phase quasi-2D perovskite with $n=3$ (n is the number of semiconducting $[\text{PbX}_6]^{4-}$ octahedron slabs within the cage-like diammonium spacers)²⁸.” had been revised to “Careful observation, reveals a clearly new peak at 10.2° in the DCI-treated film, indicating the formation of pure-phase quasi-2D perovskite with $n=3$ (n is the number of semiconducting $[\text{PbX}_6]^{4-}$ octahedron slabs within the cage-like diammonium spacers) (**Fig. S7**)²⁹”.

#1-3. In fact, the authors have merely attempted a new organic amine salt, DACl, in this work. Similar studies are too common and lack novelty. At present, an efficiency of 22% for WBG perovskite solar cells is not remarkable, even though the authors have applied it to TOPCon tandem cells.

Our reply: Thank you for your valuable comments. We appreciate the reviewer's attention to this matter and regret that the key points of novelty in our work may not have been clearly conveyed. To clarify, we have outlined the main novel aspects of this study below.

The key innovation of this study lies in the development of a novel multifunctional cage-like diammonium chloride molecule 1,4-diazabicyclo[2.2.2]octane chloride (DCI). Its key contributions are as follows:

First of all, the unique cage-like structure of the cation induces the spontaneous in-plane growth of phase-pure quasi-2D perovskites. As shown in **Table R2**, most reported conventional 2D perovskites exhibit spontaneous out-of-plane orientation, which is generally detrimental to charge carrier transport in inverted perovskite solar cells unless the films are of ultrathin thickness. However, the DCI amine salt, owing to its unique cage-like structure, promotes the spontaneous in-plane growth of phase-pure quasi-2D perovskites, thereby lowering the energy barrier for charge transport. These results suggest that the spatial dimensionality of amine salts also plays a key role in determining the orientation of 2D perovskite—an aspect that has rarely been emphasized in previous studies.

To highlight the uniqueness of our designed DCI, we additionally selected other two similar molecules—4,4-difluoropiperidine hydrochloride (2FPDCI) [Small Methods 2024, 8, e2400425. 10.1002/smt.202400425; Nat. Commun. 2024, 15, 8753. 10.1038/s41467-024-53121-8] and piperazinium hydrochloride (PCI) [Nano-Micro Lett. 2025, 17, 141. 10.1007/s40820-024-01631-x] containing the same Cl anion as DCI for comparison. Similar to DCI, both

2FPDCI and PCI incorporate Lewis acid and Lewis base functional groups within their cationic moieties. We systematically investigated the comparative effects of these three materials on crystallographic orientation of perovskite films using grazing-incidence wide-angle X-ray scattering (GIWAXS), as depicted in **Figure R12**. Careful observation, reveals a new signal at $q_z = 0.49 \text{ \AA}^{-1}$ in only the 2FPDCI-treated sample and a new signal at $q_{xy} = 0.72 \text{ \AA}^{-1}$ in only the DCI-treated sample. Evidently, the crystallographic orientation of the 2D phase at the DCI-treated perovskite surface is entirely different from that at the 2FPDCI-treated perovskite surface, which validates our hypothesis that the orientation of 2D perovskite is also influenced by the spatial dimensionality of the cations.

Figure R12 2D GIWAXS patterns of the control, PCI-treated, 2FPDCI-treated and DCI-treated perovskite films with a grazing incidence angle of 0.2° .

Second, the novelty of this work lies in the development of a novel interface dipole strategy. The acid-base neutralization reaction is employed to unilaterally protonate the 1,4-diazabicyclo[2.2.2]octane molecules, resulting in the formation of DCI molecules with permanent polarization (**Figure R13**). Meanwhile, the spontaneously polarized cage-like diammonium cation can

assemble directionally on the 3D perovskite surface, generating a substantial interfacial dipole that effectively adjusts energy band bending at the perovskite/C₆₀ interface and reduces band misalignment (Figure R9).

Figure R13 Chemical structures, electrostatic potentials (ESPs) and corresponding electric dipole moments of both 1,4-diazabicyclo[2.2.2]octane and cage-like diammonium cation.

Third, the novelty of this study resides in the identification of the ferroelectric effect of DCI and its successful integration into perovskite single-junction and tandem solar cells. The integration of molecular ferroelectric materials into perovskite solar cells necessitates the fulfillment of several stringent criteria, including high compatibility with the perovskite, high Curie temperature, and room temperature ferroelectric behavior. In accordance with these requirements, we performed a systematic screening of a wide range of candidate ferroelectric molecules and identified DCI as the only material that meets all the necessary conditions. Furthermore, in contrast to conventional organic ferroelectric molecules, DCI features distinct multiaxial ferroelectric characteristic [J. Am. Chem. Soc. 2017, 139, 13903. 10.1021/jacs.7b07715], offering the potential for higher polarization and thereby enhancing charge carrier separation and extraction.

Notably, an extensive literature review revealed a lack of reported cases

involving the use of molecular ferroelectric materials for interface engineering in wide bandgap perovskites and tandems. For the first time, ferroelectric DCI molecules were intercalated at the interface between wide-bandgap perovskite and C₆₀ to mitigate interfacial energy losses. This strategy was further implemented in perovskite/TOPCon tandem solar cells, ultimately achieving a remarkable power conversion efficiency (31.1%) and enhanced stability (T_{85.4} = 1020 h). The introduction of ferroelectric interface physics offers a promising pathway toward the development of efficient and stable perovskite-based tandem photovoltaics.

Finally, we systematically revealed the underlying correlation between the ferroelectric effect and the surface work function of the perovskite through density functional theory (DFT) calculations. At present, most studies on the ferroelectric effect have focused on thorough characterizations to firmly demonstrate the existence of ferroelectricity and its role in promoting charge transfer and suppressing charge recombination in perovskite solar cells. However, the underlying mechanism by which the ferroelectric effect enhances charge carrier separation and extraction has rarely been elucidated. In this context, we comprehensively explored the correlation between the ferroelectric effect and the surface work function of perovskites using density functional theory (DFT) calculations.

Collectively, these findings highlight the uniqueness of the DCI molecule and underscore the novelties of this work. We hope that these revisions and clarifications have effectively highlighted the novelty of this work, making it readily apparent to both the reviewers and readers. Once again, we sincerely appreciate your insightful comments and suggestions. We hope that the revised manuscript will be found suitable for publication in your esteemed journal.

In addition, as noted by the reviewer, we also acknowledge that the

efficiency of our reported device is relatively moderate when compared to recent publications (Table R3). After thoroughly reviewing recent work on wide-bandgap (WBG) perovskites (~1.68 eV), we attribute the efficiency gap to two main factors.

The first factor is the choice of hole transport layers (HTLs). Compared to MeO-2PACz, 2PACz, and conventional PTAA HTLs, Me-4PACz, 4PADCB HTLs, or their combinations with NiO_x exhibit higher V_{OC} and PCE. This improvement is primarily attributed to enhanced hole extraction and suppression of non-radiative recombination [Science 2020, 370, 1300. 10.1126/science.abd4016; Nature 2023, 618, 80. 10.1038/s41586-023-05992-y]. In the referenced studies, replacing MeO-2PACz with 4PADCB as the HTL increased the V_{OC} of WBG perovskite solar cells from 1.22 to 1.30 V, surpassing even the 1.29 V achieved with NiO_x/Me-4PACz due to superior hole extraction and energy level alignment [Nature 2023, 618, 80. 10.1038/s41586-023-05992-y].

The second factor is the use of additives. As summarized in Table R3, the incorporation of additives enhances the V_{OC} of WBG perovskite solar cells by effectively passivating bulk defects and improving crystallization. In contrast, our work did not involve the use of any additives.

In our work, we focus on defect passivation, interfacial dipole modulation, and the ferroelectric effect at perovskite/C₆₀ interface. Despite employing a MeO-2PACz hole transport layer, our DCI yields a WBG cell with a V_{OC} of 1.238 V and a fill factor of 83.6%, resulting in a final efficiency of 22.6% surpassing that of control devices. We highlight the significance of defects passivation in enhancing device stability, as well as the roles of interfacial dipole and ferroelectric effect in facilitating charge extraction.

Finally, we also explored the potential of DCI-based WBG perovskite devices by employing alternative SAMs such as 4PADCB as the HTL. This resulted in an improved PCE of 22.9%, with a V_{OC} of 1.255 V, an FF of 84.7%

and a J_{SC} of 21.6 mA cm^{-2} (Figure R14), comparable to recently reported advanced 1.68 eV perovskite solar cells. This indicates that the choice of an appropriate hole transport layer is crucial for optimizing device performance. Notably, these data are not included in current manuscript, since it is beyond our original scope of this work. Considering the better hole transport capability of 4PADCB, we will apply it in our future study on WBG perovskite and tandem solar cells.

Table R3 Summary of performance of advanced single-junction WBG perovskite solar cells.

E_g (eV)	HTLs	Additives	Post-treatment	V_{oc} (V)	PCE (%)	Ref.
1.67	Me- 4PACz	Pb(SCN) ₂	EDA _{l2}	1.24	23.1	28
1.66	4PADCB	RbSCN	EDA _{l2} +PEAI	1.30	24.5	29
1.68	PTAA	MACI+PbCl ₂	GUACI+OAmI	1.23	21.58	30
1.68	NiOx/Me- 4PACz	-	Al ₂ O ₃ nano- polishing+PI	1.29	24.5	31
1.67	Ph-2PACz	Pb(SCN) ₂ +PEAI	LiF	1.26	21.3	32
1.68	NiO	ICPS	LiF	1.27	22.0	33
1.68	Me- 4PACz	-	PI	1.27	20.6	34
1.68	NiO/MeO- 4PADBC	MACI+ODAI ₂	CF ₃ -PEAI	1.25	22.7	35
1.68	MeO- 2PACz	-	PEABr+5ATT	1.25	22.6	36
1.68	NiO/Me- 4PACZ	MACI+AESCI	-	1.28	22.8	37
1.68	NiO/MeO-	TAACI	CF ₃ -PEAI	1.28	23.0	38

	2PACz					
1.68	MeO- 4PACz	MDACI	CF ₃ -PEAI+LiF	1.27	23.2	39
1.68	Bz- PhpPACz	-	PDI+PEAI	1.27	22.5	40
1.68	MeO- 2PACz	-	DCI	1.23	22.6	This work

Figure R14 J-V curves of the DCI-treated PSC based on 4PADCB HTL.

Our response:

To emphasize the key innovation of this study, several revisions have been made in the main manuscript:

On page 1, line 16, “Herein, for the first time, a multifunctional cage-like diammonium chloride molecule, consisting of Lewis acid/base groups with strong molecular polarity, is developed to reduce the surface defects and optimize the surface band alignment, suppressing non-radiative recombination and enhancing charge transport” had been revised to “Herein, for the first time, a multifunctional cage-like diammonium chloride molecule, featuring Lewis acid/base groups and strong molecular polarity, is designed to reduce film defects and modulate the interfacial dipole, thereby suppressing non-radiative recombination and optimizing surface band alignment.”.

On page 1, line 20, “More importantly, the cage-like diammonium cation can induce the formation of a ferroelectric pure-phase quasi-2D perovskite

oriented vertically, discretely assembled on the 3D perovskite surface, facilitating carrier further apart and extraction.” had been revised to “More importantly, the unique cage-like cation can induce the formation of a phase-pure quasi-2D perovskite with spontaneous in-plane orientation and exhibits a pronounced ferroelectric effect, facilitating carrier further apart and extraction by upshifting the surface work function.”.

On page 3, line 16, “Moreover, DCI helps to induce a p-type to n-type surface energetics transformation and an interface positive dipole,” had been revised to “Moreover, the spontaneously polarized cage-like cation induces a p-to-n type surface energetics transformation and forms a positive interfacial dipole.”.

On page 3, line 20, “More importantly, the spontaneously assembled pure-phase quasi-2D perovskite by such cage-like diammonium cation is not only featured with intrinsic in-plane crystal orientation for favorable interfacial charge transport, but also endowed with additional ferroelectricity for enhancing carrier further separation and extraction.” had been revised to “More importantly, the in-plane oriented, phase-pure quasi-2D perovskite induced by the cage-like diammonium cation is endowed with pronounced ferroelectricity, promoting carrier further separation and extraction by upshifting the surface work function.”.

On page 22, line 3, “More importantly, the cage-like diammonium cation is found to discretely assemble atop 3D perovskite to form a ferroelectric pure-phase quasi-2D perovskite oriented vertically, boosting carrier further apart and extraction.” had been revised to “More importantly, the cage-like diammonium cation induces the formation of a ferroelectric phase-pure quasi-2D perovskite with vertical orientation, boosting carrier further apart and extraction by decreasing the surface work function.”.

#1-4. In Figure 2a, what is the significance of the changes in the Pb 4f XPS spectra? These seem unrelated to the paper's core content. DACI is merely a passivating amine salt.

Our reply: We thank reviewer for the critical comment on the Pb 4f XPS spectra. As the reviewer pointed out, both Pb 4f and I 3d XPS spectra indicate the formation of stronger Pb–Cl ionic bonds, despite the coordination interaction between the cage-like cation and uncoordinated Pb^{2+} , which is also consistent with our DFT calculations. As shown in Figure R15, we conducted density functional theory (DFT) calculations to determine the binding energies for DCI adsorption. The DCI is categorized into two functional groups (R_3N and R_3NHCl), and four potential adsorption structures on the PbI_2 -terminated and formamidinium iodide (FAI)-terminated (100) perovskite surfaces are given. The corresponding adsorption energies (E_{ad}) for different configurations are calculated and presented in Figure R15. Regarding the FAI termination of perovskite (Figure R15a), the E_{ad} values of $\text{R}_3\text{N}/\text{FAI}$ and $\text{R}_3\text{NHCl}/\text{FAI}$ are -0.42 and -0.50 eV, respectively, indicating a comparable binding energy. On the FAI termination, both R_3N and R_3NHCl functional groups tend to bind with perovskite surface via weak hydrogen bonding, e.g. $\text{R}_3\text{N}\cdots\text{FA}^+$, $\text{Cl}\cdots\text{FA}^+$ and $\text{R}_3\text{NH}^+\cdots\text{I}$. In contrast, for the PbI_2 termination (Figure R15b), the E_{ad} values of $\text{R}_3\text{NHCl}/\text{PbI}_2$ and $\text{R}_3\text{N}/\text{PbI}_2$ are -0.90 eV and -0.78 eV, respectively, which is significantly lower than that of $\text{R}_3\text{NHCl}/\text{FAI}$ and $\text{R}_3\text{N}/\text{FAI}$, indicating a stronger binding configuration on the PbI_2 termination, and thus the primary interaction of DCI with PbI_2 . Additionally, it is noteworthy that, for $\text{R}_3\text{NHCl}/\text{PbI}_2$ adsorption configuration, the strong interaction only occurs between Cl and Pb via Pb–Cl ionic bonding. As for R_3NH^+ group, no interaction with Pb but weak hydrogen bonding with I occurs. Accordingly, it is indicated that DCI is energetically more favorable to interact with perovskite via strong coordination bonding of R_3N group with PbI_2 and Pb–Cl ionic bonding. From the ToF-SIMS results with cage-

like diammonium cations concentrating at the perovskite surface and small chloride anions migrating downward to accumulate at the buried interface, we can infer that, with this anchoring effect, oriented molecular dipoles directed from perovskite to C₆₀ can be established by cage-like diammonium cations at the perovskite/C₆₀ interface, thereby creating a dipole layer over the perovskite surface, which is essential to regulate interfacial energy level difference, thereby promoting device performance. While for chloride anions, the formation of strong Pb-Cl ionic bonding is conducive to passivating the bulk and buried defects, thereby enhancing the device performance and stability.

Figure R15 (a) DFT modeling of the interaction of the DCI with FAI termination. (b) DFT modeling of the interaction of the DCI with PbI₂ termination.

Our response:

To emphasize the core content of this work, the following adjustments were made:

DFT modeling of the interaction of the DCI with FAI-terminated and PbI₂-terminated (100) perovskite surface had been added to **Fig. S17**.

¹H NMR spectra of DCI and DCI+PbI₂ had been revised to **Fig. S18**.

Pb 4f and I 3f XPS spectra had been revised to **Fig. S21**.

The corresponding reference [Energy Environ. Sci. 2023, 16, 2080. 10.1039/D2EE04087E], [Nat. Commun. 2025, 16, 835. 10.1038/s41467-025-56182-5] had been cited.

On page 8, line 9, “To elucidate the interaction between perovskite and DCI, high-resolution X-ray photoelectron spectroscopy (XPS) characterization was performed. In the C 1s XPS core (Fig. S6), the binding energies centered at ~284.8, 286.6, and 288.4 eV are assigned to C-C, N-C-C-N, and N-C=N, respectively. Turning to the N 1s XPS core (Fig. S7), an obvious increase in the higher binding energy peak at 402.7 eV is detected, likely corresponding to R₃-N bond. The obvious N-C-C-N and R₃-N characteristic peaks for DCI-treated film collectively indicates the existence of cage-like diammonium cation on the perovskite surface and its dominant interaction with the perovskite. Further, we also notice that with the incorporation of DCI, the characteristic peaks in the Pb 4f XPS spectra shift toward the higher binding energy (from 143.37 and 138.51 eV for 4f_{5/2} and 4f_{7/2} to 143.55 and 138.66 eV, respectively), which may originate from the bond formation between the uncoordinated Pb²⁺ in perovskite or PbI₂ and DCI (Fig. 2a and Table S1). Similar upward shift of the corresponding characteristic peaks in the I 4d XPS spectra is also observed (Fig. 2b and Table S2), in which I 3d_{3/2} (630.71 eV) and I 3d_{5/2} (619.26 eV) for the control film slightly shift toward higher binding energy position of 630.77 and 619.31 eV, respectively, after DCI post-treatment, which could be ascribed to the incorporation of Cl³⁸.

Subsequently, liquid-state ¹H nuclear magnetic resonance (¹H NMR) was carried out to further explore the interaction mechanism of DCI with perovskite, as illustrated in Fig. 2c. The chemical shift at 11.04 ppm corresponds to the R₃-NH⁺ in DCI. Upon addition of PbI₂ to DCI, ¹H signal of the R₃-NH⁺ group shows an obviously upfield shift from 11.04 ppm to 9.45 ppm, which is attributed to the shielding effect induced by anion exchange from Pb-I bond to Pb-Cl bond and the direct ionic binding of Cl⁻ with the undercoordinated Pb²⁺³⁹. Similar upfield shift trends of ¹H signal for the R₃-NH⁺ group in DCI was also found when mixing DCI with FAI, as depicted in Fig. S8. ¹H signal of the R₃-NH⁺ group in DCI moves

from 11.04 ppm to 9.68 ppm. In addition, the resonance signal (8.79 ppm) of protonated ammonium in FAI splits into two at 8.965 ppm and 8.629 ppm, respectively. These chemical shifts indicate the strong interaction of DCI with FAI by the formation of hydrogen bonds⁴⁰. Overall, the aforementioned results from c-AFM, XPS and NMR measurements collectively confirm the strong chemical interaction of DCI with the perovskite surface via its multi-functional passivation groups, which is expected to reduce the surface trap states, suppressing nonradiative recombination, thus improving photovoltaic (PV) performance.” had been revised to “To elucidate the interaction between perovskite and DCI, we conducted density functional theory (DFT) calculations to determine the binding energies of DCI adsorption. For simplicity, the DCI was categorized into two functional groups: R₃N and R₃NHCl. Four potential adsorption configurations on the PbI₂-terminated and FAI-terminated (100) perovskite surfaces were considered, with the corresponding adsorption energies (E_{ad}) presented in Fig. S17. Regarding the FAI termination (Fig. S17a), the E_{ad} values of R₃N/FAI and R₃NHCl/FAI are -0.42 and -0.50 eV, respectively, indicating comparable and relatively weak binding. On this surface, the interactions are mainly through weak hydrogen bonding (e.g., R₃N⋯FA⁺, Cl⋯FA⁺ and R₃NH⁺⋯I), as supported by liquid-state ¹H NMR spectra (Fig. S18a). In contrast, for the PbI₂ termination (Fig. S18b), R₃NHCl/PbI₂ and R₃N/PbI₂ exhibit much stronger binding energies of -0.90 eV and -0.78 eV, respectively, indicating a clear preference for adsorption on PbI₂ termination and highlighting the primary interaction of DCI with PbI₂. Notably, in the R₃NHCl/PbI₂ configuration, strong Pb-Cl ionic bonding is observed, whereas the R₃NH⁺ group exhibits only weak hydrogen bonds with I, without direct interaction with Pb. These results reveal that DCI predominantly interacts with perovskite via R₃N-Pb coordination and Pb-Cl ionic bonding, as further confirmed by high-resolution X-ray photoelectron spectroscopy (Fig. S19-S21

and Table S1,2). Furthermore, ToF-SIMS results reveal that cage-like diammonium cations accumulate at the perovskite surface, while chloride anions migrate downward (Fig. 1g). In this case, the spontaneously polarized cage-like cations directionally assemble at the perovskite/C₆₀ interface, forming a robust interfacial dipole layer that modulates band bending and alleviates energy level misalignment⁴⁰. Concurrently, strong Pb-Cl ionic bonding effectively passivates bulk and interfacial defects, contributing to improved performance and stability⁴¹.”.

In the part of “**DFT calculations**”, “**Calculations of adsorption energy**” The adsorption structures were investigated using periodic plane-wave DFT calculations performed with the Vienna Ab Initio Simulation Package (VASP). The Perdew-Burke-Ernzerhof (PBE) exchange-correlation functional was employed with a kinetic energy cutoff of 450 eV. The Brillouin zone integration was sampled using a 2×2×1 Monkhorst-Pack k-point mesh. During structural optimization, the convergence criterion for atomic forces was set to 0.02 eV/Å. To preserve the solid-phase characteristics, the cell parameters were kept fixed while allowing only the atomic positions to relax.

The adsorption energy (E_{ad}) was calculated using the following equation:

$$E_{ad} = E_{\text{adsorbate/slab}} - E_{\text{adsorbate}} - E_{\text{slab}}$$

where $E_{\text{adsorbate/slab}}$, $E_{\text{adsorbate}}$ and E_{slab} represent the adsorbate-adsorbed kaolinite substrate, the adsorbate, and the total energies of the kaolinite substrate, respectively. The 3D iso-surfaces and 1D planar-averaged results of charge density differences (CDD) between the substrate and the adsorbate were obtained by subtracting the electron densities of the isolated substrate and adsorbate from that of the combined system. These calculations were performed using VASPKIT, and the charge density difference visualizations were rendered using the VESTA program.” had been added.

Reviewer #2:

This manuscript introduces effective passivation molecules at the interface between the perovskite and the electron transport layer (ETL), which are distinct from widely used alternatives. The molecules are ferroelectric in nature and are proposed to achieve multifunctionality, including passivation and charge transport. The IV data is properly presented, including both forward and reverse scans as well as steady-state performance. Statistical distributions of PV parameters have been shown. The tandem solar cells show improved light stability after DCI treatment, but they still fall short compared to the state-of-the-art, which demonstrates stability exceeding 1,000 hours. A more detailed analysis of the factors contributing to this discrepancy would be useful, and strategies to further enhance stability should be discussed. A longer stability measurement will be necessary to better understand the stability behavior of the as-prepared cells.

Our reply: We thank the reviewer for the thorough examination of our manuscript and valuable suggestions in terms of tandem stability. To gain deeper insights into the stability behavior, we extended the stability testing duration under ISOS-L-1 protocol to 1,000 hours. As presented in **Figure R1**, the control tandem device suffers from rapid degradation, retaining only ~43.4% of its initial PCE after 640 hours of MPP tracking. In contrast, the DCI-treated tandem device maintains ~85.4% of its original PCE after 1,020 hours of MPP tracking, demonstrating stability comparable to the state-of-the-art perovskite/silicon tandems reported under similar test conditions (**Table R1**).

Figure R1 Continuous MPP tracking of the unencapsulated control and DCI-

treated perovskite/silicon TSCs under 1-sun illumination (100 mW cm^{-2}) with a xenon lamp without UV filter in ambient air ($30 \pm 5 \text{ }^\circ\text{C}$, 40–60% RH).

Table R1 Summary of MPP stability for the state-of-the-art monolithic perovskite/silicon TSCs reported in literatures.

Structure	Environment	Encapsulation	Light condition	PCE retention	Ref.
p-i-n	In ambient air; 25–35%RH; 30 °C	no	1-sun illumination	$T_{92}=100 \text{ h}$	41
p-i-n	In ambient air; 30–40%RH; 25 °C	no	LED with wavelengths of 470 and 940 nm	$T_{95.5}=300 \text{ h}$	42
p-i-n	In N_2 ; 25±1 °C	no	1-sun illumination	$T_{90}=100 \text{ h}$	43
p-i-n	In ambient air; 20%RH; 25 °C	no	1-sun illumination	$T_{98}=300 \text{ h}$	44
p-i-n	In N_2 ; 25±1 °C	no	1-sun illumination	$T_{118}=100 \text{ h}$	45
p-i-n	In N_2 ; 25 °C	no	White LED light source (940 nm LED light for extra compensation)	$T_{98}=200 \text{ h}$	46
p-i-n	In ambient air; 40–70%RH; 20–35 °C	no	Xenon lamp (100 mW cm^{-2})	$T_{93.6}=450 \text{ h}$	47
p-i-n	In ambient air; 25–75%RH, most often ≈60%RH; 20–35 °C	no	Xenon lamp (100 mW cm^{-2})	$T_{86.6}=306 \text{ h}$	48
p-i-n	In N_2 ; 25 °C	no	1-sun illuminations	$T_{80}=1200 \text{ h}$	49
p-i-n	In ambient air; 30–	no	1-sun illuminations	$T_{100}=700 \text{ h}$	50

	40%RH; 25°C In ambient air; 25%RH; 25°C	no	1-sun illuminations	T ₉₆ =527 h	51
p-i-n	In N ₂ ; 25 °C	no	1-sun LED illumination	T ₉₀ =1200 h	52
p-i-n	In ambient air; 40%RH; 25°C	no	1-sun illumination	T _{90.6} =100 h	53
p-i-n	In N ₂ ; 25 °C	no	1-sun illumination	T ₉₅ =280 h	54
n-i-p	In N ₂ ; 25 °C	no	1-sun LED illumination	T ₈₀ =700 h	55
p-i-n	In ambient air; 40– 60%RH; 25–35 °C	no	1-sun xenon- lamp illumination (without a UV filter)	T ₈₀ =755 h	56
p-i-n	In ambient air; 25 °C	yes	AM 1.5G illumination	T _{96.2} =1068 h	37
p-i-n	In ambient air; 40%RH	yes	0.8-sun infrared- enhanced white LED	T ₉₅ =1000 h	9
p-i-n	In ambient air; 25%RH; 25 °C	no	1-sun illumination	T ₉₆ =527 h	30
p-i-n	In ambient air; 40– 60%RH; 25–35 °C	no	1-sun xenon- lamp illumination (without a UV filter)	T _{85.4} =1020 h	This work

Generally, the degradation of perovskites under illumination is closely linked to defect-induced ion migration [Adv. Mater. 2023, 35, 2302552. 10.1002/adma.202302552]. Study has shown that the irreversible deterioration of perovskite film during light soaking primarily originates from interstitial iodide and PbI₂ defects [Science 2020, 369, 96. 10.1126/science.aba1628; Science 2021, 373, 902. 10.1126/science.abi6323; Joule 2022, 6, 816.

[10.1016/j.joule.2022.03.005](https://doi.org/10.1016/j.joule.2022.03.005)]. Passivating those defects by DCI can effectively suppress the iodine generation and migration in perovskite films. In this work, experimental results demonstrate that both interfacial and bulk defects are effectively passivated, benefiting from the synergistic effect enabled by the coordination interaction of multifunctional cage-like diammonium cations and the formation of Pb-Cl ionic bonding. (See below for more details.) On one hand, PbI_2 defects filling the grain boundaries are eliminated, while deep level defects with low formation energy (i.e. I_{Pb} and Pb_{I} antisite defects) at the perovskite surface are calculated to be passivated by cage-like diammonium cations. On the other hand, the formation of strong Pb-Cl ionic bonds throughout the perovskite film enhances the formation energy of interstitial iodide defect, as evidenced by our previous study [[Nano-Micro Lett. 2025, 17, 141. 10.1007/s40820-024-01631-x](https://doi.org/10.1007/s40820-024-01631-x)]. Consequently, defect-induced ion migration is expected to be significantly suppressed.

Furthermore, the activation energy for ion migration was obtained to reveal the ion-induced migration character in control and DCI-treated perovskite films using temperature-dependent conductivity measurements [[Nature 2024, 628, 93. 10.1038/s41586-024-07189-3](https://doi.org/10.1038/s41586-024-07189-3)]. The activation energy value of the control perovskite is fitted to be 46 meV. In contrast, the activation energy value for the DCI-treated film is increased to 59.3 meV (Figure R2a). This result indicates that DCI have a substantially boosted energy barrier for ion migration. This result can also be confirmed by the transient ionic current (TIC) measurements (Figure R2b) as the reduced mobile ion concentration in the DCI-treated device ($2.98 \times 10^{17} \text{ cm}^{-3}$) compared to the control device ($6.72 \times 10^{17} \text{ cm}^{-3}$).

Overall, the enhanced stability of DCI-treated perovskite/silicon tandem devices is mainly attributed to comprehensive defect passivation, driven by the well-defined cation/anion spatial distribution of DCI within the perovskite film, and the significant suppression of ion migration.

Figure R2 (a) Temperature-dependent conductivity of control and DCI-treated perovskite films. (b) Transient ionic current (TIC) for extracting mobile ion concentration within the perovskite films of the control and DCI-treated devices.

In fact, the long-term stability of perovskite/silicon tandems remains a tricky issue, some primary originations being the intrinsic iodide oxidation, phase segregation, residual stress, current mismatch, or extrinsic metal electrode diffusion and silicon surface texture [Nat. Rev. Mater. 2023, 8, 261. 10.1038/s41578-022-00521-1; Inf. Funct. Mater. 2024, 1, 160. 10.1002/ifm2.18]. Additionally, although recent results suggest that improved stability correlate with improved efficiencies under extended light-soaking for perovskite single-junction devices, further research is needed for optimization and accurate evaluation of tandem devices in such aspects, which can notably also differ from single-junction devices due to added complexity of electronics in a tandem device [Nat. Commun. 2023, 14, 4869. 10.1038/s41467-023-40585-3]. At present, further research is urgently required for enhancing the long-term stability of perovskite/silicon TSCs, particularly in addressing intrinsic iodide oxidation and phase segregation [Nat. Rev. Mater. 2023, 8, 261. 10.1038/s41578-022-00521-1]. In response, researchers have developed a range of strategies including component engineering [Nat. Sustain. 2025. 10.1038/s41893-025-01529-5], additive engineering [Science 2024, 385, 533. 10.1126/science.adp1621; Nat. Commun. 2025, 16, 40. 10.1038/s41467-024-55377-6; Adv. Mater. 2025, 2416513. 10.1002/adma.202416513; Nat.

Photonics 2022, 16, 588. 10.1038/s41566-022-01033-8], interface engineering [Nat Commun 2024, 15, 10554. 10.1038/s41467-024-54925-4], dimension engineering [Nat Commun 2024, 15, 7024. 10.1038/s41467-024-51345-2] to mitigate the intrinsic iodide oxidation and phase segregation issues, showing promising progress.

Our response:

The measurements of temperature-dependent conductivity and transient ionic current had been added in **Fig. S15** and **Fig. S16**, respectively.

The extended stability testing duration under ISOS-L-1 protocol had been added to **Fig. S56**.

The summary of MPP stability for the state-of-the-art monolithic perovskite/silicon tandems reported in literatures had been added to **Table S12**.

On **page 20, line 17**, “the control tandem device suffers from a rapid degradation to ~72.2% of the initial PCE after 259 hours of MPP tracking. By contrast, the DCI-treated tandem device still maintains ~95.9% of its original PCE after 219 hours of MPP tracking, highlighting the significant role of DCI in achieving perovskite/silicon TSCs with long-term stability.” had been revised to “the control tandem device suffers from rapid degradation, retaining only ~43.4% of the initial PCE after 640 hours of MPP tracking. In contrast, the DCI-treated tandem device maintains ~85.4% of its original PCE after 1020 hours of MPP tracking, demonstrating stability comparable to the state-of-the-art perovskite/silicon TSCs reported under similar test conditions (Table S12).”.

The reference [Nature 2024, 628, 93. 10.1038/s41586-024-07189-3] had been cited in the main manuscript.

On **page 20, line 25**, “The improved stability of DCI-treated tandems under light soaking is primarily attributed to the synergistic effects of DCI for comprehensively passivating charge trap states (Fig. 1g) and substantially suppressing ion migration (Fig. S15 and S16). Nevertheless, further research

is urgently required for enhancing the long-term stability of perovskite/silicon TSCs, particularly in addressing intrinsic iodide oxidation and phase segregation⁵⁵.” had been added.

On page 1, line 25, “Furthermore, two-terminal monolithic perovskite/silicon tandem solar cells based on tunnel oxide passivating contact yield an impressive PCE of 31.1% (1.0 cm²) and demonstrate a decent operational stability (ISOS-L-1, T₉₀ > 200 hours in ambient conditions without encapsulation).” had been revised to “Furthermore, two-terminal monolithic perovskite/silicon tandem solar cells based on tunnel oxide passivating contact yield an impressive PCE of 31.1% (1.0 cm²) and demonstrate a decent operational stability (ISOS-L-1, T₈₅ > 1020 hours in ambient conditions without encapsulation).”.

On page 3, line 24, “...eventually obtaining an impressive PCE of 31.1% (1.0 cm²) accompanied by an excellent long-term operational stability (95.9% retention after 219 hours under ISOS-standardized maximum power point tracking).” had been revised to “...eventually obtaining an impressive PCE of 31.1% (1.0 cm²) accompanied by an excellent long-term operational stability (85.4% retention after 1020 hours under ISOS-standardized maximum power point tracking).”.

On page 22, line 11, “...eventually enabling a remarkable PCE of 31.1% (1.0 cm²) accompanied by an excellent long-term operational stability under ambient conditions (ISOS-L-1, T₉₀ > 200 h).” had been revised to “...eventually enabling a remarkable PCE of 31.1% (1.0 cm²) accompanied by an excellent long-term operational stability under ambient conditions (ISOS-L-1, T₈₅ > 1020 h).”.

On page 28, line 5, “The current density versus voltage (*J–V*) curves were measured using an Enlitech SS-F5-3A solar simulator based on the Ushio Xe short arc lamp 500 under 1-sun conditions (100 mW cm⁻², AM 1.5G).” had been

revised to “The current density versus voltage ($J-V$) curves, temperature-dependent conductivity and transient ionic current were measured using an Enlitech SS-F5-3A solar simulator based on the Ushio Xe short arc lamp 500 under 1-sun conditions (100 mW cm^{-2} , AM 1.5G).”.

#2-1. The reported efficiency of 21.9% (before poling) is relatively moderate compared to recent publications, where efficiencies for wide-bandgap perovskite solar cells have surpassed 23%: e.g. *Advanced Materials*, 2024. 36(9): p. 2307701; *Nature Communications*, 2024. 15(1): p. 1-11; *Advanced Materials*, 2024: p. 2407681. The V_{oc} was around 1.22 V, considerably lower the state-of-the-art values of >1.25 V indicating a suboptimal interface that could impact both the performance and the stability. Could you elaborate on potential reasons for this lower V_{oc} and how the interface might be further optimized?

Our response: We sincerely thank the reviewer’s comments and suggestions. Indeed, we found that the efficiency of our reported device is relatively moderate compared to recent publications (Table R2). After thoroughly reviewing the reviewer-recommended and recently published literature on wide-bandgap (WBG) perovskite (~ 1.68 eV), we attribute the efficiency gap primarily to two key factors.

The first is the choice of hole transport layers (HTLs). Compared to MeO-2PACz, 2PACz and conventional PTAA HTLs, Me-4PACz, 4PADCB HTLs or their combinations with NiO_x exhibit higher V_{oc} and PCE, which is primarily due to the improved hole extraction and suppression of non-radiative recombination [*Science* 2020, 370, 1300. [10.1126/science.abd4016](https://doi.org/10.1126/science.abd4016)]. In the abovementioned references, replacing MeO-2PACZ with 4PADCB as the HTL enhanced the V_{oc} of WBG perovskite solar cells from 1.22 to 1.30 V, even surpassing the 1.29 V achieved with NiO/Me-4PACz due to the superior hole extraction capability and

energy level alignment.

The second factor involves the use of additives. As summarized in **Table R2**, the incorporation of additives further enhances the V_{oc} of WBG perovskite solar cells by effectively passivating bulk defects and improving crystallization.

In our work, we focus on defect passivation, energy level alignment modulation and ferroelectric effect at perovskite/ C_{60} interface. Despite employing a MeO-2PACz hole transport layer, our DCI yields a WBG cell with a V_{oc} of 1.23 V and an impressive fill factor of 83.2%, resulting in a final efficiency of 21.9% surpassing that of devices utilizing polymer. We highlight the significance of defects passivation in enhancing device stability, as well as the roles of interfacial dipole and ferroelectric effect in facilitating charge extraction.

Finally, we also explored the potential of DCI-based WBG perovskite devices by employing alternative SAMs such as 4PADCB as the HTL. This resulted in an improved PCE of 22.9%, with a V_{oc} of 1.255 V, an FF of 84.7% and a J_{sc} of 21.6 mA cm^{-2} (**Figure R3**), comparable to recently reported advanced 1.68 eV perovskite solar cells. This indicates that the choice of an appropriate hole transport layer is crucial for optimizing device performance. These data are not included in current manuscript, since it is beyond our original scope of this work. Considering the better hole transport capability of 4PADCB, we will apply it in our future study on WBG perovskite and tandem solar cells.

Table R2 Summary of performance of advanced single-junction WBG perovskite solar cells.

E_g (eV)	HTLs	Additives	Post-treatment	V_{oc} (V)	PCE (%)	Ref.
1.67	Me- 4PACz	Pb(SCN) ₂	EDA _l ₂	1.24	23.1	28
1.66	4PADCB	RbSCN	EDA _l ₂ +PEAI	1.30	24.5	29

1.68	PTAA	MACl+PbCl ₂	GUACl+OAml	1.23	21.58	30
1.68	NiOx/Me-4PACz	-	Al ₂ O ₃ nano-polishing+PI	1.29	24.5	31
1.67	Ph-2PACz	Pb(SCN) ₂ +PEAI	LiF	1.26	21.3	32
1.68	NiO	ICPS	LiF	1.27	22.0	33
1.68	Me-4PACz	-	PI	1.27	20.6	34
1.68	NiO/MeO-4PADBC	MACl+ODAl ₂	CF ₃ -PEAI	1.25	22.7	35
1.68	MeO-2PACz	-	PEABr+5ATT	1.25	22.6	36
1.68	NiO/Me-4PACz	MACl+AESCI	-	1.28	22.8	37
1.68	NiO/MeO-2PACz	TAACl	CF ₃ -PEAI	1.28	23.0	38
1.68	MeO-4PACz	MDACl	CF ₃ -PEAI+LiF	1.27	23.2	39
1.68	Bz-PhpPACz	-	PDI+PEAI	1.27	22.5	40
1.68	MeO-2PACz	-	DCI	1.23	21.9	This work

Figure R3 J-V curves of the DCI-treated PSC based on 4PADCB HTL.

In addition to SAM HTLs and additive engineering, interface engineering of perovskites is also critical. As shown in Table R2, dual-molecule passivation, either co-deposition or sequential deposition, has recently gained popularity for interfacial optimization, where complementary functionalities are employed to address the defective perovskite surface. Ongoing work in our group is exploring such interface engineering approaches to suppress non-radiative recombination and maximize V_{oc} .

#2-2. Although performance improves to over 22% after poling, a key question remains: how much will this enhancement contribute to energy yield over time? The ferroelectric phenomenon in the DCI-treated samples raises important concerns: Is the effect temporary or permanent? Given the light-dark cycling of solar cells, would the cells require additional biasing after being in the dark to achieve optimal ferroelectric performance again?

Our reply: We sincerely appreciate the reviewer's insightful comments. To evaluate whether the enhanced performance after poling leads to a meaningful increase in energy yield over time, we carried out the light-dark cycle measurements by MPP-tracking under continuous AM1.5G illumination in ambient air conditions [Energy Environ. Sci. 2017, 10, 604. 10.1039/C6EE03352K; Adv. Energy Mater. 2023. 10.1002/aenm.202302147]. As presented in Figure R4, the control device shows a continuous drop in maximum power point (P_{max}) without any recovery after storage in the dark, indicating poor stability. In addition, the control device consistently exhibits a gradual increase in P_{max} during each light cycle, which is detrimental to achieving optimal energy output under realistic light-dark cycling conditions. In contrast, the DCI-treated and poled DCI-treated devices maintain stable performance without noticeable degradation during 40 hours of day-night cycle testing. In case of DCI-treated and poled DCI-treated devices, although the P_{max}

drops to 90-95% of its initial value for each light cycle of 4 hours, it fully recovers to its original level after a subsequent 4-hour dark period. This reversible behavior demonstrates a self-healing effect, which could be practically beneficial for commercial perovskite solar cell operation, as it enables nighttime self-recovery from the ~5-10% photodegradation accumulated during daytime use [Energy Environ. Sci. 2017, 10, 604. 10.1039/C6EE03352K; Adv. Energy Mater. 2023. 10.1002/aenm.202302147]. Moreover, in each light cycle, the P_{\max} values of the DCI-treated and poled DCI-treated devices immediately reach their optimal levels at the onset of illumination, which is favorable for maximizing energy output under realistic light-dark cycling conditions. Notably, the poled DCI-treated device delivers a higher output power in the initial light cycle but converges to the same performance level as the DCI-treated device in the following cycles, indicating that the ferroelectric effect contributes to a slight, yet short-lived, enhancement in energy yield.

Figure R4 Operational stability of control, DCI-treated and poled DCI-treated devices by MPP tracking with an artificially induced day-night cycle.

This behavior may be ascribed to the transient ferroelectric characteristic of the DCI-mediated quasi-2D perovskite. In fact, a permanent ferroelectric state would be energetically unfavorable for 2D perovskites, as it would inevitably compromise the structural stability of the 2D perovskite, which is

supported by previous reports [Nat. Commun. 2024, 15, 8753. 10.1038/s41467-024-53121-8] (Figure R5). Before poling, the ferroelectric organic cations predominantly occupy the A-sites to maintain the structural integrity of the quasi-2D lattice. Upon applying an external electric field, these cations undergo ferroelectric switching and deviate from their equilibrium positions, generating an internal depolarization field within the ferroelectric spacer layers that facilitates charge transport. However, during the dark period, the organic cations gradually return to their original A-site locations to restore the structural stability of the 2D perovskite. This dynamic process may explain why the poled DCI-treated device delivers enhanced power output during the initial light cycle but converges to the same performance level as the unpoled DCI-treated device in subsequent cycles. These observations suggest that additional external biasing may be required after each dark period to reactivate the ferroelectric switching and maintain improved device performance.

Figure R5 (a) Atomic geometry and schematic diagram illustrating the 2D/3D interface with an antiferroelectric molecular configuration, where the black arrows indicate the electric dipole directions of ferroelectric organic cations and E_z^{ferr} represents the resulting electric field along the z direction. (b) Atomic geometry and schematic diagrams for the 2D/3D interface in a ferroelectric configuration, with the net dipole along the z -direction towards the 3D perovskite [Nat. Commun. 2024, 15, 8753. 10.1038/s41467-024-53121-8].

Actually, permanent ferroelectricity can also be achieved by decoupling

these ferroelectric organic materials from the structural constraints of the inorganic framework of 2D perovskite, thereby allowing them to exist and function independently [Adv. Mater. 2022, 34, 2110482. 10.1002/adma.202110482]. For instance, these ferroelectric organic molecules can be thermally evaporated onto the perovskite surface to form a separate organic layer. Owing to their intrinsic polarity or permanent dipole moments, the molecules can spontaneously align at the surface, thereby giving rise to stable and long-lasting ferroelectric polarization. This approach will be further explored in our future work.

Our response:

Operational stability of control, DCI-treated and poled DCI-treated devices by MPP tracking with an artificially induced day-night cycle had been added to Fig. S50.

On page 18, line 28, “Light-dark cycle measurements further indicate that the ferroelectric effect offers a slight but transient boost in energy yield, attributed to structural coupling within the 2D perovskite (Fig. S50).” had been added.

On page 29, line 10, “Light-dark cycle testing of unencapsulated devices was conducted in N₂ at MPP using a commercial multichannel stability test system (Wuhan 91PVKSolar), with 4 h illumination followed by 4 h recovery in the dark.” had been added.

#2-3. The manuscript claims that DCI induces a p-type to n-type surface energetic transformation. However, the fundamental mechanism for this transformation remains unclear. Please elaborate on how DCI facilitates such behavior.

Our reply: We sincerely appreciate the reviewer’s valuable comments and suggestions. To elucidate the fundamental mechanism underlying the surface

energy transformation, we performed DFT calculations to investigate the evolution of the perovskite surface work function before and after DCI treatment. As shown in **Figure R6**, before the adsorption of cage-like cations, the control perovskite exhibits a high SWF of -1.95 eV (**Figure R6a**). Once the cage-like cations adsorb onto the perovskite surface (**Figure R6b**), the SWF dramatically decreases to -0.25 eV, corresponding to an upshift of 1.7 eV relative to the untreated control, which is sufficient induce a reversal of the surface energy. This significant decrease in SWF is mainly attributed to the strong interfacial dipole generated by the spontaneous polarization of cage-like cations (**Figure R6b**) [Physical Review B 2003, 68, 195408. 10.1103/PhysRevB.68.195408; Energy Environ. Sci. 2021, 14, 1429. 10.1039/D0EE02216K]. The charge density difference reveals that this interfacial dipole significantly drives charge transfer from the perovskite to the cage-like cations (**Figure R7**), thereby decreasing the SWF [ACS Energy Lett. 2024, 9, 2982. 10.1021/acsenerylett.4c00839; Joule 2021, 5, 467. 10.1016/j.joule.2020.12.009]. Notably, this theoretical result aligns well with the surface work function changes observed in UPS measurements.

Figure R6 DFT calculations of the surface work functions of perovskite without and with adsorption of cage-like cations. (a) Bare perovskite, (b) Perovskite with cage-like cations adsorption.

Figure R7 Charge density difference of perovskite with cage-like cation adsorption. Yellow iso-surface signifies electron accumulation, while blue represents electron depletion.

Our response:

DFT calculated surface work functions of perovskite without and with adsorption of cage-like cations and the charge density difference had been added to Fig. 3c, Fig. 3d and Fig. 3e, respectively.

The corresponding reference [Joule 2021, 5, 467. 10.1016/j.joule.2020.12.009] had been cited.

On page 13, line 4, “Subsequent DFT calculations were conducted to explore the evolution of the perovskite surface work function (SWF). As shown in Fig. 3c, the control perovskite exhibits a high SWF of -1.95 eV. Upon adsorption of cage-like cations, the SWF dramatically decreases to -0.25 eV (Fig. 3d), corresponding to an upshift of 1.7 eV, consistent with UPS and KPFM measurements. Additionally, the introduction of cage-like cations induces a strong interfacial dipole originating from their spontaneous polarization (Fig. 3d). The charge density difference reveals that this interfacial dipole significantly drives charge transfer from the perovskite surface to the cage-like cations, thereby decreasing the SWF (Fig. 3e)⁴⁸.” had been added.

#2-4. Similarly, the claim that molecular dipoles formed by the cage-like

diammonium cation improve surface energy band alignment by altering the surface work function requires further explanation. Please provide detailed mechanisms for this process.

Our reply: We sincerely thank the reviewer's thoughtful feedback and helpful recommendations. In #2-3, we have performed DFT calculations to investigate the evolution of the perovskite surface work function before and after DCI treatment. In short, this significant decrease in SWF is mainly attributed to the strong interfacial dipole generated by the spontaneous polarization of cage-like cations (Figure R6), which can significantly drive charge transfer from the perovskite surface to the cage-like cations (Figure R7).

Our response:

DFT calculated surface work functions of perovskite without and with adsorption of cage-like cations and the charge density difference had been added to Fig. 3c, Fig. 3d and Fig. 3e, respectively.

The corresponding reference [Joule 2021, 5, 467. 10.1016/j.joule.2020.12.009] had been cited.

On page 13, line 4, "Subsequent DFT calculations were conducted to explore the evolution of the perovskite surface work function (SWF). As shown in Fig. 3c, the control perovskite exhibits a high SWF of -1.95 eV. Upon adsorption of cage-like cations, the SWF dramatically decreases to -0.25 eV (Fig. 3d), corresponding to an upshift of 1.7 eV, consistent with UPS and KPFM measurements. Additionally, the introduction of cage-like cations induces a strong interfacial dipole originating from their spontaneous polarization (Fig. 3d). The charge density difference reveals that this interfacial dipole significantly drives charge transfer from the perovskite surface to the cage-like cations, thereby decreasing the SWF (Fig. 3e)⁴⁸." had been added.

#2-5. The authors state that the cage-like diammonium cation contains both

electron donor (R_3N) and electron acceptor (R_3NH^+) functional groups, which passivate both negative and positive surface trap states. Please specify which types of positive and negative trap states are passivated by DCI.

Our reply: Thanks for the reviewer's valuable suggestion. To elucidate the passivation mechanism for DCI treatment, we used density functional theory (DFT) calculations to compare the formation energies of selected native defects on the perovskite surface. Four prevalent surface defects, including both positive Pb^{2+} traps and negative halide traps are considered: I vacancy (V_I), Pb vacancy (V_{Pb}), Pb-I antisite (Pb_I) and I-Pb antisite (I_{Pb}) (Figure R8) [Science 2019, 366, 1509. 10.1126/science.aay9698; Energy Environ. Sci. 2024, 17, 9601. 10.1039/d4ee02803a].

Figure R8 Front-view and side-view of various types of defects, including V_I , Pb_I , V_{Pb} and I_{Pb} defects.

Among these four defects, V_I and Pb_I are positively charged while V_{Pb} and I_{Pb} carry negative charges [Nature Reviews Chemistry 2023, 7, 632. 10.1038/s41570-023-00510-0]. Evidently, both positively charged defects (V_I and Pb_I) and negatively charged defects (V_{Pb} and I_{Pb}) can be effectively passivated by DCI decoration (Figure R9) owing to its multifunctional groups (R_3N and R_3NH^+). Similar results have also been reported before [J. Am. Chem. Soc. 2020, 142, 20134. 10.1021/jacs.0c09845; Science 2023, 381, 63.

[10.1126/science.adf5872](https://doi.org/10.1126/science.adf5872); Nano-Micro Lett. 2025, 17, 141. [10.1007/s40820-024-01631-x](https://doi.org/10.1007/s40820-024-01631-x)].

We further compared the defect formation energies (DFEs) of V_I , Pb_I , V_{Pb} and I_{Pb} on the surface as depicted in Figure R10. The DFEs for V_I , Pb_I , V_{Pb} and I_{Pb} are calculated to be 2.02, 1.86, 1.90 and 1.23 eV, respectively (Figure R10a). Obviously, the negatively charged I_{Pb} and positively charged Pb_I antisite defects demonstrate lower formation energies compared to other defects, making more likely to form and predominate on the surface. Additionally, it is noteworthy that I_{Pb} and Pb_I are deep level defects, thus preferentially serving as primary non-radiative recombination centers. Thus, we focused on the passivation mechanism of DCI on I_{Pb} and Pb_I defects, as illustrated in Figure R10b, c and Figure R11. When DCI is introduced onto the perovskite surface which contains I_{Pb} antisite defects, the deep level defect states are effectively inhibited (Figure R10b and c). This could be attributed to the formation of chemical bonds between I and R_3NH^+ group. A similar passivation effect could also be found on the Pb_I antisite-dominated surface (Figure R11).

Based on the DFT calculations, we can conclude that both positively and negatively charged defects, particularly Pb_I and I_{Pb} antisite defects, can be effectively passivated by DCI due to its multifunctional groups, which is conducive to suppressing nonradiative recombination, and thus enhancing device performance and stability.

Figure R9 Top-view of various types of defects passivated by DCI.

Figure R10 (a) The defect formation energies of various types of defects on perovskite surface (the inset represents the top view). (b,c) The partial density of states (PDOSs) of (b) the defective perovskite (I_{Pb} antisite) and (c) passivated perovskite by DCI.

Figure R11 The partial density of states (PDOSs) of (a) the defective perovskite (Pb_I antisite) and (b) passivated perovskite by DCI.

Furthermore, thermal admittance spectroscopy analysis was undertaken to distinguish the trap states in perovskite films (Figure R12). For the DCI-based

device, the reduction of trap states in the band range from 0.55 to 0.65 eV indicates that the deep trap states located in perovskite films are effectively passivated by the incorporation of DCI, which is consistent with the higher PLQY value and longer carrier lifetime for perovskites with DCI.

Figure R12 tDOS obtained from thermal admittance spectroscopy of control and DCI-treated devices.

Our response:

Top-view of various types of defects passivated by DCI had been added to **Fig. S3**.

On **page 4, line 5**, “the cage-like diammonium cation contains both electron donor (R_3N) and electron acceptor (R_3NH^+) functional groups, which can passivate both negative and positive surface trap states, thus reducing nonradiative recombination.” had been revised to “the cage-like diammonium cation contains both electron donor (R_3N) and electron acceptor (R_3NH^+) functional groups, which can passivate both negative and positive surface trap states (**Fig. S3**), thus reducing nonradiative recombination.”.

Front-view and side-view of various types of defects, including V_I , Pb_I , V_{Pb} and I_{Pb} defects had been added to **Fig. S24**.

The defect formation energies of various types of defects on perovskite surface had been added to **Fig. 2g**.

The partial density of states (PDOSs) of the defective perovskite (I_{Pb} antisite) and passivated perovskite by DCI had been added to **Fig. 2h** and **Fig. 2i**, respectively.

The partial density of states (PDOSs) of the defective perovskite (Pb_i antisite) and passivated perovskite by DCI had been added to **Fig. S25**.

The corresponding reference [Science 2019, 366, 1509. 10.1126/science.aay9698] had been cited.

tDOS obtained from thermal admittance spectroscopy for devices without and with DCI modification had been added to **Fig. S26**.

On page 29, line 3, “Trap density of state (tDOS) curves were obtained using a 1240A Impedance Analyzer.” had been added.

On page 10, line 11, “To gain deeper insights into the passivation mechanisms of DCI on the perovskite surface, we conducted DFT calculations considering four prevalent surface defects: I vacancy (V_I), Pb-I antisite (Pb_i), Pb vacancy (V_{Pb}), and I-Pb antisite (I_{Pb}) (Fig. S24)⁴³. Computational analysis reveals that deep-level defects, particularly I_{Pb} , exhibit lower formation energies than shallow-level defects (Fig. 2g). Consequently, both negatively charged I_{Pb} and positively charged Pb_i antisite defects form more readily and predominate on the surface, making them the primary non-radiative recombination centers. Upon DCI adsorption, deep-level states associated with I_{Pb} defects are effectively inhibited (Fig. 2h and i), likely due to chemical interactions between negative halide traps and the R_3NH^+ group. A similar passivation effect is also observed on the Pb_i -dominated surface (Fig. S25). Thermal admittance spectroscopy analysis further reveals the reduced deep trap states following DCI incorporation (Fig. S26).” had been added.

In the part of “**DFT calculations**”, “**Calculations of defect formation energy, partial density of state, surface work function and charge density difference** All first-principles calculations based on DFT were performed using VASP. The electron-ion interactions were described by projector augmented-wave (PAW) pseudopotentials, and exchange-correlation effects were treated using the PBE functional within the generalized gradient approximation (GGA).

The slab model was constructed from a cubic-phase primitive cell with a 2×2 surface supercell comprising eight atomic layers. The bottom four layers were fixed, while the remaining atoms were fully relaxed. A vacuum layer of at least 20 Å was introduced along the z-direction to prevent periodic interactions. Calculations employed a plane-wave energy cutoff of 500 eV and a k-point mesh of 1×1×1 for slab models (2×2×1 for DOS calculations). Structural optimization was performed with a force convergence threshold of 0.05 eV/Å and an energy convergence criterion of 1×10⁻⁵ eV. Van der Waals interactions were accounted for using the Grimme D3-BJ correction. The defect formation energy is calculated as:

$$E_F = E_{\text{slab+defect}} - E_{\text{slab}} \pm \sum n_i \mu_i$$

Where $E_{\text{slab+defect}}$ is the energy of the slab with defects, E_{slab} is the energy of defect-free slab, n_i is the number of atoms added or removed, μ_i is the chemical potential of the corresponding element.” had been added.

#2-6. The manuscript notes that the DCI-treated film exhibits higher RMS surface roughness (from 13.6 nm to 18.6 nm) due to additional island-like substances and columnar-like protrusions, which are said to optimize interfacial contact with the ETL. Considering the ETL is ultrathin (~10 nm), please clarify how a rougher surface facilitates improved interfacial contact.

Our reply: We appreciate the reviewer’s insightful comment regarding the relationship between surface roughness and interfacial contact with the ultrathin ETL layer. We have re-investigated the surface morphology and topology of control and DCI-treated perovskite films to verify our previous results. As shown in Figure R13, a number of discrete sheet-like PbI₂ grains filling the grain boundaries of the 3D perovskite in the control film is observed. Upon DCI modification, the perovskite surface undergoes reconstruction, forming numerous small grains (i.e. island-like substances) within the perovskite grain

surfaces, accompanied by the disappearance of residual PbI_2 . The above phenomenon indicates that DCI can coordinate well with perovskite surface and residual PbI_2 .

Figure R13 Top-view SEM images of the (left) control and (right) DCI-treated perovskite films.

Surface topology of the films without and with DCI post-treatment was then characterized using atomic force microscope (AFM). As demonstrated in **Figure R14**, consistent with the SEM images, numerous small grains can be clearly observed in the DCI-treated perovskite film. Surprisingly, the DCI-treated film exhibits a slightly lower root mean square (RMS) surface roughness than the control one, decreasing from 19.2 nm to 18.4 nm, which contradicts our initial interpretation. Furthermore, to improve the reliability of our morphological characterization, we conducted a more comprehensive AFM analysis over a wider scan range (from $1 \mu\text{m} \times 1 \mu\text{m}$ to $5 \mu\text{m} \times 5 \mu\text{m}$) to better represent the film morphology, as illustrated in **Figure R15**. Interestingly, the updated RMS roughness of the DCI-treated film decreased from 21.9 nm to 20.8 nm after treatment, indicating a slightly smoother surface rather than a rougher one.

The decrease of RMS of the DCI-treated perovskite surface will lead to a good contact with subsequently deposited C_{60} layers, as evidenced by the previous work [Adv. Mater. 2025, 37, 2500501. 10.1002/adma.202500501; Nat Commun 2024, 15, 5607. 10.1038/s41467-024-50019-3; Nature 2022, 613, 676. 10.1038/s41586-022-05541-z; Nature 2022, 611, 278. 10.1038/s41586-022-05268-x].

Notably, although the surface roughness remains relatively large after DCI modification, the C₆₀ electron transport layer can still form a uniform and conformal coating on the perovskite surface due to its thermal evaporation deposition. In addition, the thickness of C₆₀ is 15 nm (as described in **methods**). Generally, a thickness of just 2.3 nm is sufficient to enable complete C₆₀ coverage, as supported by the previous study [Adv. Energy Mater. 2022, 12, 2103567. 10.1002/aenm.202103567].

Figure R14 AFM images of the (left) control and (right) DCI-treated perovskite films with a scan range of 1 $\mu\text{m} \times 1 \mu\text{m}$.

Figure R15 AFM images of the (left) control and (right) DCI-treated perovskite films with a scan range of 5 $\mu\text{m} \times 5 \mu\text{m}$.

Our response:

The top-view SEM images of the control and DCI-treated perovskite films had been re-investigated in **Fig. 1a**.

The AFM images of the control and DCI-treated perovskite films with a scan range of 1 $\mu\text{m} \times 1 \mu\text{m}$ had been re-investigated in **Fig. 1b**.

The AFM images of the control and DCI-treated perovskite films with a scan range of 5 $\mu\text{m} \times 5 \mu\text{m}$ had been added to **Fig. S4**.

On page 4, line 19, “On one hand, some columnar-like substances, featuring out-of-plane epitaxial growth, are discretely distributed at grain boundaries (GBs) of the 3D perovskite. On the other hand, many island-like clusters with a relatively low surface coverage are randomly distributed within the grain interiors (GIs) of the 3D perovskite.” had been revised to “The perovskite surface undergoes reconstruction, forming numerous small grains uniformly distributed at the perovskite grain surfaces, accompanied by the disappearance of residual PbI_2 .”.

On page 4, line 21, “These results indicate that cage-like diammonium cation, similar to fluorinated plane-type ammonium cations, can also react with PbI_2 and 3D perovskite phase to generate discrete new substances, spontaneously assembling on the top surface of the 3D perovskite.” had been revised to “These results indicate that cage-like diammonium cation, similar to fluorinated plane-type ammonium cations, can also react with PbI_2 and 3D perovskite phase to generate new substances, spontaneously assembling on the top surface of the 3D perovskite.”.

On page 4, line 26, “As demonstrated in Fig. 1b, consistent with the SEM images, many island-like substances randomly assemble within the GIs and localized bright protrusions can be clearly observed in the DCI-treated perovskite film.” had been revised to “As demonstrated in Fig. 1b and Fig. S4, consistent with the SEM images, numerous uniformly assembled small grains are distinctly observed on the DCI-treated perovskite surface.”.

The corresponding reference [Nature 2022, 611, 278. 10.1038/s41586-022-05268-x] had been cited.

On page 5, line 1, “In addition, the DCI-treated film exhibits a slightly higher root mean square (RMS) surface roughness than the control one, increasing from 13.6 nm to 18.6 nm, likely in consequence of the additional island-like substances and columnar-like protrusions, which optimizes the interfacial

contact of the perovskite with electron transport layer.” had been revised to “In addition, the DCI-treated film exhibits a slightly **decreased** root mean square (RMS) surface roughness than the control one (**Fig. 1b and Fig. S4**), which optimizes the interfacial contact of the perovskite with electron transport layer²⁸”.

#2-7. In Figure S9, the PL spectra show significant drops at certain wavelengths for both the control and target samples. Please clarify the potential causes for these drops in the photoluminescence intensity.

Our reply: Thank you for your careful observation. We acknowledge that the PL spectra of both control and target samples show intensity drops at the same wavelength position, which is primarily due to the measurement mode. In the original manuscript, the PL spectra were obtained using the stitched mode of the grating rather than the continuous mode, which may lead to abrupt intensity discontinuities at specific wavelengths. To address this, we remeasured the PL spectra in continuous mode, as shown in **Figure R16**. The resulting spectra are smooth, and the DCI-treated perovskite film displays significantly enhanced PL intensity compared to the untreated sample, consistent with the trend observed in the original stitched-mode data. Accordingly, we have updated **Fig. S22** in the revised Supporting Information with the remeasured PL spectra.

Figure R16 PL spectra obtained in the continuous mode.

Our response:

The PL spectra had been revised in **Fig. S22**.

On **page 27, line 13**, “Steady-state photoluminescence (SSPL) and corresponding PL images were obtained using a Micro-Raman spectrometer (Renishaw inVia Reflex)” had been revised to “Steady-state photoluminescence (SSPL) (**continuous mode**) and corresponding PL images were obtained using a Micro-Raman spectrometer (Renishaw inVia Reflex).”

#2-8. In Figure S3, the XRD pattern shows a noticeable background signal. Could you please remove this background signal to improve the clarity and focus of the data?

Our reply: Thank you for your precious suggestion and kind reminder. The background signal of the raw XRD pattern has been removed as illustrated in **Figure R17**. Following signal purification process, the XRD pattern exhibits effective background subtraction, revealing enhanced diffraction peak clarity. Consistent with the original description, no new phase is formed at low angles as the concentration of DCI raises from 0.1 mg mL^{-1} to 0.6 mg mL^{-1} , and the best crystallinity of perovskite film is achieved at 0.4 mg mL^{-1} .

Figure R17 XRD patterns of the control and DCI-treated perovskite films following background signal purification process.

Our response:

The XRD pattern had been revised in **Fig. S5**.

#2-9. Figure S2: Please add clear labels to indicate what each symbol represents.

Our reply: Thanks for your valuable suggestion. The chemical labels have been added. Additionally, the chemical structures, electrostatic potentials (ESPs) and corresponding electric dipole moments of both 1,4-diazabicyclo[2.2.2]octane (DI) and DCI have also been systematically compared to highlight the uniqueness of our designed DCI. As shown in **Figure R18a-c**, DI possesses a three-dimensional cage-like architecture, where each vertex is terminated by an identical R_3N Lewis base group, forming a symmetrical configuration with uniform electron-donating sites. In this case, only positive surface defects (such as Pb^{2+} traps) can be passivated. In addition, this molecular configuration inherently suppresses dipole moment formation through geometric cancellation of opposing charge vectors, resulting in a vanishing net electric dipole moment ($\mu = 0$ Debye), thus having no impact on the surface work function [*Energy Environ. Sci.* 2021, 14, 1429. [10.1039/D0EE02216K](https://doi.org/10.1039/D0EE02216K)]. In contrast, in the case of the asymmetrical cage-like

diammonium cation (Figure R18d-f), each side terminates with a R_3NH^+ Lewis acid group carrying a strong positive charge and a R_3N Lewis base group with a strong negative charge, resulting in a large electric dipole moment ($\mu = 4.08$ Debye). Therefore, both positively and negatively charged traps (e.g. Pb^{2+} and halide ion traps) can be effectively passivated by DCI [J. Am. Chem. Soc. 2020, 142, 20134. 10.1021/jacs.0c09845] and the surface energy level can also be tuned by the oriented molecular packing and spontaneous polarity of DCI [Nat. Commun. 2025, 16, 835. 10.1038/s41467-025-56182-5].

Figure R18 Chemical structures, electrostatic potentials (ESPs) and corresponding electric dipole moments of both 1,4-diazabicyclo[2.2.2]octane and cage-like diammonium cation.

Our response:

The chemical labels, chemical structures, electrostatic potentials (ESPs) and corresponding electric dipole moments of both 1,4-diazabicyclo[2.2.2]octane (DI) and DCI had been added in **Fig. S2**.

On **page 4, line 3**, “Fig. S2 depicts the 3D structure of the cage-like diammonium cation.” had been revised to “Fig. S2 depicts the chemical structure, electrostatic potential (ESP) and corresponding electric dipole moment of the cage-like diammonium cation.”.

On page 4, line 9, “Additionally, the formation of molecular dipole for the cage-like diammonium cation itself can improve surface energy band alignment of perovskite by altering the surface work function” had been revised to “Additionally, the formation of molecular dipole (4.08 Debye) due to the spontaneous polarity of the cage-like diammonium cation can improve surface energy band alignment of perovskite by altering the surface work function”.

In the part of “**Method**”, “**DFT calculations**” had been added.

In the part of “**DFT calculations**”, “**Calculation of electrostatic potential and dipole moment**” The geometries of 1,4-diazabicyclo[2.2.2]octane and cage-like diammonium cation were optimized within the density functional theory (DFT) framework using the M06-2X functional, coupled with the DFT-D3 dispersion correction method and the 6-31G(d) basis set. For the theoretical calculations of molecular dipole moments, accuracy is highly sensitive to the choice of basis set due to the strong dependence of molecular polarization properties on diffuse functions. Herein, the def2-TZVPD basis set was employed for dipole moment calculations. All DFT computations were conducted with the Gaussian 16 program. Electrostatic surface potentials and extrema points were calculated using the Multiwfn program, while visualization of the electrostatic surface potentials was rendered using the Visual Molecular Dynamics (VMD) program.” had been added.

Reviewer #3:

In this work, the authors synthesize a multifunctional cage-like diammonium chloride molecule (DCI) and incorporate it into the perovskite/C₆₀ interface to reduce energy losses. Furthermore, the authors claimed that the DCI molecule promotes the formation of a pure-phase quasi-2D perovskite with favorable crystal orientation and ferroelectric properties, further improving charge separation and extraction. As a result, DCI-treated perovskite solar cells achieve high efficiencies (22.6% for 0.1 cm² and 21.0% for 1.21 cm²), and when incorporated into perovskite/silicon tandem cells, a record PCE of 31.1% is achieved. Though the manuscript is interesting but it currently shows many scientific flaws that restrict its publication in Nature Communications.

Our response: We thank the reviewer for the overall positive comment of our work. We have addressed all comments and believe that the revision has led to an improved manuscript.

#3-1. As depicted in Figures 1a and 1b. It seems that DCI materials didn't coordinate well with the perovskite layer as there is no change in surface crystallinity and morphology. The post-treated material can be seen deposited on top of perovskite crystals. The increased surface roughness also confirms this. So, the authors should carefully reinvestigate this and should provide its diffusion properties.

Our reply: We thank the reviewer for these valuable comments and suggestions. We have re-investigated the surface morphology and crystallinity of control and DCI-treated perovskite films. As shown in **Figure R1**, a number of discrete sheet-like PbI₂ grains filling the grain boundaries of the 3D perovskite in the control film is observed. Upon DCI modification, the perovskite surface undergoes reconstruction, forming numerous small grains within the perovskite grain surfaces, accompanied by the disappearance of residual PbI₂. The above

phenomenon indicates that DCI can coordinates well with perovskite surface and residual PbI_2 . In addition, GIWAXS measurements reveal that the surface crystallinity of the perovskite is increased upon DCI incorporation, accompanied by the formation of a new 2D phase (Figure R2a) and a reduction in PbI_2 content (Figure R2b)

Figure R1 Top-view SEM images of the (left) control and (right) DCI-treated perovskite films.

Figure R2 (a) The (001) in-plane line cuts of GIWAXS images of the control and DCI-treated perovskite films. (b) The (001) out-of-plane line cuts of GIWAXS images of the control and DCI-treated perovskite films.

Surface topology of the films without and with DCI post-treatment was then characterized using atomic force microscope (AFM). As demonstrated in Figure R3, consistent with the SEM images, numerous small grains can be clearly observed in the DCI-treated perovskite film. In addition, the DCI-treated film exhibits a slightly lower root mean square (RMS) surface roughness than the control one, decreasing from 19.2 nm to 18.4 nm, which is expected to optimize the interfacial contact of the perovskite with electron transport layer [Adv. Mater.

2025, 37, 2500501. 10.1002/adma.202500501; Nat Commun 2024, 15, 5607. 10.1038/s41467-024-50019-3; Nature 2022, 613, 676. 10.1038/s41586-022-05541-z; Nature 2022, 611, 278. 10.1038/s41586-022-05268-x]. A wider AFM scan further confirmed a slightly smoother surface after treatment, with the RMS surface roughness decreasing from 21.9 nm (pristine) to 20.8 nm (Figure R4).

Figure R3 AFM images of the (left) control and (right) DCI-treated perovskite films with a scan range of $1 \mu\text{m} \times 1 \mu\text{m}$.

Figure R4 AFM images of the (left) control and (right) DCI-treated perovskite films with a scan range of $5 \mu\text{m} \times 5 \mu\text{m}$.

Notably, these small grains on the DCI-treated perovskite surface (Fig. 1a and b) should be ascribed to the formation of a 2D phase rather than the post-treated DCI material itself, as evidenced in Figure R5. The peak position at 10.2° of the synthesized single-crystal 2D perovskite precisely matches that observed in GIXRD (Fig. S4) and GIWAXS measurements (Figure R2a)

Figure R5 XRD patterns of DCI powder and synthesized single-crystal 2D perovskite.

To further identify the distribution states of DCI on the perovskite surface, we performed atomic force microscopy-based infrared spectroscopy (AFM-IR) measurements on both the control and DCI-treated perovskite films. AFM-IR is a powerful technique for analyzing the chemical composition distribution of halide perovskite materials, offering a significantly higher spatial resolution—better than 20 nm—compared to traditional IR microscopy [Sci. Adv., *5*, eaaw6619. 10.1126/sciadv.aaw6619]. This method overcomes the diffraction limit (approximately half the wavelength) by detecting the IR response at specific wavenumbers through the force signal of the AFM cantilever [Sci. Adv., *8*, eabq4524. 10.1126/sciadv.abq4524]. Fourier transform infrared (FTIR) spectroscopy reveals a distinct peak at 1457.9 cm^{-1} for DCI molecules (**Figure R6**), which serves as a characteristic optical response for detecting their nanoscale distribution in IR mapping profiles. As shown in **Figure R7**, no distinct IR signal associated with DCI is detected on the control perovskite surface. In contrast, DCI molecules achieve nearly full coverage across both grain surfaces and boundaries in the DCI-treated perovskite film. This indicates that DCI can uniformly coat the perovskite surface under dynamic spin-coating conditions, facilitating effective and comprehensive defect passivation while regulating interfacial dipoles on the perovskite top surface.

Figure R6 FTIR spectra of the pristine perovskite film and DCI powders.

Figure R7 AFM-IR morphology and the corresponding 1457.9 cm^{-1} (red) absorption images of the control and DCI-treated perovskite films.

Furthermore, we performed time-of-flight secondary-ion mass spectrometry (ToF-SIMS) measurements to explore the diffusion properties of DCI. The characteristic ionic species and corresponding depth profiles of the DCI-treated perovskite film are presented in **Figure R8**. Evidently, cage-like diammonium cations exhibit a gradient distribution, with a strong signal at the top surface that sharply decreases with probe depth (**Figure R8a**). This suggests that cage-like diammonium cations are primarily enriched at the top surface of the perovskite layers, likely stemming from their large steric hindrance effect [Nat. Commun. 2024, 15, 882. 10.1038/s41467-024-45228-9; Adv. Mater. 2023, 36, 2307743. 10.1002/adma.202307743]. In stark contrast to cage-like diammonium cations, small chloride anions are found to migrate downward and ultimately distribute throughout the entire perovskite film, with primary enrichment at both interfaces, consistent with previous reports (**Figure**

R8b) [Nat. Energy 2024, 9, 298. 10.1038/s41560-023-01441-2; Nano-Micro Lett. 2025, 17, 141. 10.1007/s40820-024-01631-x]. As a result, a well-defined cation/anion spatial distribution of DCI within the perovskite film is established, as visually illustrated in Figure R9. This distribution not only facilitates defect passivation and interfacial dipole regulation at the perovskite top surface through multifunctional cage-like diammonium cations but also mitigates bulk and buried trap states via the formation of Pb-Cl ionic bonds. Ultimately, this enables effective suppression of ion migration and enhances device performance. [Nano-Micro Lett. 2025, 17, 141. 10.1007/s40820-024-01631-x].

Figure R8 ToF-SIMS 2D depth profiles of DCI-treated perovskite film.

Figure R9 ToF-SIMS 3D images of (a) cage-like diammonium cations ($C_6H_{13}N_2^+$) and (b) chloride anions (Cl^-).

Our response:

The top-view SEM images of the control and DCI-treated perovskite films had been re-investigated in Fig. 1a.

The AFM images of the control and DCI-treated perovskite films had been re-investigated in Fig. 1b.

The surface crystallinity of control and DCI-treated perovskite films had been re-investigated in **Fig. 1e** and **Fig. S8**.

The AFM images of the control and DCI-treated perovskite films with a scan range of $5\ \mu\text{m} \times 5\ \mu\text{m}$ had been added to **Fig. S4**.

On **page 4, line 19**, “On one hand, some columnar-like substances, featuring out-of-plane epitaxial growth, are discretely distributed at grain boundaries (GBs) of the 3D perovskite. On the other hand, many island-like clusters with a relatively low surface coverage are randomly distributed within the grain interiors (GIs) of the 3D perovskite.” had been revised to “**The perovskite surface undergoes reconstruction, forming numerous small grains uniformly distributed at the perovskite grain surfaces, accompanied by the disappearance of residual PbI_2 .**”.

On **page 4, line 21**, “These results indicate that cage-like diammonium cation, similar to fluorinated plane-type ammonium cations, can also react with PbI_2 and 3D perovskite phase to generate discrete new substances, spontaneously assembling on the top surface of the 3D perovskite.” had been revised to “**These results indicate that cage-like diammonium cation, similar to fluorinated plane-type ammonium cations, can also react with PbI_2 and 3D perovskite phase to generate new substances, spontaneously assembling on the top surface of the 3D perovskite.**”.

On **page 4, line 26**, “As demonstrated in Fig. 1b, consistent with the SEM images, many island-like substances randomly assemble within the GIs and localized bright protrusions can be clearly observed in the DCI-treated perovskite film.” had been revised to “As demonstrated in Fig. 1b **and Fig. S4**, consistent with the SEM images, **numerous uniformly assembled small grains are distinctly observed on the DCI-treated perovskite surface.**”.

The corresponding reference [Nature 2022, 611, 278. 10.1038/s41586-022-05268-x] had been cited.

On page 4, line 28, “In addition, the DCI-treated film exhibits a slightly higher root mean square (RMS) surface roughness than the control one, increasing from 13.6 nm to 18.6 nm, likely in consequence of the additional island-like substances and columnar-like protrusions, which optimizes the interfacial contact of the perovskite with electron transport layer.” had been revised to “In addition, the DCI-treated film exhibits a slightly decreased root mean square (RMS) surface roughness than the control one (Fig. 1b and Fig. S4), which optimizes the interfacial contact of the perovskite with electron transport layer²⁸.”.

The XRD patterns of DCI powder and synthesized single-crystal 2D perovskite had been added to Fig. S7.

On page 5, line 9, “Careful observation, reveals a clearly new peak at 10.2° in the DCI-treated film, indicating the formation of pure-phase quasi-2D perovskite with $n=3$ (n is the number of semiconducting $[\text{PbX}_6]^{4-}$ octahedron slabs within the cage-like diammonium spacers)²⁸.” had been revised to “Careful observation, reveals a clearly new peak at 10.2° in the DCI-treated film, indicating the formation of pure-phase quasi-2D perovskite with $n=3$ (n is the number of semiconducting $[\text{PbX}_6]^{4-}$ octahedron slabs within the cage-like diammonium spacers)²⁸ (Fig. S7).”.

AFM-IR absorption images of the control and DCI-treated perovskite films had been added to Fig. 1f.

ToF-SIMS 2D depth profiles of DCI-treated perovskite film had been added to Fig. S12.

ToF-SIMS 3D images of DCI-treated perovskite film had been added to Fig. 1g.

On page 6, line 2, “We subsequently employed atomic force microscopy-based infrared (AFM-IR) spectroscopy to reveal the nanoscale distribution and coverage of DCI molecules on perovskite surfaces (Fig. S11). As shown in Fig.

1f, no distinct IR signal associated with DCI is detected on the control perovskite surface, whereas the DCI-treated film exhibits nearly full IR signal coverage across both grain surfaces and boundaries, indicating a complete coating, conducive to uniform surface passivation and interfacial dipole regulation. To further explore the diffusion properties of DCI, time-of-flight secondary-ion mass spectrometry (ToF-SIMS) was performed (Fig. S12). The results reveal a gradient distribution of cage-like diammonium cations, with strong enrichment at the top surface that sharply decreases with probe depth, likely stemming from large steric hindrance effects and indicating an ultrathin capping layer³⁶. In stark contrast, chloride anions migrate downward and ultimately distribute throughout the entire perovskite film, with primary enrichment at both interfaces. As a result, a well-defined cation/anion spatial distribution is established (Fig. 1g), achieving synergistic defect passivation through multifunctional cage-like diammonium cations coordination and Pb-Cl ionic bonding, ultimately suppressing ion migration³⁶.” had been added.

C-AFM images and corresponding line profiles of the control and DCI-treated perovskite films had been switched to **Fig. S13** and **Fig. S14**, respectively.

On **page 7, line 2**, the corresponding discussion: “Moreover, the same current distribution between island-like clusters and their surrounding grain regions also reveals that grain regions between the clusters also feature DCI, which is conducive to forming comprehensive defect passivation on the perovskite top surface.” had been deleted.

#3-2. In Figure 1f,g, a decreased conductivity was observed due to the increased resistance because of the low-dimensional phase. According to the line profile shown in Figure 1g, the current value of about 1.5 μm distance shows a slightly high value. However, Figure 1h has shown a similar current

value at all distances. Therefore, an explanation of the line profile is necessary.

Our reply: Thanks for the reviewer's careful observation. Close inspection reveals that the yellow dashed line in Figure 1g does not actually intersect the bright spot at 1.5 μm , but merely approaches it, which could be easily misinterpreted as physical overlap. Similar findings have also been observed in previous reports [Angew. Chem. Int. Ed. Engl. 2024, 63, e202402775. 10.1002/anie.202402775; Adv. Funct. Mater. 2024, 34. 10.1002/adfm.202410737]. To avoid misleading conclusion, we conducted a statistical analysis of the currents across the entire scanning area. As shown in **Figure R10**, the DCI-treated sample reveals a lower current than the control one in the entire scanning area, which suggests that DCI helps suppress ion migration.

Figure R10 Statistical plots of the current extracted from the C-AFM images for the control and DCI-treated perovskite films.

Our response:

The statistical plots of the current extracted from the c-AFM images for the control and DCI-treated perovskite films had been added in **Fig. S14b**.

On **page 6, line 10**, “in the entire scanning region, seems to have a lower current than the control one (Fig. 1h)” had been revised to “in the entire scanning region, seems to have a lower current than the control one (**Fig. S14**)”.

On **page 10, line 6**, “which is benefited from the simultaneous passivation

of GBs and GIs by the DCI-induced quasi-2D perovskite, as reflected by the SEM and c-AFM images” had been revised to “which is benefited from the simultaneous passivation of GBs and GIs by the DCI-induced quasi-2D perovskite, as reflected by the SEM and AFM-IR images.”.

#3-3. The DCI material didn't passivate well, so according to c-AFM data, why does the author think DCI treatment reduces ion migration? The authors should explain the reason and the mechanism.

Our reply: Thanks a lot for the reviewer's insightful comments. As the reviewer pointed out, the inhibition of ion migration is generally associated with effective defect passivation. Therefore, to further elucidate the passivation effect of DCI on the perovskite film, we first investigated its spatial distribution using time-of-flight secondary-ion mass spectrometry (ToF-SIMS) and force microscopy-based infrared spectroscopy (AFM-IR) measurements. For ToF-SIMS measurements, the characteristic ionic species and corresponding depth profiles of the DCI-treated perovskite film are presented in Figure R8. Evidently, cage-like diammonium cations exhibit a gradient distribution, with a strong signal at the top surface that sharply decreases with probe depth (Figure R8a). This suggests that cage-like diammonium cations are primarily enriched at the top surface of the perovskite layers, likely stemming from their large steric hindrance [Nano-Micro Lett. 2025, 17, 141. 10.1007/s40820-024-01631-x; Nat. Commun. 2024, 15, 882. 10.1038/s41467-024-45228-9; Adv. Mater. 2023, 36, 2307743. 10.1002/adma.202307743]. In stark contrast to cage-like diammonium cations, small chloride anions are found to migrate downward and ultimately distribute throughout the entire perovskite film, with primary enrichment at both interfaces, consistent with previous reports (Figure R8b) [Nat. Energy 2024, 9, 298. 10.1038/s41560-023-01441-2; Nano-Micro Lett. 2025, 17, 141. 10.1007/s40820-024-01631-x]. As a result, a well-defined

cation/anion spatial distribution of DCI within the perovskite film is established, as visually illustrated in **Figure R9**. This distribution not only passivates defects at the perovskite top surface through multifunctional cage-like diammonium cations but also mitigates bulk and buried trap states via the formation of Pb-Cl ionic bonds, ultimately enabling effective suppression of ion migration and enhancing device performance.

Figure R8 ToF-SIMS 2D depth profiles of DCI-treated perovskite film.

Figure R9 ToF-SIMS 3D images of (a) cage-like diammonium cations ($C_6H_{13}N_2^+$) and (b) chloride anions (Cl^-).

In addition, the defective grain boundaries (GBs) and surfaces of the perovskite film are prone to serving as pathways for ion migration [*Adv. Mater.* 2023, 35, 2302552. 10.1002/adma.202302552]. Therefore, we utilized the AFM-IR system to gain deeper insights into the nanoscale distribution and coverage ratio of DCI molecules on perovskite surfaces. Fourier transform infrared (FTIR) spectroscopy reveals a distinct peak at 1457.9 cm^{-1} for DCI molecules (**Figure R6**), which serves as a characteristic optical response for detecting their nanoscale distribution in IR mapping profiles.

Figure R6 FTIR spectra of the pristine perovskite film and DCI powders.

As shown in **Figure R7**, DCI molecules are found to achieve nearly full coverage across both grain surfaces and boundaries of the perovskite film, which indicates that DCI enables effective and comprehensive defect passivation on the perovskite top surface. This, in turn, significantly inhibits ion migration.

Figure R7 AFM-IR morphology and the corresponding 1457.9 cm^{-1} (red) absorption images of the control and DCI-treated perovskite films.

To further confirm the substantial mitigation of ion migration through DCI modification, the activation energy (E_a) of ion migration for control and DCI-treated perovskite films through temperature-dependent conductivity measurements were subsequently conducted [Science 365, 473-478 (2019), 10.1126/science.aax3294; Nature 2024, 628, 93, 10.1038/s41586-024-07189-3]. The activation energy value of the control perovskite is determined to be 46 meV, whereas for the DCI-treated film, it increases to 59.3 meV (**Figure R11a**). This result indicates that DCI have a substantially boosted energy barrier for ion migration. This result can also be confirmed by the transient ionic current (TIC)

measurements (Figure R11b) as the reduced mobile ion concentration in the DCI-treated device ($2.98 \times 10^{17} \text{ cm}^{-3}$) compared with control device ($6.72 \times 10^{17} \text{ cm}^{-3}$). The concentration of mobile ions in the perovskite film was estimated from the TIC measurement based on the following formula [Energy Environ Sci 8, 1256-1260 (2015).10.1039/C4EE04064C]:

$$n = \frac{\int_{t_1}^{t_2} J dt}{eL}$$

where t is the time, J is the current density, e is the elementary charge and L is the thickness of the perovskite film. The transient ionic relaxation current was measured in the dark under an external forward bias at V_{oc} for 60 seconds. The observed current primarily originates from the redistribution of mobile ions within the perovskite layer after the removal of the applied voltage, as dictated by the dark test conditions. These results confirm the suppression of ion migration in the DCI-modified perovskite films, aligning with the findings from c-AFM experiments. Overall, the mechanism of suppressed ion migration in the DCI-treated perovskite is mainly attributed to the comprehensive defect passivation induced by the well-defined cation/anion spatial distribution of DCI within the perovskite film.

Figure R11 (a) Temperature-dependent conductivity of control and DCI-treated perovskite films. (b) Transient ionic current (TIC) for extracting mobile ion concentration within the perovskite films of the control and DCI-treated devices.

Our response:

The measurements of temperature-dependent conductivity and transient

ionic current had been added in Fig. S15 and Fig. S16, respectively.

On page 7, line 1, “This is further verified by temperature-dependent conductivity (Fig. S7) and transient ionic current (Fig. S8) measurements.” had been added.

On page 6, line 2, “We subsequently employed atomic force microscopy-based infrared (AFM-IR) spectroscopy to reveal the nanoscale distribution and coverage of DCI molecules on perovskite surfaces (Fig. S11). As shown in Fig. 1f, no distinct IR signal associated with DCI is detected on the control perovskite surface, whereas the DCI-treated film exhibits nearly full IR signal coverage across both grain surfaces and boundaries, indicating a complete coating, conducive to uniform surface passivation and interfacial dipole regulation.” had been added.

On page 28, line 5, “The current density versus voltage ($J-V$) curves were measured using an Enlitech SS-F5-3A solar simulator based on the Ushio Xe short arc lamp 500 under 1-sun conditions (100 mW cm^{-2} , AM 1.5G).” had been revised to “The current density versus voltage ($J-V$) curves, temperature-dependent conductivity and transient ionic current were measured using an Enlitech SS-F5-3A solar simulator based on the Ushio Xe short arc lamp 500 under 1-sun conditions (100 mW cm^{-2} , AM 1.5G).”.

#3-4. There are certain surface defects for both with and without DCI-treated films, therefore, a more thorough investigation related to charge carrier dynamics should be performed especially using impedance, TPC, and SCLC techniques.

Our reply: We appreciate the reviewer’s valuable suggestions. To gain deep insights into the carrier transport and recombination dynamics of the control and DCI-treated devices, we performed electrochemical impedance spectroscopy (EIS) and fitted the spectra with the corresponding equivalent

circuit models, as shown in Figure R12 and Table R1. The charge transport resistance (R_{ct}) and recombination resistance (R_{rec}) for the control device are $9.7 \times 10^5 \Omega$ and $3.4 \times 10^4 \Omega$, whereas the DCI-treated device exhibits a lower R_{ct} ($3.5 \times 10^4 \Omega$) but a higher R_{rec} ($8.6 \times 10^5 \Omega$), which indicates that carrier transport is faster and carrier recombination is more difficult in the DCI-treated device than those in the control device. Similar results are also obtained based on the transient photocurrent (TPC) and transient photovoltage (TPV) measurements (Figure R13). The TPC decay time for the target device decreases from 0.66 μs for control to 0.55 μs for DCI-treated perovskite, suggesting improved charge extraction. Moreover, compared with the control device (1.71 μs), the DCI-treated device has a prolonged photovoltage decay time (4.31 μs), indicating reduced surface recombination. To quantify the defect state density in perovskite films, space charge limited current analysis was performed using electron-only devices (Glass/ITO/ALD-SnO_x/perovskite/C₆₀/BCP/Ag), as demonstrated in Figure R14. The defect state density (N_{trap}) can be estimated using the following equation:

$$N_{trap} = \frac{2\epsilon_r \epsilon_0 V_{TFL}}{2qL^2}$$

Where q is the elementary charge, V_{TFL} is the trap filling limit voltage, ϵ_r signifies the relative dielectric constant of the perovskite, and ϵ_0 is the vacuum dielectric constant. L represents the perovskite film thickness. From the dark current-voltage curves (electron-only devices) in Figure R14, the V_{TFL} decreases from 0.19 V to 0.08 V after DCI introduction, corresponding to a decrease in N_{trap} from $2.59 \times 10^{15} \text{ cm}^{-3}$ to $1.09 \times 10^{15} \text{ cm}^{-3}$. These results suggest that DCI treatment can not only facilitate efficient charge transport but also reduce defect density, which is account for the formation of positive interfacial dipole layer induced by the spontaneous polarity of ferroelectric DCI and thus providing an additional accelerating field, and the effective passivation effect of bifunctional DCI molecule.

Figure R12 EIS spectra of the control and DCI-treated devices (inset: equivalent circuit).

Figure R13 (a) TPC and (b) TPV decay curves of the control and DCI-treated devices.

Figure R14 Dark J-V curves of electron-only devices with and without DCI treatment.

Table R1 EIS fitting parameters of the control and DCI-treated devices.

Sample	R_s (Ω)	R_{ct} ($\times 10^4 \Omega$)	R_{rec} ($\times 10^4 \Omega$)
Control	8.2	97	3.4
DCI	8.1	3.5	86

Our response:

EIS spectra of the control and DCI-treated devices had been added to **Fig.**

S41.

EIS fitting parameters of the control and DCI-treated devices had been added to **Table S8**.

TPC and TPV decay curves of the control and DCI-treated devices had been added to **Fig. S42**.

Dark J-V curves of electron-only devices with and without DCI treatment had been added to **Fig. S43**.

On page 16, line 21, “To understand the underlying mechanism for the enhanced PV performance of DCI-treated PSCs in depth, electrochemical impedance spectroscopy (EIS) were first conducted to assess charge transport and recombination dynamics. As illustrated in Fig. S41 and Table S8, the DCI-treated device exhibits a reduced charge transport resistance ($R_{ct} = 3.5 \times 10^4 \Omega$) and an increased recombination resistance ($R_{rec} = 8.6 \times 10^5 \Omega$) compared to the control device, signifying ameliorated charge transport and suppressed charge recombination. Consistent results were observed in transient photocurrent (TPC) and transient photovoltage (TPV) measurements (Fig. S42), where the DCI-treated device displays a shorter carrier transport lifetime ($\tau_{TPC} = 0.55 \mu s$) and a prolonged carrier recombination lifetime ($\tau_{TPV} = 4.31 \mu s$). Additionally, space charge limited current analysis using electron-only devices (Glass/ITO/ALD-SnO_x/perovskite/C₆₀/BCP/Ag) reveals a reduction in the trap filling limit voltage (V_{TFL}) from 0.19 V to 0.08 V, corresponding to a decrease in trap density (N_{trap}) from $2.59 \times 10^{15} \text{ cm}^{-3}$ to $1.09 \times 10^{15} \text{ cm}^{-3}$ (Fig. S43). These results evidence that DCI treatment facilitates efficient charge transport and reduces defect density, and thus enhancing V_{oc} and FF. This improvement is ascribed to the effective defect passivation and favorable interfacial dipole induced by DCI.” had been added.

On page 28, line 27, in the part of “**Device Characterization**”, “Mott-Schottky plots were measured on an electrochemical workstation (CH1660E).”

had been revised to “Mott-Schottky plots and electrical impedance spectroscopy (EIS) measurements were measured on an electrochemical workstation (CH1660E).”.

On page 28, line 29, in the part of “**Device Characterization**”, “**Transient photocurrent (TPC) and transient photovoltage (TPV) measurements were conducted on a TranPVC W900 system (Oriental Spectra Technology Co., Ltd., Guangzhou) using a 517 nm pulsed laser excitation source (NanoQ series).**” had been added.

#3-5. In Fig. 2c and Fig. S8, the authors showed the interaction of DCI with PbI_2 and FA ions, respectively. However, since perovskite is a crystal composed of various cations and anions, it should be explained the interaction between DCI to perovskite crystals. For example, it is necessary to explain what is the main interaction in the perovskite structure either Pb-Cl bonding or hydrogen bonding with FA ions.

Our reply: Thanks for the reviewer’s helpful suggestion. To gain a deeper insight into the interfacial interaction of DCI with perovskite, we conducted density functional theory (DFT) calculations to determine the binding energies for DCI adsorption. For simplicity, the DCI is categorized into two functional groups (R_3N and R_3NHCl), and four potential adsorption structures on the PbI_2 -terminated and formamidinium iodide (FAI)-terminated (100) perovskite surfaces are given. The corresponding adsorption energies (E_{ad}) for different configurations are calculated and presented in **Figure R15**. Regarding the FAI termination of perovskite (**Figure R15a**), the E_{ad} values of $\text{R}_3\text{N}/\text{FAI}$ and $\text{R}_3\text{NHCl}/\text{FAI}$ are -0.42 and -0.50 eV, respectively, indicating a comparable binding energy. On the FAI termination, both R_3N and R_3NHCl functional groups tend to bind with perovskite surface via weak hydrogen bonding, e.g. $\text{R}_3\text{N}\cdots\text{FA}^+$, $\text{Cl}\cdots\text{FA}^+$ and $\text{R}_3\text{NH}^+\cdots\text{I}$. In contrast, for the PbI_2 termination (**Figure R15b**), the

E_{ad} values of R_3NHCl/PbI_2 and R_3N/PbI_2 are -0.90 eV and -0.78 eV, respectively, which is significantly lower than that of R_3NHCl/FAI and R_3N/FAI , indicating a stronger binding configuration on the PbI_2 termination, and thus the primary interaction of DCI with PbI_2 . Additionally, it is noteworthy that, for R_3NHCl/PbI_2 adsorption configuration, the strong interaction only occurs between Cl and Pb via Pb-Cl ionic bonding. As for R_3NH^+ group, no interaction with Pb but weak hydrogen bonding with I occurs. Accordingly, it is indicated that DCI is energetically more favorable to interact with perovskite via strong coordination bonding of R_3N group with PbI_2 and Pb-Cl ionic bonding. From the ToF-SIMS results with cage-like diammonium cations concentrating at the perovskite surface and small chloride anions migrating downward to accumulate at the buried interface, we can infer that, with this anchoring effect, oriented molecular dipoles directed from perovskite to C_{60} can be established by cage-like diammonium cations at the perovskite/ C_{60} interface, thereby creating a dipole layer over the perovskite surface, which is essential to regulate interfacial energy level difference, thereby promoting device performance. While for chloride anions, the formation of strong Pb-Cl ionic bonding is conducive to passivating the bulk and buried defects, thereby enhancing the device performance and stability.

Figure R15 (a) DFT modeling of the interaction of the DCI with FAI termination. (b) DFT modeling of the interaction of the DCI with PbI_2 termination.

Our response:

DFT modeling of the interaction of the DCI with FAI-terminated and PbI_2 -terminated (100) perovskite surface had been added to **Fig. S17**.

^1H NMR spectra of DCI and $\text{DCI}+\text{PbI}_2$ had been revised to **Fig. S18**.

Pb 4f and I 3f XPS spectra had been revised to **Fig. S21**.

On **page 7, line 13**, “To elucidate the interaction between perovskite and DCI, high-resolution X-ray photoelectron spectroscopy (XPS) characterization was performed. In the C 1s XPS core (Fig. S6), the binding energies centered at ~284.8, 286.6, and 288.4 eV are assigned to C-C, N-C-C-N, and N-C=N, respectively. Turning to the N 1s XPS core (Fig. S7), an obvious increase in the higher binding energy peak at 402.7 eV is detected, likely corresponding to $\text{R}_3\text{-N}$ bond. The obvious N-C-C-N and $\text{R}_3\text{-N}$ characteristic peaks for DCI-treated film collectively indicates the existence of cage-like diammonium cation on the perovskite surface and its dominant interaction with the perovskite. Further, we also notice that with the incorporation of DCI, the characteristic peaks in the Pb 4f XPS spectra shift toward the higher binding energy (from 143.37 and 138.51 eV for $4f_{5/2}$ and $4f_{7/2}$ to 143.55 and 138.66 eV, respectively), which may originate from the bond formation between the uncoordinated Pb^{2+} in perovskite or PbI_2 and DCI (Fig. 2a and Table S1). Similar upward shift of the corresponding characteristic peaks in the I 4d XPS spectra is also observed (Fig. 2b and Table S2), in which I $3d_{3/2}$ (630.71 eV) and I $3d_{5/2}$ (619.26 eV) for the control film slightly shift toward higher binding energy position of 630.77 and 619.31 eV, respectively, after DCI post-treatment, which could be ascribed to the incorporation of Cl^{38} .”

Subsequently, liquid-state ^1H nuclear magnetic resonance (^1H NMR) was carried out to further explore the interaction mechanism of DCI with perovskite, as illustrated in Fig. 2c. The chemical shift at 11.04 ppm corresponds to the $\text{R}_3\text{-NH}^+$ in DCI. Upon addition of PbI_2 to DCI, ^1H signal of the $\text{R}_3\text{-NH}^+$ group shows

an obviously upfield shift from 11.04 ppm to 9.45 ppm, which is attributed to the shielding effect induced by anion exchange from Pb-I bond to Pb-Cl bond and the direct ionic binding of Cl^- with the undercoordinated Pb^{2+39} . Similar upfield shift trends of ^1H signal for the $\text{R}_3\text{-NH}^+$ group in DCI was also found when mixing DCI with FAI, as depicted in Fig. S8. ^1H signal of the $\text{R}_3\text{-NH}^+$ group in DCI moves from 11.04 ppm to 9.68 ppm. In addition, the resonance signal (8.79 ppm) of protonated ammonium in FAI splits into two at 8.965 ppm and 8.629 ppm, respectively. These chemical shifts indicate the strong interaction of DCI with FAI by the formation of hydrogen bonds⁴⁰. Overall, the aforementioned results from c-AFM, XPS and NMR measurements collectively confirm the strong chemical interaction of DCI with the perovskite surface via its multi-functional passivation groups, which is expected to reduce the surface trap states, suppressing nonradiative recombination, thus improving photovoltaic (PV) performance.” had been revised to “To elucidate the interaction between perovskite and DCI, we conducted density functional theory (DFT) calculations to determine the binding energies of DCI adsorption. For simplicity, the DCI was categorized into two functional groups: R_3N and R_3NHCl . Four potential adsorption configurations on the PbI_2 -terminated and FAI-terminated (100) perovskite surfaces were considered, with the corresponding adsorption energies (E_{ad}) presented in Fig. S17. Regarding the FAI termination (Fig. S17a), the E_{ad} values of $\text{R}_3\text{N}/\text{FAI}$ and $\text{R}_3\text{NHCl}/\text{FAI}$ are -0.42 and -0.50 eV, respectively, indicating comparable and relatively weak binding. On this surface, the interactions are mainly through weak hydrogen bonding (e.g., $\text{R}_3\text{N}\cdots\text{FA}^+$, $\text{Cl}\cdots\text{FA}^+$ and $\text{R}_3\text{NH}^+\cdots\text{I}$), as supported by liquid-state ^1H NMR spectra (Fig. S18a). In contrast, for the PbI_2 termination (Fig. S18b), $\text{R}_3\text{NHCl}/\text{PbI}_2$ and $\text{R}_3\text{N}/\text{PbI}_2$ exhibit much stronger binding energies of -0.90 eV and -0.78 eV, respectively, indicating a clear preference for adsorption on PbI_2 termination and highlighting the primary interaction of DCI with PbI_2 . Notably, in the

R_3NHCl/PbI_2 configuration, strong Pb-Cl ionic bonding is observed, whereas the R_3NH^+ group exhibits only weak hydrogen bonds with I, without direct interaction with Pb. These results reveal that DCI predominantly interacts with perovskite via R_3N -Pb coordination and Pb-Cl ionic bonding, as further confirmed by high-resolution X-ray photoelectron spectroscopy (Fig. S19-S21 and Table S1,2). Furthermore, ToF-SIMS results reveal that cage-like diammonium cations accumulate at the perovskite surface, while chloride anions migrate downward (Fig. 1g). In this case, the spontaneously polarized cage-like cations directionally assemble at the perovskite/ C_{60} interface, forming a robust interfacial dipole layer that modulates band bending and alleviates energy level misalignment⁴⁰. Concurrently, strong Pb-Cl ionic bonding effectively passivates bulk and interfacial defects, contributing to improved performance and stability⁴¹.”.

In the part of “**DFT calculations**”, “**Calculations of adsorption energy**”
The adsorption structures were investigated using periodic plane-wave DFT calculations performed with the Vienna Ab Initio Simulation Package (VASP). The Perdew-Burke-Ernzerhof (PBE) exchange-correlation functional was employed with a kinetic energy cutoff of 450 eV. The Brillouin zone integration was sampled using a $2 \times 2 \times 1$ Monkhorst-Pack k-point mesh. During structural optimization, the convergence criterion for atomic forces was set to 0.02 eV/Å. To preserve the solid-phase characteristics, the cell parameters were kept fixed while allowing only the atomic positions to relax.

The adsorption energy (E_{ad}) was calculated using the following equation:

$$E_{ad} = E_{\text{adsorbate/slab}} - E_{\text{adsorbate}} - E_{\text{slab}}$$

where $E_{\text{adsorbate/slab}}$, $E_{\text{adsorbate}}$ and E_{slab} represent the adsorbate-adsorbed kaolinite substrate, the adsorbate, and the total energies of the kaolinite substrate, respectively. The 3D iso-surfaces and 1D planar-averaged results of charge density differences (CDD) between the substrate and the adsorbate

were obtained by subtracting the electron densities of the isolated substrate and adsorbate from that of the combined system. These calculations were performed using VASPKIT, and the charge density difference visualizations were rendered using the VESTA program.” had been added.

#3-6. The authors should provide the mechanism of passivation for untreated and DCI-treated films.

Our reply: Thanks for the reviewer’s suggestion. To further elucidate the passivation mechanism for untreated and DCI-treated films, we used density functional theory (DFT) calculations to compare the formation energies of selected native defects on the perovskite surface. Particularly taken into consideration were Pb and I-involving point defects, Pb vacancy (V_{Pb}), I vacancy (V_I), and Pb-I antisite (Pb_I and I_{Pb} , corresponding to I site substitution by Pb and Pb site substitution by I, respectively) (Figure R16) because the band edges of perovskite were reported to be composed of Pb and I orbitals [Science 2019, 366, 1509. 10.1126/science.aay9698; Energy Environ. Sci. 2024, 17, 9601. 10.1039/d4ee02803a].

Figure R16 Front-view and side-view of various types of defects, including V_I , Pb_I , V_{Pb} and I_{Pb} defects.

Among these prevalent defects, V_I and Pb_I are positively charged while V_{Pb}

and I_{Pb} carry negative charges [Nature Reviews Chemistry 2023, 7, 632. 10.1038/s41570-023-00510-0]. Meanwhile, V_I and V_{Pb} are classified as shallow-level defects, whereas I_{Pb} and Pb_I are considered deep-level defects [Appl. Phys. Lett. 2014, 104, 063903. 10.1063/1.4864778]. Evidently, both positively charged defects and negatively charged defects can be effectively passivated by DCI decoration, owing to its multifunctional groups (R_3N and R_3NH^+). Similarly, both deep and shallow level defects can be simultaneously passivated, as shown in Figure R17.

Figure R17 Top-view of various types of defects passivated by DCI.

The defect formation energies of V_I , Pb_I , V_{Pb} and I_{Pb} on the surface are calculated to be 2.02, 1.86, 1.90 and 1.23 eV (Figure R18a), respectively. It is evident that deep-level defects, particularly the I_{Pb} antisite defect, exhibit lower formation energies compared to shallow-level defects. Therefore, both I_{Pb} and Pb_I antisite defects are more likely to form and predominate on the surface, making them the primary non-radiative recombination centers. When DCI is introduced onto the perovskite surface which has I_{Pb} antisite defects, the deep level defect states are effectively inhibited (Figure R18b and c). This could be attributed to the formation of chemical bonds between negative halide traps and positive R_3NH^+ group. A similar passivation effect is also found on the Pb_I antisite-dominated surface (Figure R19).

Figure R18 (a) The defect formation energies of various types of defects on perovskite surface (the inset represents the top view). (b,c) The partial density of states (PDOSs) of (b) the defective perovskite (I_{Pb} antisite) and (c) passivated perovskite by DCI.

Figure R19 The partial density of states (PDOSs) of (a) the defective perovskite (Pb_I antisite) and (b) passivated perovskite by DCI.

Furthermore, thermal admittance spectroscopy analysis was undertaken to distinguish the trap states in perovskite films (Figure R20). For the DCI-based device, the reduction of trap states in the band range from 0.55 to 0.65 eV indicates that the deep trap states located in perovskite films are effectively passivated by the incorporation of DCI, which is consistent with the higher PLQY value and longer carrier lifetime for perovskites with DCI.

Figure R20 tDOS obtained from thermal admittance spectroscopy for devices without and with DCI modification.

Our response:

Front-view and side-view of various types of defects, including V_I , Pb_I , V_{Pb} and I_{Pb} defects had been added to **Fig. S24**.

The defect formation energies of various types of defects on perovskite surface had been added to **Fig. 2g**.

The partial density of states (PDOSs) of the defective perovskite (I_{Pb} antisite) and passivated perovskite by DCI had been added to **Fig. 2h** and **Fig. 2i**, respectively.

The partial density of states (PDOSs) of the defective perovskite (Pb_I antisite) and passivated perovskite by DCI had been added to **Fig. S25**.

tDOS obtained from thermal admittance spectroscopy for devices without and with DCI modification had been added to **Fig. S26**.

On page 10, line 11, “To gain deeper insights into the passivation mechanisms of DCI on the perovskite surface, we conducted DFT calculations considering four prevalent surface defects: I vacancy (V_I), Pb-I antisite (Pb_I), Pb vacancy (V_{Pb}), and I-Pb antisite (I_{Pb}) (Fig. S24)⁴³. Computational analysis reveals that deep-level defects, particularly I_{Pb} , exhibit lower formation energies than shallow-level defects (Fig. 2g). Consequently, both negatively charged I_{Pb} and positively charged Pb_I antisite defects form more readily and predominate on the surface, making them the primary non-radiative recombination centers. Upon DCI adsorption, deep-level states associated with I_{Pb} defects are effectively inhibited (Fig. 2h and i), likely due to chemical interactions between negative halide traps and the R_3NH^+ group. A similar passivation effect is also observed on the Pb_I -dominated surface (Fig. S25). Thermal admittance spectroscopy analysis further reveals the reduced deep trap states following DCI incorporation (Fig. S26).” had been added.

On page 29, line 3, “Trap density of state (tDOS) curves were obtained using a 1240A Impedance Analyzer.” had been added.

In the part of “**DFT calculations**”, “**Calculations of defect formation energy, partial density of state, surface work function and charge density difference**” All first-principles calculations based on DFT were performed using VASP. The electron-ion interactions were described by projector augmented-wave (PAW) pseudopotentials, and exchange-correlation effects were treated using the PBE functional within the generalized gradient approximation (GGA). The slab model was constructed from a cubic-phase primitive cell with a 2×2 surface supercell comprising eight atomic layers. The bottom four layers were fixed, while the remaining atoms were fully relaxed. A vacuum layer of at least 20 Å was introduced along the z-direction to prevent periodic interactions. Calculations employed a plane-wave energy cutoff of 500 eV and a k-point mesh of 1×1×1 for slab models (2×2×1 for DOS calculations). Structural optimization was performed with a force convergence threshold of 0.05 eV/Å and an energy convergence criterion of 1×10⁻⁵ eV. Van der Waals interactions were accounted for using the Grimme D3-BJ correction. The defect formation energy is calculated as:

$$E_F = E_{\text{slab+defect}} - E_{\text{slab}} \pm \sum n_i \mu_i$$

Where $E_{\text{slab+defect}}$ is the energy of the slab with defects, E_{slab} is the energy of defect-free slab, n_i is the number of atoms added or removed, μ_i is the chemical potential of the corresponding element.” had been added.

#3-7. The authors performed PFM analysis to evaluate the ferroelectric properties, which is shown in Figure 3g. It was confirmed that the phase difference was 180 degrees due to the incorporation of DCI. However, to understand ferroelectric properties, further explanation is needed as to why there is a difference in intensity for each phase. For example, whether the

intensity difference for each phase is a meaningful change or a negligible difference.

Our response: We sincerely thank the reviewer for the valuable comment. In piezoresponse force microscopy (PFM), ferroelectricity is typically assessed by two key indicators:

The phase contrast. A 180° shift between oppositely poled domains indicates polarization reversal [Angew. Chem. Int. Ed. Engl. 2023, 62, e202217526. 10.1002/anie.202217526; Adv. Mater. 2022, 34, 2110482. 10.1002/adma.202110482; Nat. Commun. 2025, 16, 1042.10.1038/s41467-025-56409-5].

The amplitude response. It provides information on the piezoelectric activity of the material [Nat. Commun. 2024, 15, 8753. 10.1038/s41467-024-53121-8; Nano Energy 2023, 107. 10.1016/j.nanoen.2022.108114; Adv. Mater. 2022, 34, 2110482. 10.1002/adma.202110482].

While the intensity difference in the phase image (i.e., the variation in brightness across domains) can sometimes be observed, this feature is not a reliable or standard criterion for confirming ferroelectricity. It may result from a variety of factors such as tip-sample contact condition [Angew. Chem. Int. Ed. Engl. 2023, 62, e202217526. 10.1002/anie.202217526; Adv. Mater. 2022, 34, 2110482. 10.1002/adma.202110482], local topography, or even scanning artifacts. Therefore, these contrast differences are generally considered negligible and non-intrinsic to the ferroelectric properties.

As shown in Fig. 3g, the clear 180° phase shift upon domain switching provides direct and convincing evidence of ferroelectric behavior induced by DCI incorporation. The amplitude image (Fig. S33) further supports this by showing an increased amplitude contrast, confirming the strong ferroelectric response of DCI [Nano Energy 2023, 107. 10.1016/j.nanoen.2022.108114; Adv. Mater. 2022, 34, 2110482. 10.1002/adma.202110482].

Our response:

The external biases for ferroelectric polarization had been marked in Fig. 3g and Fig. 3h.

#3-8. Figure S18, what factors are responsible for different EQE spectra, with and without DCI-treated films?

Our reply: We thank the reviewer for the insightful comment. It is widely recognized that an enhanced photocurrent response is typically attributed to a reduction in trap states at the perovskite surface and bulk, as well as improved optical absorption [ACS Appl Mater Interfaces 2018, 10, 22074. 10.1021/acsami.8b04439; ACS Energy Lett. 2024, 9, 1682. 10.1021/acseenergylett.4c00306]. In DCI-treated perovskites, defects at the top surface are effectively passivated due to the enrichment of multifunctional cage-like diammonium cations. Additionally, chloride anions spontaneously migrate downward into the perovskite bulk and buried interface, further reducing defects by forming Pb–Cl bonds. As a result, an enhanced EQE response across the entire wavelength range is expected. However, a slight decrease in EQE between 400 and 480 nm is observed in the DCI-treated device, likely due to its comparatively higher reflectance (Figure R21). Similar results have also been reported before [Nature 2023, 618, 80. 10.1038/s41586-023-05992-y; ACS Energy Lett. 2024, 9, 5310. 10.1021/acseenergylett.4c02098; ACS Energy Lett. 2020, 5, 271. 10.1021/acsenergylett.9b02604; Energy Environ. Sci. 2023, 16, 2080. 10.1039/D2EE04087E].

Figure R21 Reflectance spectra of the control and DCI-treated devices.

Our response:

The reflectance spectra of the control and DCI-treated perovskite solar cells had been added to Fig. S38.

The corresponding reference [ACS Energy Lett. 2024, 9, 1682. 10.1021/acsenerylett.4c00306] had been cited.

On page 16, line 10, “The slight EQE reduction between 400-480 nm in the DCI-treated device likely stems from its comparatively higher reflectance (Fig. S38)⁵².” had been added.

#3-9. The surface morphology for with and without DCI-treated films is almost similar, though DCI-treated films are rougher, so why there is a steady decrease in PCE for the control device and very minor loss for the DCI -treated device? Also, the DCI-treated perovskite films didn't show any adequate passivation.

Our reply: We thank the reviewer for pointing out this. Similar post-treatment strategies, such as employing CF₃-PEA, CF₃-PA and PDI as surface passivating layers or utilizing sequential interface engineering (EDA1₂ + PEACl), have also demonstrated that surface passivation does not alter film morphology but significantly enhances device stability [Nature 2022, 603, 73. 10.1038/s41586-021-04372-8; Nat. Energy 2023, 8, 610. 10.1038/s41560-023-01250-7; Energy Environ. Sci. 2023, 16, 5029. 10.1039/D3EE00869J; Adv. Mater. 2023, 36, 2308370. 10.1002/adma.202308370]. This indicates that

device stability is not directly determined by film morphology but rather by effective defect passivation and the suppression of ion migration [Adv. Energy Mater. 2024, 14. 10.1002/aenm.202401414; Adv. Mater. 2023, 35, 2302552. 10.1002/adma.202302552; Nat. Rev. Mater. 2023, 8, 569. 10.1038/s41578-023-00582-w], even residual strain [Natl. Sci. Rev. 2021, 8, nwab047. 10.1093/nsr/nwab047]. Regarding defect passivation, our ToF-SIMS measurements confirm that defects at the top surface are effectively passivated due to the enrichment of multifunctional cage-like diammonium cations at the perovskite surface (Figure R8), in consistent with previous reports [Nano-Micro Lett. 2025, 17, 141. 10.1007/s40820-024-01631-x; Nat. Commun. 2024, 15, 882. 10.1038/s41467-024-45228-9].

Figure R8 (a,b) ToF-SIMS 2D depth profiles of the DCI-treated perovskite film. (c) ToF-SIMS 3D reconstructed image of cage-like diammonium cations. (d) ToF-SIMS 3D reconstructed image of chloride anions.

Additionally, to gain further insights into the nanoscale distribution and coverage of DCI on the perovskite surface, we further employed atomic force microscopy coupled with infrared spectroscopy (AFM-IR), as depicted in Figure R7. DCI molecules are found to achieve nearly full coverage across both grain surfaces and boundaries of the perovskite film, which indicates that DCI enables effective and comprehensive defect passivation on the perovskite top

surface.

Figure R7 AFM-IR morphology and the corresponding 1457.9 cm^{-1} (red) absorption images of the control and DCI-treated perovskite films.

Additionally, chloride anions are found to spontaneously migrate downward into the perovskite bulk and buried interface, which can further reduce bulk and buried defects by forming strong Pb–Cl chemical bonding (Figure R8b and d) [Nano-Micro Lett. 2025, 17, 141. 10.1007/s40820-024-01631-x; Adv. Mater. 2024, 36, 2313524. 10.1002/adma.202313524; Adv. Mater. 2023, 35, 2301624. 10.1002/adma.202301624]. The Cl incorporation with smaller ionic radius than its isoelectronic element I makes it suitable not only to reduce the strain but also to effectively prevent the formation of vacancy defects [Energy Environ. Sci. 2023, 16, 5029. 10.1039/D3EE00869J], even suppress ion migration [Energy Environ. Sci. 2023, 16, 2080. 10.1039/D2EE04087E].

Overall, benefiting from the comprehensive defect passivation of DCI in the perovskite bulk and at dual-interfaces, trap states are substantially reduced, ultimately resulting in a 90 mV increase in V_{oc} . This suppression minimizes the interaction between the perovskite surface and the ambient environment, preventing the formation of deep traps and reducing ion migration, thereby enhancing the long-term stability of PSCs. [Energy Environ. Sci. 2023, 16, 5029. 10.1039/D3EE00869J; Nat. Energy 2019, 4, 408. 10.1038/s41560-019-0382-6].

Moreover, temperature-dependent conductivity measurements were

conducted to further investigate ion migration characteristics in both control and DCI-treated perovskite films by determining their activation energy for ion migration [Nature 2024, 628, 93. 10.1038/s41586-024-07189-3]. The activation energy value of the control perovskite is determined to be 46 meV. whereas for the DCI-treated film, it increases to 59.3 meV (Figure R22). This result indicates that DCI have a substantially boosted energy barrier for ion migration, which may also contribute to improved perovskite crystal stability.

Figure R22 Temperature-dependent conductivity of control and DCI-treated perovskite films.

Our response:

Temperature-dependent conductivity of control and DCI-treated perovskite films had been added to **Fig. S15**.

On page 20, line 25, “The improved stability of DCI-treated tandems under light soaking is primarily attributed to the synergistic effects of DCI for comprehensively passivating charge trap states (Fig. 1g) and substantially suppressing ion migration (Fig. S15 and S16).” had been added.

#3-10. The authors explain that charge recombination occurs at the interface between perovskite and charge transport layers, and it is particularly emphasized between perovskite/ C_{60} . Therefore, to understand carrier dynamics at the perovskite/ C_{60} , the mechanism of charge recombination at the perovskite/ C_{60} interface needs to be explained.

Our reply: Thanks for the reviewer’s suggestion. It is widely recognized

that the performance of inverted perovskite solar cells is predominantly limited by the non-radiative recombination at the perovskite/C₆₀ interface [Nat. Energy 2018, 3, 847. 10.1038/s41560-018-0219-8; Energy Environ. Sci. 2019, 12, 2778. 10.1039/C9EE02020A; ACS Energy Lett. 2023, 2532. 10.1021/acseenergylett.3c00697]. Specifically, the first monolayers of C₆₀ molecules can act as deep trap states, arising either from charge transfer states or density of states (DOS) broadening at the interface. This effect pins the LUMO of C₆₀ below the conduction band of the perovskite when in direct contact with the perovskite, leading to severe across-interface recombination losses [Adv. Energy Mater. 2022, 12, 2201109. 10.1002/aenm.202201109; Adv. Energy Mater. 2022, 12, 2103567. 10.1002/aenm.202103567], as illustrated in Figure R23.

Figure R23 Recombination mechanisms and causes of C₆₀ induced nonradiative losses showing increased nonradiative recombination either via a charge transfer state or low lying C₆₀ states at the interface due to DOS broadening [Adv. Energy Mater. 2022, 12, 2103567. 10.1002/aenm.202103567].

To study how the passivating DCI molecules affect interfacial charge transport and recombination, we explored the interaction between the C₆₀ layer and the perovskite surface and charge extraction at the perovskite/C₆₀ interface. Figure R24 demonstrates the calculated charge density difference (yellow,

accumulation; blue, depletion) along with the corresponding plane-averaged profiles for C₆₀ ETL anchoring onto both the control and DCI passivated (100) perovskite surfaces. As shown in **Figure R24**, capping DCI passivators induce notable charge redistribution. Charge accumulation within C₆₀ molecules is significantly repressed when passivated by DCI molecules, thus less charge recombination. The adsorption energy (E_a) of C₆₀ on the control (bare) and DCI passivated (100) perovskite surfaces are -0.02 and -0.14 eV, respectively, indicating the stronger interaction between C₆₀ and DCI passivated perovskite surface, which is conducive to reducing device charge transfer resistance and accelerating interfacial charge transfer [Nat. Energy 2023, 8, 610. 10.1038/s41560-023-01250-7; ACS Appl. Mater. Interfaces 2018, 10, 32471. 10.1021/acsami.8b11459]. The corresponding plane-averaged profiles further verify the significantly stronger charge transfer between C₆₀ and the DCI-passivated perovskite compared to that between C₆₀ and the bare perovskite.

Figure R24 Charge density difference (yellow, accumulation; blue, depletion) along with the corresponding plane-averaged profiles for C₆₀ ETL anchoring onto both the control and DCI passivated (100) perovskite surfaces. The calculated adsorption energy (E_a) is also shown.

Our response:

On page 2, line 13, “The first monolayers of C₆₀ can introduce deep trap states at the interface through energy-level pinning and band misalignment, exacerbating across-interface recombination losses^{6,8}.” had been added.

The corresponding reference [Nat. Energy 2023, 8, 610. 10.1038/s41560-

023-01250-7] had been cited.

On page 10, line 26, “As shown in Fig. S11 and Table S3, the DCI sample exhibits a much longer lifetime of slow decay (~43.0 ns) and shorter lifetime of fast decay (~10.5 ns) in comparison with the control sample (~22.9 and 10.8 ns, respectively) suggesting that there is reduced across-interface non-radiative recombination loss and enhanced charge extraction at the perovskite/DCI/C₆₀ interface.” had been revised to “As shown in Fig. S27 and Table S3, the DCI sample exhibits a much longer lifetime of slow decay (~43.0 ns) and shorter lifetime of fast decay (~10.5 ns) in comparison with the control sample (~22.9 and 10.8 ns, respectively), suggesting reduced across-interface non-radiative recombination and enhanced charge extraction at the perovskite/DCI/C₆₀ interface, further supported by DFT calculations. Fig. S28 illustrates that charge accumulation within C₆₀ molecules is significantly repressed by DCI molecules. Additionally, the adsorption energy of C₆₀ on DCI-passivated (100) perovskite surfaces (-0.14 eV) is substantially lower than on bare perovskite surfaces (-0.02 eV), indicating stronger interaction, reduced charge transfer resistance, and enhanced interfacial charge transfer⁵⁷”.

In the part of “**DFT calculations**”, “**Calculations of adsorption energy**” The adsorption structures were investigated using periodic plane-wave DFT calculations performed with the Vienna Ab Initio Simulation Package (VASP). The Perdew-Burke-Ernzerhof (PBE) exchange-correlation functional was employed with a kinetic energy cutoff of 450 eV. The Brillouin zone integration was sampled using a 2×2×1 Monkhorst-Pack k-point mesh. During structural optimization, the convergence criterion for atomic forces was set to 0.02 eV/Å. To preserve the solid-phase characteristics, the cell parameters were kept fixed while allowing only the atomic positions to relax.

The adsorption energy (E_{ad}) was calculated using the following equation:

$$E_{ad} = E_{adsorbate/slab} - E_{adsorbate} - E_{slab}$$

where $E_{\text{adsorbate/slab}}$, $E_{\text{adsorbate}}$ and E_{slab} represent the adsorbate-adsorbed kaolinite substrate, the adsorbate, and the total energies of the kaolinite substrate, respectively. The 3D iso-surfaces and 1D planar-averaged results of charge density differences (CDD) between the substrate and the adsorbate were obtained by subtracting the electron densities of the isolated substrate and adsorbate from that of the combined system. These calculations were performed using VASPKIT, and the charge density difference visualizations were rendered using the VESTA program.” had been added.

Reviewer #4:

This study demonstrates a multifunctional cage-like diammonium chloride molecule to reduce the surface defects and suppress non-radiative recombination. By using this molecule, the PCE of wide bandgap perovskite solar cells reach 22.6% and that of 2T perovskite/silicon tandem solar cells reaches 31.1%. The results are interesting but there are some issues that need to be addressed.

Our response:

We thank the reviewer's valuable comments. In the following, we address the reviewer's comments with point-by-point responses.

#4-1. The standard deviation of the devices PCE measurements for each condition should be provided along with the number of samples measured so that the statistical reproducibility of the device can be more clearly demonstrated.

Our reply: Thanks for your good suggestion. The average and standard deviation of device parameters including V_{oc} , FF, J_{sc} and PCE for each condition has been provided along with the number of samples. For single-junction perovskite solar cells without poling, the corresponding photovoltage (PV) parameters are listed in Table R1. Obviously, DCI-treated device shows a remarkable improvement in all PV parameters compared to the control device. Notably, the average and standard deviation of device parameters for single-junction perovskite solar cells without poling is calculated from 22 individual devices.

Table R1 The average and standard deviation of device parameters extracted from reverse-scan J-V curves for single-junction perovskite solar cells without poling.

Sample	V_{oc} [V]	J_{sc} [mA cm ⁻²]	FF [%]	PCE [%]
Control	1.137±0.017	20.83±0.20	77.80±2.12	18.42±0.50
DCI	1.214±0.004	21.01±0.27	82.70±0.53	20.99±0.38

For single-junction perovskite solar cells after poling, the corresponding PV parameters are demonstrated in **Table R2**. The average and standard deviation of device parameters is calculated from 10 individual devices. Upon polarization, i.e., keeping the device under illumination at -0.5 V before the J-V scan, the DCI-treated device also illustrates a remarkable improvement in all PV parameters compared to the biased control device. The average PCE of the polarized DCI-treated device reaches 21.9%, far exceeding that of the polarized control device.

Table R2 The average and standard deviation of device parameters for single-junction perovskite solar cells after poling.

Sample	V_{oc} [V]	J_{sc} [mA cm ⁻²]	FF [%]	PCE [%]
Control	1.141±0.003	21.04±0.37	78.26±0.57	18.76±0.35
DCI	1.224±0.009	21.53±0.35	83.32±0.63	21.94±0.44

Our response:

The average and standard deviation of device parameters for single-junction perovskite solar cells without poling had been added in **Table S7**.

The statistical distributions of PV parameters for single-junction perovskite solar cells after poling had been added to **Table S9**.

On **page 16, line 12**, “The high reproducibility of DCI-treated PSCs is also confirmed, as shown in Fig. S19.” had been revised to “The high reproducibility

of DCI-treated PSCs is also confirmed, as shown in Fig. S39 and Table S7.”.

On page 17, line 9, “Since it has been proved that DCI-treated perovskite exhibits remarkable ferroelectric response, thus the effects of ferroelectric interface on device performance were additionally studied by polarizing the devices with an external poling bias.” had been revised to “Since it has been proved that DCI-treated perovskite exhibits remarkable ferroelectric response, thus the effects of ferroelectric interface on device performance were additionally studied by polarizing the devices with an external poling bias (Table S9).”.

#4-2. Instead of only the stability results under ISOS-L-1, more stability tests under other conditions, such as damp heat or light soaking, should be provided, and comparisons with results reported in other literature should also be included.

Our reply: Thanks for your constructive suggestions. To enable more comprehensive stability evaluation, on the one hand, the stability testing duration under ISOS-L-1 protocol was extended to 1,000 hours (Figure R1). On the other hand, light-soaking stability was additionally provided (Figure R2). As presented in Figure R1, the control tandem device suffers from a rapid degradation to ~43.4% of the initial PCE after 640 hours of MPP tracking. In contrast, the DCI-treated tandem device still maintains ~85.4% of its original PCE after 1020 hours of MPP tracking, demonstrating stability comparable to the reported state-of-the-art perovskite/silicon TSCs under similar test conditions (Table R3).

Figure R1 Continuous MPP tracking of the unencapsulated control and DCI-treated perovskite/silicon TSCs under 1-sun illumination (100 mW cm^{-2}) with a xenon lamp without UV filter in ambient air ($30 \pm 5 \text{ }^\circ\text{C}$, 40–60% RH).

Table R3 Summary of MPP stability for the state-of-the-art monolithic perovskite/silicon TSCs reported in literatures.

Structure	Environment	Encapsulation	Light condition	PCE retention	Ref.
	In ambient				
p-i-n	air; 25–35%RH; 30 °C	no	1-sun illumination	$T_{92}=100 \text{ h}$	41
p-i-n	air; 30–40%RH; 25 °C	no	LED with wavelengths of 470 and 940 nm	$T_{95.5}=300 \text{ h}$	42
p-i-n	In N_2 ; 25±1 °C	no	1-sun illumination	$T_{90}=100 \text{ h}$	43
p-i-n	In ambient air; 20%RH; 25 °C	no	1-sun illumination	$T_{98}=300 \text{ h}$	44
p-i-n	In N_2 ; 25±1 °C	no	1-sun illumination	$T_{118}=100 \text{ h}$	45
p-i-n	In N_2 ; 25 °C	no	White LED light source (940 nm LED light for extra compensation)	$T_{98}=200 \text{ h}$	46
p-i-n	In ambient air; 40– 70%RH; 20–35 °C	no	Xenon lamp (100 mW cm^{-2})	$T_{93.6}=450 \text{ h}$	47
p-i-n	In ambient air; 25–	no	Xenon lamp (100 mW	$T_{86.6}=306 \text{ h}$	48

	75%RH, most often ≈60%RH; 20–35 °C		cm ⁻²)			
p-i-n	In N ₂ ; 25 °C	no	1-sun illuminations	T ₈₀ =1200 h	49	
p-i-n	In ambient air; 30– 40%RH; 25°C	no	1-sun illuminations	T ₁₀₀ =700 h	50	
p-i-n	In ambient air; 25%RH; 25°C	no	1-sun illuminations	T ₉₆ =527 h	51	
p-i-n	In N ₂ ; 25 °C	no	1-sun LED illumination	T ₉₀ =1200 h	52	
p-i-n	In ambient air; 40%RH; 25°C	no	1-sun illumination	T _{90.6} =100 h	53	
p-i-n	In N ₂ ; 25 °C	no	1-sun illumination	T ₉₅ =280 h	54	
n-i-p	In N ₂ ; 25 °C	no	1-sun LED illumination	T ₈₀ =700 h	55	
p-i-n	In ambient air; 40– 60%RH; 25–35 °C	no	1-sun xenon- lamp illumination (without a UV filter)	T ₈₀ =755 h	56	
p-i-n	In ambient air; 25 °C	yes	AM 1.5G illumination	T _{96.2} =1068 h	37	
p-i-n	In ambient air; 40%RH	yes	0.8-sun infrared- enhanced white LED	T ₉₅ =1000 h	9	
p-i-n	In ambient air; 25%RH; 25 °C	no	1-sun illumination	T ₉₆ =527 h	30	
p-i-n	In ambient air; 40– 60%RH; 25–35 °C	no	1-sun xenon- lamp illumination (without a UV filter)	T _{85.4} =1020 h	This work	

The light-soaking stability test was performed under N₂ atmosphere to

eliminate the moisture effect, which shows that the DCI-treated tandem retains the 91.4% of the initial PCE after 792 h, which is higher than the control tandem device (84.6%), as shown in **Figure R2**. It has been reported that the substantial release of iodine compound from perovskite film during light soaking is mainly due to interstitial iodide or PbI_2 defects [Science 2020, 369, 96. 10.1126/science.aba1628; Science 2021, 373, 902. 10.1126/science.abi6323]. Passivating those defects by DCI passivation suppresses the iodine generation and migration in perovskite films. The activation energy for ion migration was additionally obtained to reveal the ion-induced migration character in control and DCI-treated perovskite films using temperature-dependent conductivity measurements [Nature 2024, 628, 93. 10.1038/s41586-024-07189-3]. The activation energy value of the control perovskite is fitted to be 46 meV. In contrast, the activation energy value for the DCI-treated film is increased to 59.3 meV (**Figure R3a**). This result indicates that DCI have a substantially boosted energy barrier for ion migration, which might also improve the perovskite crystal stability. This result can also be confirmed by the transient ionic current (TIC) measurements (**Figure R3b**) as the reduced mobile ion concentration in the DCI-treated device ($2.98 \times 10^{17} \text{ cm}^{-3}$) compared to the control device ($6.72 \times 10^{17} \text{ cm}^{-3}$).

Overall, the enhanced stability of DCI-treated perovskite/silicon tandem devices is mainly attributed to comprehensive defect passivation, driven by the well-defined cation/anion spatial distribution of DCI within the perovskite film, and the significant suppression of ion migration.

Figure R2 Light soaking stability of unencapsulated control and DCI-treated

perovskite/silicon TSCs under 1-sun illumination (LED, 100 mW cm^{-2}) in N_2 atmosphere at open-circuit conditions.

Figure R3 (a) Temperature-dependent conductivity of control and DCI-treated perovskite films. (b) Transient ionic current (TIC) for extracting mobile ion concentration within the perovskite films of the control and DCI-treated devices.

Our response:

The originally provided MPP tracking had been revised to **Fig. S56**.

The summary of MPP stability for the state-of-the-art monolithic perovskite/silicon TSCs reported in literatures had been added to **Table S12**.

The measurements of temperature-dependent conductivity and transient ionic current had been added in **Fig. S15** and **Fig. S16**, respectively.

On **page 28, line 5**, “The current density versus voltage (J – V) curves were measured using an Enlitech SS-F5-3A solar simulator based on the Ushio Xe short arc lamp 500 under 1-sun conditions (100 mW cm^{-2} , AM 1.5G).” had been revised to “The current density versus voltage (J – V) curves, **temperature-dependent conductivity and transient ionic current** were measured using an Enlitech SS-F5-3A solar simulator based on the Ushio Xe short arc lamp 500 under 1-sun conditions (100 mW cm^{-2} , AM 1.5G).”.

On **page 20, line 17**, “the control tandem device suffers from a rapid degradation to $\sim 72.2\%$ of the initial PCE after 259 hours of MPP tracking. By contrast, the DCI-treated tandem device still maintains $\sim 95.9\%$ of its original PCE after 219 hours of MPP tracking, highlighting the significant role of DCI in

achieving perovskite/silicon TSCs with long-term stability.” had been revised to “the control tandem device suffers from rapid degradation, retaining only ~43.4% of the initial PCE after 640 hours of MPP tracking. In contrast, the DCI-treated tandem device maintains ~85.4% of its original PCE after 1020 hours of MPP tracking, demonstrating stability comparable to the state-of-the-art perovskite/silicon TSCs reported under similar test conditions (Table S12).”.

On page 20, line 22, “The light-soaking stability was subsequently tracked under N₂ atmosphere, which shows that the DCI-treated tandem device retains 91.4% of the initial PCE after 792 h, significantly outperforming the control tandem (84.6% retention after 624 h) (Fig. S57). The improved stability of DCI-treated tandems under light soaking is primarily attributed to the synergistic effects of DCI for comprehensively passivating charge trap states (Fig. 1g) and substantially suppressing ion migration (Fig. S15 and S16).” had been added.

On page 1, line 25, “Furthermore, two-terminal monolithic perovskite/silicon tandem solar cells based on tunnel oxide passivating contact yield an impressive PCE of 31.1% (1.0 cm²) and demonstrate a decent operational stability (ISOS-L-1, T₉₀ > 200 hours in ambient conditions without encapsulation).” had been revised to “Furthermore, two-terminal monolithic perovskite/silicon tandem solar cells based on tunnel oxide passivating contact yield an impressive PCE of 31.1% (1.0 cm²) and demonstrate a decent operational stability (ISOS-L-1, T₈₅ > 1020 hours in ambient conditions without encapsulation).”.

On page 3, line 24, “...eventually obtaining an impressive PCE of 31.1% (1.0 cm²) accompanied by an excellent long-term operational stability (95.9% retention after 219 hours under ISOS-standardized maximum power point tracking).” had been revised to “...eventually obtaining an impressive PCE of 31.1% (1.0 cm²) accompanied by an excellent long-term operational stability (85.4% retention after 1020 hours under ISOS-standardized maximum power

point tracking).”.

On **page 22, line 11**, “...eventually enabling a remarkable PCE of 31.1% (1.0 cm²) accompanied by an excellent long-term operational stability under ambient conditions (ISOS-L-1, T₉₀ > 200 h).” had been revised to “...eventually enabling a remarkable PCE of 31.1% (1.0 cm²) accompanied by an excellent long-term operational stability under ambient conditions (ISOS-L-1, **T₈₅ > 1020 h**).”.

#4-3. It would be useful to provide a control experiment in which a caged diammonium salt molecule is not used, but rather a similar molecule with only Lewis acid/base function is tested to highlight the uniqueness of the method.

Our reply: We are of great gratitude for your instructive suggestions. To highlight the uniqueness of our designed DCI, we additionally selected other two similar molecules—4,4-difluoropiperidine hydrochloride (2FPDCI) [Small Methods 2024, 8, e2400425. 10.1002/smt.202400425; Nat. Commun. 2024, 15, 8753. 10.1038/s41467-024-53121-8] and piperazinium hydrochloride (PCI) [Nano-Micro Lett. 2025, 17, 141. 10.1007/s40820-024-01631-x] containing the same Cl anion as DCI for comparison. Similar to DCI, both 2FPDCI and PCI incorporate Lewis acid and Lewis base functional groups within their cationic moieties. We systematically investigated the comparative effects of these three materials on the morphology and crystallographic orientation of perovskite films. As shown in **Figure R4**, for the control sample, the perovskite surface consists of clear steps (or terraced edges) and discontinuous PbI₂ filling the grain boundaries. In contrast, the PCI treatment drastically smooths out these steps and reacts with residual PbI₂, which is in line with our previous report [Nano-Micro Lett. 2025, 17, 141. 10.1007/s40820-024-01631-x]. For the 2FPDCI-treated perovskite film, platelet-like substances with significantly different sizes are observed on the surface compared to the control and PCI-treated samples,

which is due to the formation of $(2\text{FPD})_2\text{PbI}_4$ 2D phase on the surface of the 2FPDCI-treated sample [Small Methods 2024, 8, e2400425. 10.1002/smt.202400425]. Obviously, both PCI-treated and 2FPDCI-treated samples show different surface morphologies in comparison with the DCI-treated sample. To qualitatively analyze the structure of the samples, we further performed grazing-incidence wide-angle X-ray scattering (GIWAXS) measurement with a grazing-incidence angle of 0.2° , as depicted in Figure R5. Careful observation, reveals a new signal at $q_z = 0.49 \text{ \AA}^{-1}$ in only the 2FPDCI-treated sample and a new signal at $q_{xy} = 0.72 \text{ \AA}^{-1}$ in only the DCI-treated sample. Evidently, the crystallographic orientation of the 2D phase at the DCI-treated perovskite surface is entirely different from that at the 2FPDCI-treated perovskite surface, indicating that the orientation of 2D perovskite is also influenced by the spatial dimensionality of the molecules, which has rarely been highlighted before. In conclusion, from the aforementioned SEM and GIWAXS results, although these three molecules contain similar Lewis acid/base groups, the morphology and crystallographic orientation of the resultant 2D perovskites are completely different. For PCI treatment, no new phase is formed. For 2FPDCI treatment, horizontally-oriented (i.e. out-of-plane orientation) $(2\text{FPD})_2\text{PbI}_4$ 2D perovskite is formed, which is normally a representation of the orientation for most 2D perovskites. However, for DCI treatment, the resultant 2D perovskite features vertically oriented growth, i.e. in-plane orientation, highlighting that the spatial dimensionality of the molecules also plays a key role in determining the orientation of 2D perovskite.

Figure R4 SEM images of the control, PCI-treated and 2FPDCI-treated

perovskite films.

Figure R5 2D GIWAXS patterns of the control, PCI-treated, 2FPDCI-treated and DCI-treated perovskite films with a grazing incidence angle of 0.2° .

Our response:

The SEM images of the control, PCI-treated and 2FPDCI-treated perovskite films had been added to **Fig. S9**.

The 2D GIWAXS patterns of the control, PCI-treated, 2FPDCI-treated and DCI-treated perovskite films had been added to **Fig. S10**.

On page 5, line 27, “It is additionally revealed that the spatial dimensionality of cations is pivotal in dictating the orientation of 2D perovskites. Comparatively, 3D caged diammonium cation is more inclined to form in-plane oriented 2D perovskite (Fig. S9 and S10).” had been added.

On page 22, line 23, in the part of “**Materials**”, “1,4-diazabicyclo[2.2.2]octane and hydrochloric acid (37%) were ordered from Aladdin.” had been revised to “1,4-diazabicyclo[2.2.2]octane, 4,4-difluoropiperidine hydrochloride (2FPDCI) and hydrochloric acid (37%) were ordered from Aladdin.”.

On page 22, line 26, in the part of “**Materials**”, “PCI was synthesized

according to our previous report³⁶.” had been added.

#4-4. Certain data trends (e.g., increase work-function after DCI treatment) should be cross-validated by other measurements such as density functional theory (DFT) calculations.

Our reply: We are truly grateful for the reviewer’s constructive suggestion. To elucidate the fundamental mechanism underlying the surface energy transformation, we performed DFT calculations to investigate the evolution of the perovskite surface work function before and after DCI treatment. As shown in Figure R6, before the adsorption of cage-like cations, the control perovskite exhibits a high SWF of -1.95 eV (Figure R6a). Once the cage-like cations adsorb onto the perovskite surface (Figure R6b), the SWF dramatically decreases to -0.25 eV, corresponding to an upshift of 1.7 eV relative to the untreated control, consistent with UPS and KPFM measurements. This significant decrease in SWF is mainly attributed to the strong interfacial dipole generated by the spontaneous polarization of cage-like cations (Figure R6b) [Physical Review B 2003, 68, 195408. 10.1103/PhysRevB.68.195408; Energy Environ. Sci. 2021, 14, 1429. 10.1039/D0EE02216K]. The charge density difference reveals that this interfacial dipole significantly drives charge transfer from the perovskite to the cage-like cations (Figure R7), thereby decreasing the SWF [ACS Energy Lett. 2024, 9, 2982. 10.1021/acsenerylett.4c00839].

Figure R6 DFT calculations of the surface work functions of perovskite without

and with adsorption of cage-like cations. (a) Bare perovskite, (b) Perovskite with cage-like cations adsorption.

Figure R7 Charge density difference of perovskite with cage-like cation adsorption. Yellow iso-surface signifies electron accumulation, while blue represents electron depletion.

Our response:

DFT calculated surface work functions of perovskite without and with adsorption of cage-like cations and the charge density difference had been added to Fig. 3c, Fig. 3d and Fig. 3e, respectively.

On page 13, line 4, “Subsequent DFT calculations were conducted to explore the evolution of the perovskite surface work function (SWF). As shown in Fig. 3c, the control perovskite exhibits a high SWF of -1.95 eV. Upon adsorption of cage-like cations, the SWF dramatically decreases to -0.25 eV (Fig. 3d), corresponding to an upshift of 1.7 eV, consistent with UPS and KPFM measurements. Additionally, the introduction of cage-like cations induces a strong interfacial dipole originating from their spontaneous polarization (Fig. 3d). The charge density difference reveals that this interfacial dipole significantly drives charge transfer from the perovskite surface to the cage-like cations, thereby decreasing the SWF (Fig. 3e)⁴⁸.” had been added.

On page 31, line 15, in the part of “DFT calculations”, “Calculations of defect formation energy, partial density of state, surface work function

and charge density difference The bottom four layers were fixed, while the remaining atoms were fully relaxed. A vacuum layer of at least 20 Å was introduced along the z-direction to prevent periodic interactions. Calculations employed a plane-wave energy cutoff of 500 eV and a k-point mesh of 1×1×1 for slab models (2×2×1 for DOS calculations). Structural optimization was performed with a force convergence threshold of 0.05 eV/Å and an energy convergence criterion of 1×10⁻⁵ eV.” had been added.

#4-5. The discussion of how the ferroelectric properties of quasi-two-dimensional packet crystals can enhance carrier extraction is interesting, but a firmer theoretical foundation would be helpful. In addition, a quantitative estimate of the built-in electric field generated by the ferroelectric effect would have strengthened the argument.

Our reply: We deeply appreciate the reviewer’s valuable feedback and guidance. We systematically elucidated the intrinsic correlation between the ferroelectric effect and enhanced carrier extraction through density functional theory (DFT) calculations. **Figure R8** simulates the ferroelectric switching of electric dipole induced by external polarization.

Figure R8 Schematic diagram of ferroelectric polarization for the perovskite surface.

Our DFT results demonstrate that the enhancement of charge extraction by the ferroelectric effect originates from a ferroelectric polarization-induced decrease in the perovskite surface work function (SWF). As illustrated in **Figure R9**, we calculated the SWF changes of perovskite surfaces with adsorbed cage-like cations possessing various dipole orientations.

In the absence of cage-like cations, the control perovskite exhibits a relatively high SWF of -1.95 eV (Figure R9a). Upon adsorption of randomly oriented cage-like cations, the SWF dramatically decreases to -0.25 eV (Figure R9b), primarily due to the formation of a strong interfacial dipole. This dipole arises from the spontaneous polarization of the cage-like cations, which induces charge transfer from the perovskite to the cations, effectively decreasing the SWF (Figure R7). The lower SWF induces an additional dipole field that facilitates electron extraction while concurrently suppressing hole injection.

Upon preliminary polarization, certain electric dipoles undergo ferroelectric switching, leading to a further reduction of the SWF to -0.06 eV (Figure R9c). In this state, the surface electric dipoles evolve from a disordered to a directionally ordered configuration, thereby generating a stronger interfacial dipole that further enhances charge separation and transport efficiency. When the electric dipoles are fully polarized, a uniformly oriented interfacial dipole layer forms, further upshifting the SWF to 0.03 eV (Figure R9d), a condition that further favors charge separation and extraction.

Accordingly, these results indicate that ferroelectric-enhanced charge transport originates from a decrease in SWF caused by the oriented align of spontaneously polarized cage-like cations under ferroelectric polarization. The decrease of SWF induced by external polarization will contribute to the initial dipole field (Figure R9b), thereby further enhancing the E_{bi} of the device. This enhancement promotes more effective charge carrier separation and transport to the electron transport layer, ultimately leading to improved device performance.

Notably, a decrease in SWF is usually accompanied by an enhancement in the built-in electric field for inverted perovskite solar cells [Science 2022, 375, 434. 10.1126/science.abl5676]. We sincerely thank the reviewer for the

valuable comment. Although our calculations do not directly provide the variation in the built-in electric field, the change in the surface work function (SWF) serves as a reliable indirect indicator. Since the SWF is closely related to the surface potential and internal field distribution, its variation effectively reflects the modulation of the internal electric field induced by ferroelectric polarization. Therefore, the observed changes in SWF can reasonably be used to infer variations in the built-in electric field during the ferroelectric polarization process.

Figure R9 DFT calculations of the surface work functions of perovskite without and with adsorption of cage-like cations. (a) Bare perovskite, (b) Cage-like cation adsorption in a disordered orientation, (c) Preliminary polarization, (d) Complete polarization.

Our response:

On page 18, line 11, “In a word, E_{bi} enhancement driven by the external polarization of the DCI-mediated ferroelectric quasi-2D perovskite is key to boosting device performance.” had been deleted.

DFT calculated surface work functions of perovskite without and with adsorption of cage-like cations and the charge density difference had been

added to **Fig. 3c**, **Fig. 3d** and **Fig. 3e**, respectively.

DFT calculated surface work functions of perovskite during the ferroelectric polarization process had been added to **Fig. S49**.

On page 18, line 12, “To delineate the ferroelectric enhancement mechanism, we performed DFT calculations on the surface work function (SWF) evolution of the perovskite (Fig. S48). From Fig. 3c and d, the adsorption of randomly oriented cage-like cations leads to a dramatic upshift in the SWF from -1.95 eV to -0.25 eV, generating an additional dipole field that facilitates electron extraction while concurrently suppressing hole injection. Upon preliminary polarization, certain electric dipoles undergo ferroelectric switching, further lowering the SWF to -0.06 eV (Fig. S49a). In this state, the surface electric dipoles evolve from a disordered to a directionally ordered configuration, forming a stronger interfacial dipole. When fully polarized, a uniformly oriented interfacial dipole layer forms, upshifting the SWF to 0.03 eV (Fig. S49b), favoring more charge separation and extraction. Accordingly, these results indicate that ferroelectric-enhanced charge transport primarily originates from SWF reduction caused by the oriented align of spontaneously polarized cage-like cations under ferroelectric polarization. The polarization-induced SWF reduction establishes a reinforced interfacial dipole field that amplifies the E_{bi} of the device, which promotes more effective charge carrier separation and transport to the electron transport layer, ultimately improving device performance.” had been added.

On page 31, line 15, in the part of “**DFT calculations**”, “**Calculations of defect formation energy, partial density of state, surface work function and charge density difference**” The bottom four layers were fixed, while the remaining atoms were fully relaxed. A vacuum layer of at least 20 Å was introduced along the z-direction to prevent periodic interactions. Calculations employed a plane-wave energy cutoff of 500 eV and a k-point mesh of $1 \times 1 \times 1$ for slab models ($2 \times 2 \times 1$ for

DOS calculations). Structural optimization was performed with a force convergence threshold of 0.05 eV/\AA and an energy convergence criterion of $1 \times 10^{-5} \text{ eV}$ had been added.

#4-6. Are there potential limitations if cage-like diammonium molecules are used in other perovskite compositions?

Our reply: We appreciate your professional comment. To verify the potential impacts of cage-like diammonium molecules on other perovskite compositions, we integrated DCI into single-junction solar cells based on 1.68 eV MA-free and 1.79 eV wide-bandgap perovskites. As depicted in Figure R10, both 1.68 eV MA-free and 1.79 eV wide-bandgap perovskite solar cells exhibit decent photovoltage performances. For 1.68 eV MA-free perovskite solar cells, the best-performing device achieves a PCE of 22.9%, with an V_{oc} of 1.25 V (Figure R10a). For 1.79 eV wide-bandgap perovskite solar cells, the corresponding devices yield a champion PCE of 19.4% with an V_{oc} of 1.30 V (Figure R10b). Both types of devices show negligible hysteresis. These findings demonstrate the generality of DCI in terms of various perovskite compositions and highlight the potential of DCI in all perovskite tandem solar cells. The corresponding integrated J_{sc} values of the champion perovskite solar cells based on 1.68 eV MA-free and 1.79 eV perovskites are 21.3 mA cm^{-2} and 17.5 mA cm^{-2} , respectively (Figure R11), which matches well with the J_{sc} values extracted from the J-V curves.

Figure R10 J-V curves of perovskite solar cells fabricated with (a) 1.68 eV MA-

free and (b) 1.79 eV wide-bandgap perovskites.

Figure R11 EQE curves and the corresponding integrated photocurrent densities for 1.68 eV MA-free and 1.79 eV wide-bandgap perovskite solar cells.

Our response:

J–V curves and EQE spectra and the corresponding integrated J_{sc} s for 1.68 eV MA-free and 1.79 eV wide-bandgap perovskite solar cells had been added in **Fig. S40**.

On page 16, line 18, “Finally, DCI was successfully applied to 1.68 eV MA-free and 1.79 eV wide-bandgap perovskites, yielding PCEs of 22.9% and 19.4%, respectively (Fig. S40).” had been added.

On page 22, line 16, in the part of “**Materials**”, “Methylammonium bromide (MABr) and Formamidinium iodide (FAI) were acquired from GreatCell Solar Ltd.” had been revised to “Methylammonium bromide (MABr), Lead(II) chloride ($PbCl_2$) and Formamidinium iodide (FAI) were acquired from GreatCell Solar Ltd.”.

On page 22, line 19, in the part of “**Materials**”, “Lead(II) bromide ($PbBr_2$), lead(II) iodide (PbI_2) and [2-(3,6-dimethoxy-9H-carbazol-9-yl) ethyl] phosphonic acid (MeO-2PACz) were supplied by TCI.” had been revised to “Lead(II) bromide ($PbBr_2$), lead(II) iodide (PbI_2), Lead(II) Thiocyanate ($Pb(SCN)_2$), Potassium Iodide (KI) and [2-(3,6-dimethoxy-9H-carbazol-9-yl) ethyl] phosphonic acid (MeO-2PACz) were supplied by TCI.”.

On page 22, line 24, in the part of “**Materials**”, “(4-(7H-dibenzo[c,g]carbazol-7yl)butyl)phosphonic acid (4PADCB) was purchased from

Suzhou LiWei Tech Co., Ltd.” had been added.

On page 23, line 19, in the part of “**Single-junction perovskite solar cells fabrication**”, “For 4PADCBC-based devices, 4PADCBC was dissolved in ethanol (0.5 mg mL^{-1}), and the resultant solution was spin-coated at 3000 rpm for 30 seconds, followed by heating at $100 \text{ }^\circ\text{C}$ for 10 minutes without additional washing.” had been added.

On page 23, line 25, in the part of “**Single-junction perovskite solar cells fabrication**”, “For 1.6 M MA-free $\text{Cs}_{0.17}\text{FA}_{0.83}\text{Pb}(\text{I}_{0.83}\text{Br}_{0.17})_3$ perovskite precursor solution, PbCl_2 (19.5 mg), CsI (61.8 mg), PbBr_2 (131.0mg), FAI (199.9 mg), PbI_2 (495.3 mg) were dissolved in 0.88 mL mixed solvent of DMF/DMSO (v/v, 3:1). For 1.2 M $\text{FA}_{0.8}\text{Cs}_{0.2}\text{Pb}(\text{I}_{0.6}\text{Br}_{0.4})_3$ perovskite precursor solution, 3.9 mg $\text{Pb}(\text{SCN})_2$, 4.0 mg KI, 62.4 mg CsI, 264.2 mg PbBr_2 , 165.1 mg FAI, 221.3 mg PbI_2 were dissolved in 1 mL mixed solvent of DMF/DMSO (v/v, 4:1). Both precursor solutions were stirred for 2 hours before use.” had been added.

On page 24, line 11, in the part of “**Single-junction perovskite solar cells fabrication**”, “For the fabrication of MA-free $\text{Cs}_{0.17}\text{FA}_{0.83}\text{Pb}(\text{I}_{0.83}\text{Br}_{0.17})_3$ perovskite films (1.68 eV), the perovskite precursor solution was spin-coated onto 4PADCBC-coated substrates at 4,000 rpm for 40 seconds. During spinning, 200 μL of CB anti-solvent was dripped onto the substrate at 10 seconds before the end of spin-coating, and the as-prepared films were immediately annealed at $100 \text{ }^\circ\text{C}$ for 20 minutes. For the $\text{FA}_{0.8}\text{Cs}_{0.2}\text{Pb}(\text{I}_{0.6}\text{Br}_{0.4})_3$ perovskite films (1.79 eV), the perovskite precursor solution was dripped onto 4PADCBC-coated substrates and spin-coated at 4,000 rpm for 40 seconds. 200 μL of CB was dripped after 30 seconds of spin-coating, followed by annealing at $100 \text{ }^\circ\text{C}$ for 15 min. Both film types were subsequently treated with 0.4 mg mL^{-1} DCI via spin-coating at 6,000 rpm for 35 seconds, followed by final annealing at $100 \text{ }^\circ\text{C}$ for 10 minutes.” had been added.

#4-7. The impact of the present findings on the field of 2T perovskite/silicon tandem cells could be more clearly clarified, especially in terms of scalability and commercial viability.

Our reply: Thanks for your constructive suggestions. We all know that once the active area of a perovskite solar cell is enlarged, the problem of shunting caused by defects within the perovskite will be more deleterious, leading to a reduction in efficiency. To gain more insights into the scalability of DCI on perovskite/silicon tandem solar cells, we further enlarged the perovskite/silicon tandem area to 4 cm². As demonstrated in Figure R12, for the DCI-based tandem cells, we obtained an impressive efficiency of 28.4% (26.7%), with a V_{OC} of 1.909 (1.898) V, an FF of 76.8% (72.5%) and a J_{SC} of 19.4 (19.4) mA cm⁻² under reverse (forward) scan (Figure R12a). Additionally, a stabilized PCE of 27.9% during 600 s with a constant bias of 1.58 V was achieved (Figure R12b). Notably, our large-area tandem photovoltaic devices were solution-processed exclusively through spin-coating deposition. To our knowledge, the impressive PCE of 28.4% is one of the highest values for reported large-area (≥ 4 cm²) perovskite/silicon tandem solar cells based on spin-coating deposition method (Table R4) [Inf Funct Mater. 2024, 1, 160. 10.1002/ifm2.18].

Additionally, DCI also held a high potential to enhance the commercial viability of perovskite/silicon tandem solar cells owing to its remarkable attributes, such as simple synthesis process (see Methods), low cost of the raw material and improved stability (Figure R1 and R2).

Figure R12 (a) J–V curves and (b) steady-state PCE with a constant bias of 1.58 V for the large-area monolithic perovskite/silicon tandem solar cells with an active area of 4 cm².

Table R4 Summary of large-area monolithic perovskite/silicon tandems based on the spin-coating method reported in literatures. HJ: homojunction; SHJ: silicon heterojunction; TOPCon: tunnel oxide passivated contact.

PCE [%]	J_{sc} [mA cm^{-2}]	V_{oc} [V]	FF [%]	silicon	type	Area [cm^2]	ref.
20.5	16.1	1.68	78.0	HJ	n-i-p	4	1
21.8	16.2	1.74	78.0	HJ	n-i-p	16	2
22.6	17.5	1.72	75.0	SHJ	p-i-n	57.4	3
23.1	16.5	1.78	81.0	HJ	n-i-p	4	4
27.1	19.1	1.88	75.5	SHJ	p-i-n	3.8	5
17.3	14.1	1.78	67.0	TOPerc	n-i-p	25	6
28.3	19.1	1.84	80.2	TOPCon	p-i-n	4	7
27.1	/	/	/	SHJ	n-i-p	24	8
26.4	19.0	1.91	72.6	SHJ	p-i-n	8.9	9
28.4	19.4	1.91	76.8	TOPCon	p-i-n	4	This work

Our response:

J–V curves and steady-state PCE of the 4 cm² large-area monolithic

perovskite/silicon tandem solar cells had been added to **Fig. 4g**.

Summary of large-area monolithic perovskite/silicon TSCs based on the spin-coating method reported in literatures had been added to **Table S11**.

On page 20, line 5, “We further enlarged the tandem area to 4 cm², achieving a remarkable a PCE of 28.4% with a V_{oc} of 1.909 V, an FF of 76.8% and a J_{sc} of 19.4 mA cm⁻² under reverse scan (Fig. 4g). A stabilized PCE of 27.9% is also obtained (Fig. S55). Notably, all large-area tandem cells were fabricated solely by spin-coating. To the best of our knowledge, this is among the highest PCEs reported for spin-coated perovskite/silicon TSCs with areas ≥ 4 cm² (Table S11), underscoring the scalability of DCI. Moreover, DCI offers strong commercial potential due to its facile synthesis (Fig. S1), low-cost precursors, and enhanced device stability.” had been added.

On page 26, line 3, in the part of “**Monolithic perovskite/silicon tandem solar cells fabrication**”, “For large-area perovskite/silicon TSCs with an active area of 4 cm², the perovskite precursor was deposited via spin-coating (3,500 rpm, 40 seconds) onto MeO-2PACz-functionalized substrates. During the spinning process, 220 μ L of CB was precisely dripped onto the film center 15 seconds prior to the spin-coating ended. The coated substrates were immediately transferred to a pre-heated hotplate for thermal annealing (100 °C, 20 minutes). After that, a 0.4 mg mL⁻¹ DCI solution (100 μ L) was dynamically spin coated (6,000 rpm, 35 seconds) onto the cooled substrates, followed by annealing at 100 °C for 10 minutes. Subsequently, 13 nm of C₆₀ was thermally evaporated onto the substrates, after which 20 nm of SnO₂ was deposited via atomic layer deposition (MNT-PD100Oz-L3S1G3). The subsequent IZO process remained unchanged.” had been added.

Reference

1. Zheng J, *et al.* Large area efficient interface layer free monolithic perovskite/homo-junction-silicon tandem solar cell with over

20% efficiency. *Energy Environ Sci* **11**, 2432–2443 (2018).

2. Zheng J, *et al.* 21.8% Efficient Monolithic Perovskite/Homo-Junction-Silicon Tandem Solar Cell on 16 cm². *ACS Energy Lett* **3**, 2299–2300 (2018).

3. Kamino BA, *et al.* Low-Temperature Screen-Printed Metallization for the Scale-Up of Two-Terminal Perovskite–Silicon Tandems. *ACS Appl Energy Mater* **2**, 3815–3821 (2019).

4. Zheng J, *et al.* Large-area 23%-efficient monolithic perovskite/homojunction-silicon tandem solar cell with enhanced UV stability using down-shifting material. *ACS Energy Lett* **4**, 2623–2631 (2019).

5. Liu J, *et al.* 28.2%-efficient, outdoor-stable perovskite/silicon tandem solar cell. *Joule* **5**, 3169–3186 (2021).

6. Hyun JY, *et al.* Perovskite/Silicon Tandem Solar Cells with a Voc of 1784 mV Based on an Industrially Feasible 25 cm² TOPCon Silicon Cell. *ACS Appl Energy Mater* **5**, 5449–5456 (2022).

7. Walter A, *et al.* Rear textured p-type high temperature passivating contacts and their implementation in perovskite/silicon tandem cells. *Energy Advances* **2**, 1818–1822 (2023).

8. Ugur E, *et al.* Front-contact passivation through 2D/3D perovskite heterojunctions enables efficient bifacial perovskite/silicon tandem solar cells. *Matter* **6**, 2919–2934 (2023).

9. Zhu Z, *et al.* Low-Temperature Atomic Layer Deposition of Hole Transport Layers for Enhanced Performance and Scalability in Textured Perovskite/Silicon Tandem Solar Cells. *Adv Energy Mater* **14**, (2024).

10. Yang L, *et al.* Modulating Binding Strength and Acidity of Benzene-Derivative Ligands Enables Efficient and Hysteresis-Free Perovskite/Silicon Tandem Solar Cells. *Angew Chem Int Ed*, e202500350 (2025).

11. Xu Z, *et al.* Efficient and Stable Inverted MA/Br-Free 2D/3D Perovskite Solar Cells Enabled by α -to- δ Phase Transition Inhibition and Crystallization Modulation. *Energy Environ Sci* **18**, 1354–1365 (2025).

-
12. Xu Y, *et al.* Multicomponent Solvent Engineered Spatially Uniform 2D/3D Perovskite Heterojunction for Solar Cells. *ACS Energy Lett*, 2035-2044 (2025).
 13. Wang J, *et al.* Dipolar Carbazole Ammonium for Broadened Electric Field Distribution in High-Performance Perovskite Solar Cells. *Journal of the American Chemical Society* **147**, 8663-8671 (2025).
 14. Shih M-C, *et al.* A 2D/3D Heterostructure Perovskite Solar Cell with a Phase-Pure and Pristine 2D Layer. *Adv Mater*, 2416672 (2025).
 15. Chen H, *et al.* Quantum-size-tuned heterostructures enable efficient and stable inverted perovskite solar cells. *Nat Photonics* **16**, 352-358 (2022).
 16. Liu Z, *et al.* All-perovskite tandem solar cells achieving >29% efficiency with improved (100) orientation in wide-bandgap perovskites. *Nat Mater*, (2025).
 17. Ye S, *et al.* Expanding the low-dimensional interface engineering toolbox for efficient perovskite solar cells. *Nat Energy* **8**, 284-293 (2023).
 18. Wen J, *et al.* Heterojunction formed via 3D-to-2D perovskite conversion for photostable wide-bandgap perovskite solar cells. *Nat Commun* **14**, 7118 (2023).
 19. Kang Z, *et al.* Synchronous Dimension-Crystallization Engineering Enables Highly Efficient 2D/3D Tin Perovskite Solar Cells. *Energy Environ Sci*, (2025).
 20. Chang X, *et al.* Solvent-dripping modulated 3D/2D heterostructures for high-performance perovskite solar cells. *Nat Commun* **16**, 1042 (2025).
 21. Huang Y, *et al.* Finite perovskite hierarchical structures via ligand confinement leading to efficient inverted perovskite solar cells. *Energy Environ Sci* **16**, 557-564 (2023).
 22. Shen L, *et al.* Ion Diffusion Management Enables All-Interface Defect Passivation of Perovskite Solar Cells. *Adv Mater* **35**, 2301624

(2023).

23. Zhang A, *et al.* pi-pi Stacking at the Perovskite/C(60) Interface Enables High-Efficiency Wide-Bandgap Perovskite Solar Cells. *Small* **20**, e2401197 (2024).

24. Wang Y, *et al.* Homogenized contact in all-perovskite tandems using tailored 2D perovskite. *Nature* **635**, 867-873 (2024).

25. Kang S, *et al.* Boosting Carrier Transport in Quasi-2D/3D Perovskite Heterojunction for High-Performance Perovskite/Organic Tandems. *Adv Mater* **37**, e2411027 (2024).

26. Choi Y, *et al.* Vertically oriented two-dimensional Ruddlesden–Popper phase perovskite passivation layer for efficient and stable inverted perovskite solar cells. *Energy Environ Sci* **15**, 3369-3378 (2022).

27. Azmi R, *et al.* Damp heat–stable perovskite solar cells with tailored-dimensionality 2D/3D heterojunctions. *Science* **376**, 73-77 (2022).

28. Li S, *et al.* A Generic Strategy to Stabilize Wide Bandgap Perovskites for Efficient Tandem Solar Cells. *Adv Mater* **36**, 2307701 (2023).

29. Wang Z, *et al.* Regulation of Wide Bandgap Perovskite by Rubidium Thiocyanate for Efficient Silicon/Perovskite Tandem Solar Cells. *Adv Mater* **36**, e2407681 (2024).

30. Pei F, *et al.* A binary 2D perovskite passivation for efficient and stable perovskite/silicon tandem solar cells. *Nat Commun* **15**, 7024 (2024).

31. Fang Z, *et al.* Surface reconstruction of wide-bandgap perovskites enables efficient perovskite/silicon tandem solar cells. *Nat Commun* **15**, 10554 (2024).

32. Wang G, *et al.* Molecular engineering of hole-selective layer for high band gap perovskites for highly efficient and stable perovskite-silicon tandem solar cells. *Joule* **7**, (2023).

33. Qiao L, *et al.* Freezing Halide Segregation Under Intense Light

for Photostable Perovskite/Silicon Tandem Solar Cells. *Adv Energy Mater* **14**, 2302983 (2023).

34. Mariotti S, *et al.* Interface engineering for high-performance, triple-halide perovskite–silicon tandem solar cells. *Science* **381**, 63-69 (2023).

35. Li Z, *et al.* Stabilized hole-selective layer for high-performance inverted p-i-n perovskite solar cells. *Science* **382**, 284-289 (2023).

36. Zhou W, *et al.* A Universal Strategy for Defects and Interface Management Enables Highly Efficient and Stable Inverted Perovskite Solar Cells. *Energy Environ Sci*, (2025).

37. Ye T, *et al.* Molecular Bridge in Wide-Bandgap Perovskites for Efficient and Stable Perovskite/ Silicon Tandem Solar Cells. *Adv Funct Mater*, 2419391 (2025).

38. Wang L, *et al.* Highly Efficient Monolithic Perovskite/TOPCon Silicon Tandem Solar Cells Enabled by “Halide Locking”. *Adv Mater*, 2416150 (2025).

39. Tian L, *et al.* Divalent cation replacement strategy stabilizes wide-bandgap perovskite for Cu(In,Ga)Se₂ tandem solar cells. *Nat Photonics*, (2025).

40. Qu G, *et al.* Self-assembled materials with an ordered hydrophilic bilayer for high performance inverted Perovskite solar cells. *Nat Commun* **16**, 86 (2025).

41. Chen B, *et al.* Blade-Coated Perovskites on Textured Silicon for 26%-Efficient Monolithic Perovskite/Silicon Tandem Solar Cells. *Joule* **4**, 850-864 (2020).

42. Al-Ashouri A, *et al.* Monolithic perovskite/silicon tandem solar cell with >29% efficiency by enhanced hole extraction. *Science* **370**, 1300-1309 (2020).

43. Li Y, *et al.* Wide Bandgap Interface Layer Induced Stabilized Perovskite/Silicon Tandem Solar Cells with Stability over Ten Thousand Hours. *Adv Energy Mater* **11**, 2102046 (2021).

44. Li R, *et al.* CsPbCl₃-Cluster-Widened Bandgap and Inhibited Phase Segregation in a Wide-Bandgap Perovskite and its Application to NiO_x-Based Perovskite/Silicon Tandem Solar Cells. *Adv Mater* **34**, 2201451 (2022).

45. Ren N, *et al.* 50 °C low-temperature ALD SnO₂ driven by H₂O₂ for efficient perovskite and perovskite/silicon tandem solar cells. *Appl Phys Lett* **121**, 033502 (2022).

46. Ji SG, *et al.* Stable pure-iodide wide-band-gap perovskites for efficient Si tandem cells via kinetically controlled phase evolution. *Joule* **6**, 2390-2405 (2022).

47. Wang X, *et al.* Long-chain anionic surfactants enabling stable perovskite/silicon tandems with greatly suppressed stress corrosion. *Nat Commun* **14**, 2166 (2023).

48. Li X, *et al.* Surface Reconstruction for Efficient and Stable Monolithic Perovskite/Silicon Tandem Solar Cells with Greatly Suppressed Residual Strain. *Adv Mater* **35**, e2211962 (2023).

49. Liu J, *et al.* Perovskite/silicon tandem solar cells with bilayer interface passivation. *Nature*, (2024).

50. Liu J, *et al.* Textured Perovskite/Silicon Tandem Solar Cells Achieving Over 30% Efficiency Promoted by 4-Fluorobenzylamine Hydroiodide. *Nano-Micro Letters* **16**, 189 (2024).

51. Pei F, *et al.* A binary 2D perovskite passivation for efficient and stable perovskite/silicon tandem solar cells. *Nat Commun* **15**, 7024 (2024).

52. Sun Y, *et al.* Ionic Liquid Modified Polymer Intermediate Layer for Improved Charge Extraction toward Efficient and Stable Perovskite/Silicon Tandem Solar Cells. *Small* **20**, 2308553 (2024).

53. Wang X, *et al.* Ultrathin (~30 μm) flexible monolithic perovskite/silicon tandem solar cell. *Science Bulletin* **69**, 1887-1894 (2024).

54. Yang T, *et al.* Efficient and Stable Perovskite/Silicon Tandem Solar Cells Modulated with Triple-Functional Passivator. *Adv Energy*

Mater n/a, 2303149 (2023).

55. Ding Z, *et al.* Highly passivated TOPCon bottom cells for perovskite/silicon tandem solar cells. *Nat Commun* **15**, 8453 (2024).

56. Li X, *et al.* Top-Down Dual-Interface Carrier Management for Highly Efficient and Stable Perovskite/Silicon Tandem Solar Cells. *Nano-Micro Letters* **17**, 141 (2025).

57. Li T, *et al.* Inorganic wide-bandgap perovskite subcells with dipole bridge for all-perovskite tandems. *Nat Energy* **8**, 610-620 (2023).

Reviewer #1:

Accept as it is

Response: We appreciate the reviewer's support for the publication of this manuscript.

Reviewer #2:

The revised manuscript and accompanying response letter address some of the technical questions raised and provide clarification on a few points from the original submission. However, the revision does not fundamentally enhance the quality or impact of the work. Several critical issues remain unresolved, including a lack of substantial novelty, only moderate device performance, unclear scalability, unclear relationship between the ferroelectricity and device stability, and insufficient evidence to support the claimed role of ferroelectricity in enhancing device performance, etc. These limitations prevent the manuscript from making a significant contribution to the field and, in my opinion, it does not meet the high standards required for publication in *Nature Communications*.

Our response: We sincerely thank the reviewer for the critical and constructive evaluation of our revised manuscript. We appreciate the reviewer's recognition that some of the technical questions were addressed and certain points clarified. At the same time, we acknowledge the reviewer's concerns regarding the overall novelty, device performance, scalability, and the clarity of the correlation between ferroelectricity and device stability and performance enhancement.

We take these comments seriously and fully understand that meeting the high standards of *Nature Communications* requires not only technical soundness but also significant conceptual advances and broader impact. In the following point-by-point responses, we have made further efforts to address the reviewer's concerns in detail. We hope that our point-by-point responses and the substantial revisions we have made will better convey the value of our work and its potential contribution to the field.

1. The authors state: "To the best of our knowledge, the impressive PCE of 28.4% is among the highest values for reported large-area ($\geq 4 \text{ cm}^2$) perovskite/silicon

tandem solar cells based on spin-coating method... These results demonstrate that DACI is not only effective at the lab scale but also holds promise for scalable and broadly applicable device engineering.” However, a 4 cm² device is still far too small to draw any meaningful conclusions about the scalability of the proposed passivation strategy, particularly since the molecule is applied using a spin-coating method. This approach is not compatible with industrial-scale processing, where cell areas are well above 100 cm². Therefore, how this passivation method could be transferred to large-area, scalable fabrication remains an open and significant challenge.

Our reply:

Sorry for the unclear expression. We fully agree that true industrial-scale implementation requires device areas well beyond 100 cm² and processing methods compatible with scalable manufacturing techniques like slot-die coating, dynamic spray coating or thermal evaporation [DOI: [10.1002/anie.202407151](https://doi.org/10.1002/anie.202407151); [10.1126/science.adv4280](https://doi.org/10.1126/science.adv4280)] rather than spin coating.

We fully understand the importance of demonstrating scalability to industrially relevant areas (e.g., >100 cm²). However, given the limitations of our current laboratory environment and processing tools, it is not yet feasible for us to fabricate such large-area perovskite/silicon tandem solar cells using slot-die coating or thermal evaporation to demonstrate the industrial-scale scalability of DCI so far. We sincerely apologize for this limitation and appreciate the reviewer’s understanding, and we are grateful to the reviewer for pointing out this important aspect of scalability.

Our response:

To reflect the current stage of our work more scientifically, we have revised the relevant sentence in the main manuscript:

On page 20, line 8, “Notably, all large-area tandem cells were fabricated solely by spin-coating.” had been deleted.

On page 20, line 9, “To the best of our knowledge, this is among the highest PCEs reported for spin-coated perovskite/silicon TSCs with areas $\geq 4 \text{ cm}^2$ (Table S11), underscoring the scalability of DCI. Moreover, DCI offers strong commercial potential due to its facile synthesis (Fig. S1), low-cost precursors, and enhanced device stability.” had been revised to “To the best of our knowledge, this is among the highest PCEs reported for spin-coated perovskite/silicon TSCs with areas $\geq 4 \text{ cm}^2$ (Table S11).”.

We hope these more cautious and realistic descriptions addresses the reviewer’s concern appropriately.

2. The manuscript also lacks a clear explanation of the role of ferroelectricity in improving device stability. The authors claim enhanced stability due to ferroelectric effects, but do not present sufficient experimental or theoretical evidence to support this mechanism.

Our reply:

We apologize for the inconvenience caused to the reviewer due to our inaccurate expression. We would like to clarify that our manuscript does not attribute the enhanced device stability directly to the ferroelectricity of the cage-like diammonium cations. Instead, we concluded that the improved stability primarily arises from the synergistic effects of DCI for comprehensively passivating charge trap states (Fig. 1g) and substantially suppressing ion migration (Fig. S15 and S16) [DOI: [10.1038/s41467-025-56182-5](https://doi.org/10.1038/s41467-025-56182-5); [10.1039/d5ee00640f](https://doi.org/10.1039/d5ee00640f); [10.1038/s41467-025-56182-5](https://doi.org/10.1038/s41467-025-56182-5)]. The ferroelectric property of this molecule contributes to the enhancement of photovoltaic parameters including V_{OC} , FF and J_{SC} , likely by promoting efficient charge separation and reducing non-radiative recombination at the perovskite/ C_{60} interface. However, it remains challenging to accurately distinguish the effects of ferroelectricity from those of conventional chemical passivation on device stability at the

current stage. Although, it is worth noting that our long-term MPPT stability tests were conducted on DCI-modified tandem devices prior to polarization, meaning that the ferroelectric properties of the quasi-2D perovskite were not yet activated. Therefore, before poling, the observed stability enhancement in DCI-treated tandem can be primarily ascribed to effective defect passivation and suppressed ion migration, rather than to any ferroelectric effects. In our future work, we will be committed to investigating the impact of ferroelectricity on device stability.

3. Regarding device performance, the reported efficiencies are relatively modest compared to recent state-of-the-art results. Moreover, Table R2 does not present a complete or representative overview of the literature. Several recent studies have demonstrated efficiencies exceeding 23% for wide-bandgap perovskite sub-cells with similar bandgaps. Below are just a few relevant examples (not exhaustive):

1. Li, S., et al., A Generic Strategy to Stabilize Wide Bandgap Perovskites for Efficient Tandem Solar Cells, *Advanced Materials*, 2024, 36(9), 2307701.

2. Fang, Z., et al., Surface reconstruction of wide-bandgap perovskites enables efficient perovskite/silicon tandem solar cells, *Nature Communications*, 2024, 15(1), 1–11.

3. Wang, Z., et al., Regulation of Wide Bandgap Perovskite by Rubidium Thiocyanate for Efficient Silicon/Perovskite Tandem Solar Cells, *Advanced Materials*, 2024, 2407681.

The authors suggest that the relatively low efficiency in their study may be due to suboptimal hole transport layers or additive selection. If this is the case, the claim should be substantiated by experiments using improved HTLs or optimized perovskite compositions.

Our reply:

We sincerely appreciate the reviewer's thoughtful feedback and the helpful reference suggestions.

On the relatively modest efficiency: We fully agree with the reviewer that the reported efficiency of our wide-bandgap perovskite sub-cell is lower than some of the recent state-of-the-art results. In our current work, the focus was placed on understanding the interfacial effects of DCI treatment and validating device stability through certified measurements. Therefore, the device structure and perovskite composition were not yet fully optimized for maximum efficiency.

We hypothesize that the relatively modest efficiency may be attributed to the use of a non-ideal hole transport layer and unoptimized additive systems. We have now initiated further experiments with improved HTLs-(4-(7H-dibenzo[c,g]carbazol-7yl)butyl)phosphonic acid (4PADCB), which has also been employed in the literature recommended by the reviewer [[DOI: 10.1002/adma.202407681](https://doi.org/10.1002/adma.202407681)], and preliminary result shows encouraging enhancement in device performance. The 4PADCB-based device obtains an improved PCE of 22.9%, with a V_{oc} of 1.255 V, an FF of 84.7% and a J_{sc} of 21.6 mA cm⁻² (Figure R1). This indicates that the choice of an appropriate hole transport layer is crucial for optimizing device performance. Considering the better hole transport capability of 4PADCB, we will apply it in our future study on WBG perovskite and tandem solar cells.

In terms of additive engineering, we would like to note that identifying a suitable additive specifically compatible with our current perovskite composition remains a significant challenge. Although we have attempted preliminary screenings, none of the tested additives have led to stable or reproducible improvements in our system so far. Therefore, we have not included these results in the manuscript to maintain scientific rigor. We agree that this is a promising direction, and we will systematically investigate and develop suitable additives for this perovskite system in our future work.

Figure R1 J-V curves of the DCI-treated perovskite solar cells based on 4PADCB selective contact.

Our response:

J-V curves of the DCI-treated perovskite solar cells based on 4PADCB hole transport layer had been added to Fig. S40.

On page 16, line 18, “Finally, DCI was successfully applied to 1.68 eV MA-free and 1.79 eV wide-bandgap perovskites, yielding PCEs of 22.9% and 19.4%, respectively (Fig. S40)” had been revised to “Finally, DCI was successfully applied to 4PADCB-based, 1.68 eV MA-free and 1.79 eV wide-bandgap perovskites, yielding PCEs of 22.9%, 22.9% and 19.4% (Fig. S40), respectively, corroborating its universality in various perovskite devices.” had been added.

4. It is important to note that neither the single-junction nor the tandem solar cell efficiencies reported in the manuscript have been independently certified. Certification is a critical benchmark for validating photovoltaic performance, particularly in high-impact journals.

Our reply: We appreciate the reviewer's valuable suggestion. In response, we submitted our tandem solar cells to a third-party accredited laboratory

(Ningbo New Materials Testing and Evaluation Center, NBMTEC) for certification in accordance with IEC-60904 and IEC-60891 protocols.

The measurement procedures are summarized below (the detail is shown in the report of NBMTEC):

1. The solar simulator was calibrated using a standard reference device.
2. The current density–voltage (J–V) characteristics and stabilized power output were measured under calibrated illumination. Reflective surfaces within the illuminated area, including the copper plate, were masked with a black cover.
3. The external quantum efficiency (EQE) of the same tandem device was measured for both perovskite and silicon sub-cells after spectrometer calibration, with appropriate masking applied.
4. The mask area used for J–V, stabilized PCE, and EQE measurements was verified.
5. Spectral mismatch factors were calculated based on the spectral responses of the tandem and reference devices, as well as the simulator spectrum. These factors, along with the mask area, were used to correct the J_{sc} of the tandem device.

As recommended, we certified the PCE of the DCI-treated tandem solar cells. As shown in **Figure R2**, the device achieved a stabilized efficiency of 31.1% (under 5 min steady-state MPP tracking) with no significant performance drop. J–V measurements yielded a PCE of 30.9%, with a V_{oc} of 1.913 V, J_{sc} of 20.7 mA cm^{-2} , and FF of 78.0% under forward scan. Under reverse scan, the device exhibited minimal hysteresis with a PCE of 31.4%, V_{oc} of 1.915 V, J_{sc} of 20.8 mA cm^{-2} , and FF of 78.8%.

Figure R2 Certificated results of perovskite/silicon tandem from an accredited photovoltaic certification laboratory (NBMTEC, China).

Our response:

Certificated results of DCI-treated perovskite/silicon tandem solar cells had been added to Fig. S55.

On page 19, line 23, “the polarized DCI-treated tandem obtains an improved PCE of 30.9% with a V_{oc} of 1.903 V, an FF of 79.7% and a J_{sc} of 20.4 mA cm^{-2} under forward $J-V$ scan, and PCE of 31.1% with a V_{oc} of 1.905 V, an FF of 80.1% and a J_{sc} of 20.4 mA cm^{-2} under reverse $J-V$ scan, exhibiting negligible hysteresis” had been revised to “the polarized DCI-treated tandem obtains an improved PCE of 30.9% with a V_{oc} of 1.903 V, an FF of 79.7% and a J_{sc} of 20.4 mA cm^{-2} under forward $J-V$ scan, and PCE of 31.1% with a V_{oc}

of 1.905 V, an FF of 80.1% and a J_{sc} of 20.4 mA cm⁻² under reverse $J-V$ scan, exhibiting negligible hysteresis and a certified steady-state PCE of 31.1% (Fig. S55).".

Reviewer #3:

The authors have carefully addressed the reviewer's comments point by point and have made adequate revisions accordingly. I believe this version of the manuscript is suitable for publication.

Our response: We sincerely thank the reviewer for the positive recommendation.

Reviewer #4:

The authors have responded to all comments properly and added more detail data and relevant explanation. Now it is ready for publication in this journal.

Our response:

We sincerely appreciate the reviewer for their acceptance and support of the publication of our article.